# Employment of Fracture Mechanics Criteria for Accurate Assessment of the Full Set of Elastic Constants of Orthorhombic/Tetragonal Mono-Crystalline YBCO

## Reaz A. Chaudhuri

Department of Materials Science and Engineering, 122 S, Central Campus Dr., Room 304, University of Utah, Salt Lake City, UT 84112-0560, USA; r.chaudhuri@utah.edu; Tel.: +1-(801)-550-0661

**Abstract:** The effect of elastic constants, $c_{ij}$, on the nature (easy or difficult) of a cleavage system in mono-crystalline $YBa_2Cu_3O_{7-\delta}$ is investigated by employing a novel three-dimensional eigenfunction expansion technique, based in part on the separation of the thickness variable and partly on a modified Frobenius-type series expansion technique in conjunction with Eshelby–Stroh formalism. Out of the three available, complete sets of elastic constants, only the experimental measurements using resonant ultrasound spectroscopy merit serious attention, despite reported values of $c_{12}$ and, to a lesser extent, $c_{66}$ being excessively high. The present investigation considers six through-thickness crack systems weakening orthorhombic mono-crystalline Yttrium barium copper oxide (YBCO) plates. More importantly, the present investigation establishes sufficient conditions for crack path stability/instability, which entail a cleavage system being easy or difficult, i.e., whether a crack would propagate in its original plane/direction or deflect to a different one. This criterion of fracture mechanics is then employed for accurate determination of the full set of elastic constants of superconducting mono-crystalline YBCO. Finally, heretofore unavailable results pertaining to the through-thickness variations of stress intensity factors and energy release rates for a crack corresponding to symmetric and skew-symmetric hyperbolic cosine loads, which also satisfy the boundary conditions on the plate surfaces, bridge a longstanding gap.

**Keywords:** three-dimensional stress singularity; fracture mechanics criterion; sufficient condition for fracture; easy cleavage system; orthorhombic single crystal; elastic constants of mono-crystalline YBCO





## 1. Introduction

The elastic constants of engineering materials are crucial for understanding the deformation and failure behaviors of structural components. From a microscopic perspective, their importance arises from their intimate relationships to such solid-state phenomena as specific heat, Debye temperature ($\Theta_D$), and the Grünelsen parameter [1].

Modern applications of mono-crystalline high-Tc (critical temperature) superconductors (HTS), discovered by Bednorz and Muller [2], include Josephson junctions, which can act as a switch for magnetic fields, or alternatively, perform the function of a magnetic device, such as superconducting quantum interference devices (SQUID) [3]. The $\Theta_D$ of a superconductor such as mono-crystalline YBCO ($YBa_2Cu_3O_{7-\delta}$) can be determined from the knowledge of the elastic constants, $c_{ij}$, in a manner described by Equations (2) and (3) of Lei et al. [1], which, in combination with the electron–phonon coupling parameter, $\lambda^*$ can be used to compute the superconducting transition temperature, $T_c$ [4,5].

The poor fracture toughness of these HTS restricts their practical utility to a cryogenic temperature range [6–14]. A review of the Literature reveals that studies of singular stress fields at the tips of cracks and notches are mostly limited to two-dimensionality [15–20], and primarily employ the Lekhnitskii [15] (and also occasionally "Stroh" [16]) -based methodology; see Nejati et al. [21] for an extensive survey of the Literature. Three-dimensional (3D)

fracture toughness studies have been, until recently [22], marked by their near-complete absence, because of being fraught with an extraordinary level of analytical difficulties. Furthermore, there is no three-dimensional analog of the complex variable theory, the next one in line being the four-dimensional space-based quaternion theory [23]. This aspect of the development of mathematical theories calls for an innovative approach as investigated in what follows.

Stenger et al. [24] have discussed the lack of accuracy of popular numerical methods, such as finite difference, finite elements, and boundary elements in the immediate vicinity of a crack or wedge front. Various categories of three-dimensional stress singularities include: (i) through-thickness crack/anti-crack [25,26], as well as their bi- and tri-material interface counterparts [27–29]; (ii) corresponding wedges/notches [30–34]; (iii) bi-material free/fixed straight edge-face [35–37]; (iv) tri-material junction [38]; (v) penny-shaped crack/anti-crack [39,40] and their bi-material interface counterparts [41,42]; (vi) through/part-through hole/rigid inclusion [43,44] and their bi-material counterparts [45,46], as well as elastic inclusion [47,48]; (vii) fiber-matrix interfacial debond [49–51]; (viii) fiber breaks and matrix cracking in composites [52]; (ix) interfacial bond line of a tapered jointed plate [53]; and (x) circumferential junction corner line of an island/substrate [54]; among others. A review of the Literature regarding three-dimensional fracture mechanics reveals successful attempts at analyses of the penny-shaped crack/anti-crack [39] (and their bi-material counterparts [41]), and the hole [43], bi-material hole [45], and inclusion problems [47]. This success notwithstanding, the solution of the corresponding through-thickness crack problem was engulfed in controversies [25,55].

More recently, cracked/anti-cracked transversely isotropic (smeared-out composite) [56] as well as cubic/orthorhombic/diamond cubic mono-crystalline plates subjected to mode I/II far-field loadings [22,57,58] and cubic/orthorhombic/monoclinic/diamond cubic mono/tri-crystalline plates under mode III loading [22,58–61] have been solved by a novel three-dimensional eigenfunction expansion technique, based in part on the separation of the thickness variable and partly on a modified Frobenius type series expansion technique in conjunction with Eshelby [62]–Stroh [16] formalism. Three-dimensional asymptotic stress fields in the vicinity of the front of the kinked carbon fiber-matrix junction [63] have also been derived by employment of the same approach (see also Ref. [64] for its 2D counterpart with isotropic glass fibers).

The above-mentioned importance notwithstanding, relatively fewer attempts at experimental determination of elastic constants of mono-crystalline YBCO have been reported in the Literature about solid-state physics [1,65–73]; these are summarized in Table 1 of Lei et al. [1]. Golding et al. [66] and Saint-Paul and co-workers [70,71] have reported experimental results on $c_{11}$ and $c_{33}$, and $c_{33}$, $c_{44}$, $c_{66}$, and $c_{12}$, respectively, by employing the ultrasound technique, while Baumgart et al. [68,69] and Zouboulis et al. [72] have resorted to Brillouin spectroscopy/scattering to determine $c_{11}$, $c_{33}$ and $c_{44}$, and $c_{44}$, $c_{55}$, and $c_{66}$, respectively. Only two experimental investigations [1,67] report complete sets of elastic constants accessible to the present author. Worse still, those reported by Reichard et al. [67] assume tetragonal symmetry; see Table 1. Only the experimental measurements due to Lei et al. [1], marked * in Table 1, correctly assume orthorhombic symmetry, and were determined by resonant ultrasound spectroscopy, described in detail by Migliori et al. [74,75]. However, their measured magnitude of $c_{12}$ was deemed to be unacceptably out of the range. The reasoning given by the authors [1] is that "No wave speed in the crystal depends only upon $c_{12}$, it is no way to estimate it directly." It was previously experimentally determined by the present author and his co-workers [76–78] that $c_{12}$ and $c_{66}$ are always coupled in vibrations-based measurements [77,78], although these elastic stiffnesses can be measured independently by static tests. Finally, those reported by Ledbetter and Lei [79] are just estimates (marked ** in Table 1).

**Table 1.** Elastic stiffness constants of YBCO single crystals.

| Material (Technique) | $c_{11}$ (GPa) | $c_{12}$ (GPa) | $c_{13}$ (GPa) | $c_{22}$ (GPa) | $c_{23}$ (GPa) | $c_{33}$ (GPa) | $c_{44}$ (GPa) | $c_{55}$ (GPa) | $c_{66}$ (GPa) |
|---|---|---|---|---|---|---|---|---|---|
| YBCO * [1] (Resonant Ultrasound) | 231.0 | 132.0 | 71.0 | 268.0 | 95.0 | 186.0 | 49.0 | 37.0 | 95.0 |
| YBCO ** [79] (Estimate) | 223.0 | 37.0 | 89.0 | 244.0 | 93.0 | 138.0 | 61.0 | 47.0 | 97.0 |
| YBCO *** (Inference) | 231.0 | 66.0 | 71.0 | 268.0 | 95.0 | 186.0 | 49.0 | 37.0 | 82.0 |
| YBCO$^{T}$ [67] (Neutron Scattering) | 230.0 | 100.0 | 100.0 | 230.0 | 100.0 | 150.0 | 50.0 | 50.0 | 85.0 |

* All values measured by resonant ultrasound spectroscopy (except $c_{12}$) by Lei et al. [1]. ** Estimated by Ledbetter and Lei [79]. *** Same as *, except $c_{12}$ and $c_{66}$ measured by ultrasound by Saint-Paul and Henry [71].

The above review of the literature reveals an absence of reliable and accurate experimentally measured complete sets of the nine elastic constants needed for characterization of the deformation/fracture, as well as other solid-state (e.g., $\Theta_D$, $T_c$, etc.) behaviors of superconducting (orthorhombic) YBCO single crystals. This calls for a reliable criterion for assessment of the measured data that would allow us to come up with a reasonably accurate complete set of nine elastic constants. This is the primary objective of the present investigation. One effective way to address this important issue is to analytically examine the effects of elastic constants on crack path stability/instability in mono-crystalline YBCO and compare them with the experimental results for easy cleavage planes, reported by Cook et al. [6], Raynes [9], and Granozio and di Uccio [14], among others. In what follows, the above-mentioned modified eigenfunction expansion technique, based in part on the separation of the thickness variable and partly on the Eshelby–Stroh type affine transformation, is developed to derive a three-dimensional asymptotic stress field in the vicinity of the front of a semi-infinite through-thickness crack weakening an infinite orthorhombic mono-crystalline plate of finite thickness and subjected to far-field mode I/II loadings. Crack-face boundary conditions and those that are prescribed on the top and bottom (free or fixed) surfaces of the plate are exactly satisfied. The present investigation considers six through-crack systems—(010)[001] with the [100] length direction, (0$\bar{1}$0)[100] with the [001] length direction, ($\bar{1}$00)[001] with the [010] length direction, (100)[010] with the [001] length direction, (001)[0$\bar{1}$0] with the [100] length direction, and (001)[100] with the [010] length direction—weakening orthorhombic mono-crystalline plates. Explicit expressions for the singular stresses in the vicinity of the front of a through-thickness crack weakening an orthorhombic mono-crystalline plate, subjected to far-field mode I/II loadings, are presented. In addition, through-thickness distribution of the stress intensity factors, and energy release rates are also presented.

Next, the important issue of easy or difficult cleavage plane and the related question of crack deflection criterion is discussed. The latter is based on the relative fracture energy (or the energy release rate) available for possible "fracture paths" [18]. This said, it is noteworthy that the Griffith energy balance-based criterion cannot be regarded as a sufficient condition [80]. This is because Griffith's criterion is "Not really a fracture criterion but only a necessary condition for fracture" [80]. This calls for establishment of a sufficient condition for determination of an easy or difficult cleavage plane, and the related question of the crack path stability/instability criterion. This is the second and somewhat more important objective of the present study.

One Holy Grail issue in fracture mechanics of anisotropic solids is to find a dimensionless parameter akin to Reynold's number in fluid flow problems, crossing a critical value which signifies transition from one regime to another, such as the critical value of Reynold's number, above which the flow is turbulent and below which it is laminar. It is an attendant issue relating to crack deflection that remains heretofore unaddressed, which is the third objective of the present investigation. In a similar vein, just as the introduction of Reynold's number facilitated the design and setting up of experiments in addition to experimental verification of analytical and computational solutions in fluid dynamics, a similar expectation-cum-need exists in the important field of fracture mechanics of anisotropic solids, which is the final objective of the present study. Here, the accuracy and efficacy of the available experimental results on the elastic constants of YBCO single crystals, measured by modern experimental techniques with resolutions at the atomic scale (or nearly so), such as X-ray diffraction [65], the ultrasound technique [66,70,71], neutron diffraction [73]/scattering [67], Brillouin spectroscopy [68,69] /scattering [72], resonant ultrasound spectroscopy [1,74,75], and the like, is assessed with a powerful theoretical analysis on crack path stability/instability, in part based on a dimensionless parameter, such as the planar anisotropic ratio.

The present study, although to a smaller extent a review of earlier work on this topic, is largely based on original research on this subject. The topic, which covers mathematics (e.g., branch point/branch cut, asymptotic, solution to 3D mixed boundary-value problem, and necessary and sufficient condition for fracture), solid-state physics/chemistry (e.g., stoichiometry, superconductivity, crystal structures, and single crystal cleavage) and engineering (e.g., 3D fracture mechanics), has, so far, remained largely unexplored in the Literature.

## 2. Formulation of the Problem

One of the most important cleavage systems for mono-crystalline orthorhombic YBCO is (001) [100] × [010] ({crack plane}<crack front>x<initial propagation direction>). In what follows, the deformation behavior in the vicinity of the front of a semi-infinite through-thickness crack, weakening an infinite orthorhombic YBCO plate of thickness, 2 h (Figures 1 and 2) is analyzed in detail. Here, the $\tilde{z}$-axis is placed along the straight crack front, [100], while the coordinates $\tilde{x}$ [010], $\tilde{y}$ [001] are used to define the directions along the length of the crack (propagation direction) and the direction transverse to it, respectively, in the middle plane of the single crystal plate. $\tilde{u}$, $\tilde{v}$ and $\tilde{w}$ represent the components of the displacements in $\tilde{x}$ [010], $\tilde{y}$ [001] and $\tilde{z}$ [100] directions, respectively. The corresponding stress-strain relations are given as follows:

$$\begin{Bmatrix} \widetilde{\sigma}_{\widetilde{x}} \\ \widetilde{\sigma}_{\widetilde{y}} \\ \widetilde{\sigma}_{\widetilde{z}} \\ \widetilde{\tau}_{\widetilde{y}\widetilde{z}} \\ \widetilde{\tau}_{\widetilde{x}\widetilde{z}} \\ \widetilde{\tau}_{\widetilde{x}\widetilde{y}} \end{Bmatrix} = \begin{bmatrix} c_{22} & c_{23} & c_{12} & 0 & 0 & 0 \\ c_{23} & c_{33} & c_{13} & 0 & 0 & 0 \\ c_{12} & c_{13} & c_{11} & 0 & 0 & 0 \\ 0 & 0 & 0 & c_{55} & 0 & 0 \\ 0 & 0 & 0 & 0 & c_{66} & 0 \\ 0 & 0 & 0 & 0 & 0 & c_{44} \end{bmatrix} \begin{Bmatrix} \widetilde{\varepsilon}_{\widetilde{x}} \\ \widetilde{\varepsilon}_{\widetilde{y}} \\ \widetilde{\varepsilon}_{\widetilde{z}} \\ \widetilde{\gamma}_{\widetilde{y}\widetilde{z}} \\ \widetilde{\gamma}_{\widetilde{x}\widetilde{z}} \\ \widetilde{\gamma}_{\widetilde{x}\widetilde{y}} \end{Bmatrix}, \tag{1}$$

where $c_{ij}$, i, j = 1,...,6, denotes the elastic stiffness constants of an orthorhombic mono-crystalline plate. $\widetilde{\sigma}_{\widetilde{x}}, \widetilde{\sigma}_{\widetilde{y}}, \widetilde{\sigma}_{\widetilde{z}}$ represent the normal stresses, and $\widetilde{\tau}_{\widetilde{x}\widetilde{y}}, \widetilde{\tau}_{\widetilde{x}\widetilde{z}}, \widetilde{\tau}_{\widetilde{y}\widetilde{z}}$ denote the shear stresses, while $\widetilde{\varepsilon}_{\widetilde{x}}, \widetilde{\varepsilon}_{\widetilde{y}}, \widetilde{\varepsilon}_{\widetilde{z}}$ denote normal strains, and $\widetilde{\gamma}_{\widetilde{x}\widetilde{y}} \widetilde{\gamma}_{\widetilde{x}\widetilde{z}}, \widetilde{\gamma}_{\widetilde{y}\widetilde{z}}$ represent the shear strains. For the special case of a tetragonal single crystal,

$$c_{11} = c_{22}, c_{13} = c_{23}, c_{44} = c_{55}. \tag{2}$$

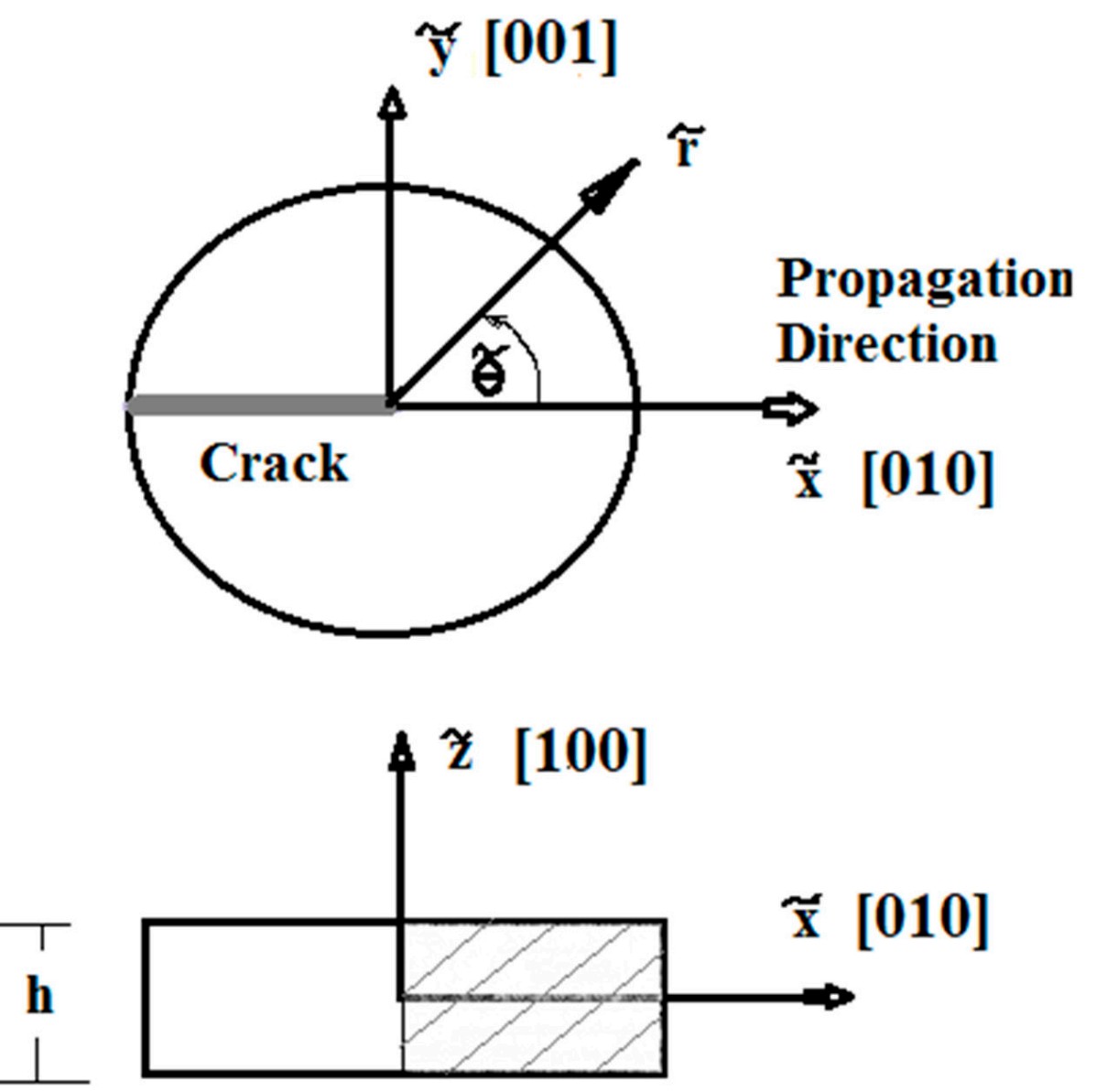

**Figure 1.** Schematic of a through-thickness semi-infinite crack in an infinite orthorhombic mono-crystalline plate.

The three equilibrium equations can now be expressed in terms of the displacement components, $\tilde{u}$, $\tilde{v}$, and $\tilde{w}$, as follows:

$$c_{22}\frac{\partial^2 \tilde{u}}{\partial \tilde{x}^2} + c_{44}\frac{\partial^2 \tilde{u}}{\partial \tilde{y}^2} + c_{66}\frac{\partial^2 \tilde{u}}{\partial \tilde{z}^2} + (c_{23} + c_{44})\frac{\partial^2 \tilde{v}}{\partial \tilde{x}\partial \tilde{y}} + (c_{12} + c_{66})\frac{\partial^2 \tilde{w}}{\partial \tilde{x}\partial \tilde{z}} = 0, \tag{3}$$

$$(c_{23} + c_{44})\frac{\partial^2 \tilde{u}}{\partial \tilde{x}\partial \tilde{y}} + c_{44}\frac{\partial^2 \tilde{v}}{\partial \tilde{x}^2} + c_{33}\frac{\partial^2 \tilde{v}}{\partial \tilde{y}^2} + c_{55}\frac{\partial^2 \tilde{v}}{\partial \tilde{z}^2} + (c_{13} + c_{55})\frac{\partial^2 \tilde{w}}{\partial \tilde{y}\partial \tilde{z}} = 0, \tag{4}$$

$$(c_{12} + c_{66})\frac{\partial^2 \tilde{u}}{\partial \tilde{x}\partial \tilde{z}} + (c_{13} + c_{55})\frac{\partial^2 \tilde{v}}{\partial \tilde{y}\partial \tilde{z}} + c_{66}\frac{\partial^2 \tilde{w}}{\partial \tilde{x}^2} + c_{55}\frac{\partial^2 \tilde{w}}{\partial \tilde{y}^2} + c_{11}\frac{\partial^2 \tilde{w}}{\partial \tilde{z}^2} = 0, \tag{5}$$

The boundary conditions include those at the plate faces and crack-side surfaces. The boundary conditions on the plate faces, $\tilde{z} = \pm h$, are given by [22,25]

$$\tilde{\sigma}_{\tilde{z}} = \tilde{\tau}_{\tilde{x}\tilde{z}} = \tilde{\tau}_{\tilde{y}\tilde{z}} = 0, \tag{6}$$

while those at the crack-side surfaces are more conveniently expressed in local cylindrical polar coordinates (Figure 1), which are given as follows:

$$\tilde{\sigma}_{\tilde{\theta}} = \tilde{\tau}_{\tilde{r}\tilde{\theta}} = \tilde{\tau}_{\tilde{\theta}\tilde{z}} = 0, \tag{7}$$

$$\tilde{\theta} = \pm \pi \tag{8}$$

where $\tilde{\sigma}_{\tilde{r}}$, $\tilde{\sigma}_{\tilde{\theta}}$, $\tilde{\sigma}_{\tilde{r}}$ represent the normal stresses, and $\tilde{\tau}_{\tilde{r}\tilde{\theta}}$, $\tilde{\tau}_{\tilde{r}\tilde{z}}$, $\tilde{\tau}_{\tilde{\theta}\tilde{z}}$ are the shear stresses, while $\tilde{\varepsilon}_{\tilde{r}}$, $\tilde{\varepsilon}_{\tilde{\theta}}$, $\tilde{\varepsilon}_{\tilde{z}}$ denote the normal strains, and $\tilde{\gamma}_{\tilde{r}\tilde{\theta}}$, $\tilde{\gamma}_{\tilde{r}\tilde{z}}$, $\tilde{\gamma}_{\tilde{\theta}\tilde{z}}$ are the shear strains in the cylindrical polar coordinate system ($\tilde{r}$, $\tilde{\theta}$, $\tilde{z}$). $\tilde{u}_{\tilde{r}}$ and $\tilde{u}_{\tilde{\theta}}$ represent the components of the displacement in $\tilde{r}$ and $\tilde{\theta}$ directions, respectively.

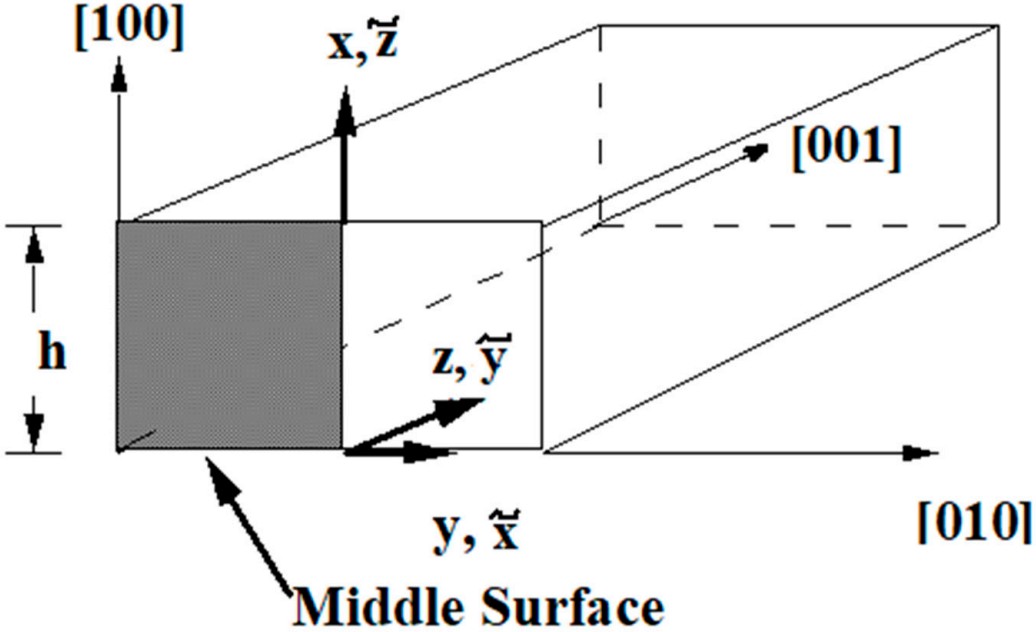

**Figure 2.** Schematic of the top half of an orthorhombic mono-crystalline plate weakened by a (001) [100] through-thickness crack initially propagating in [010] direction.

### 3. Singular Stress Fields in the Vicinity of a Crack Front Weakening an Orthotropic/Orthorhombic Lamina/Single Crystal under General Loading

The assumed displacement functions for the three-dimensional crack problem under consideration are selected on the basis of the separation of $\widetilde{z}$ variables. These are as given below [22,56–58,64]:

$$\widetilde{u}(\widetilde{x},\widetilde{y},\widetilde{z}) = e^{i\widetilde{k}\widetilde{z}}\widetilde{U}(\widetilde{x},\widetilde{y}), \tag{9}$$

$$\widetilde{v}(\widetilde{x},\widetilde{y},\widetilde{z}) = e^{i\widetilde{k}\widetilde{z}}\widetilde{V}(\widetilde{x},\widetilde{y}), \tag{10}$$

$$\widetilde{w}(\widetilde{x},\widetilde{y},\widetilde{z}) = e^{i\widetilde{k}\widetilde{z}}\widetilde{W}(\widetilde{x},\widetilde{y}). \tag{11}$$

It may be noted that since the $\widetilde{z}$-dependent term and its first partial derivative can either be bounded and integrable at most admitting ordinary discontinuities, or the first partial derivative at worst be square integrable (in the sense of Lebesgue integration) in its interval $\widetilde{z} \in [-h, h]$, i.e., admitting singularities weaker than square root (i.e., $\widetilde{z}^{(-1/2+\varepsilon)}$, $\varepsilon$), it can be best represented by a Fourier series [22,25,58]. The latter case is justified by Parseval's theorem [81], and its physical implication is that of satisfying the criterion of finiteness of local strain energy and path independence [82]. Substitution of Equations (9)–(11) into Equations (2)–(5) yields the following system of coupled partial differential equations (PDE's):

$$c_{22}\frac{\partial^2 \widetilde{U}}{\partial \widetilde{x}_1^2} + c_{44}\frac{\partial^2 \widetilde{U}}{\partial \widetilde{y}_1^2} + c_{66}\widetilde{U} + (c_{23} + c_{44})\frac{\partial^2 \widetilde{V}}{\partial \widetilde{x}_1 \partial \widetilde{y}_1} + (c_{12} + c_{66})\frac{\partial \widetilde{W}}{\partial \widetilde{x}_1} = 0, \tag{12}$$

$$(c_{23} + c_{44})\frac{\partial^2 \widetilde{U}}{\partial \widetilde{x}_1 \partial \widetilde{y}_1} + c_{44}\frac{\partial^2 \widetilde{V}}{\partial \widetilde{x}_1^2} + c_{33}\frac{\partial^2 \widetilde{V}}{\partial \widetilde{y}_1^2} + c_{55}\widetilde{V} + (c_{13} + c_{55})\frac{\partial \widetilde{W}}{\partial \widetilde{y}_1} = 0, \tag{13}$$

$$(c_{12} + c_{66})\frac{\partial \widetilde{U}}{\partial \widetilde{x}_1} + (c_{13} + c_{55})\frac{\partial \widetilde{V}}{\partial \widetilde{y}_1} + c_{66}\frac{\partial^2 \widetilde{W}}{\partial \widetilde{x}_1^2} + c_{55}\frac{\partial^2 \widetilde{W}}{\partial \widetilde{y}_1^2} + c_{11}\widetilde{W} = 0, \tag{14}$$

where

$$\widetilde{x}_1 = i\widetilde{k}\widetilde{x}, \tag{15}$$

$$\widetilde{y}_1 = i\widetilde{k}\widetilde{y}. \tag{16}$$

The solution to the system of coupled partial differential Equations (12)–(14) subjected to the most general loading can now be sought in the form of the following modified Frobenius type series in terms of the variable $\widetilde{x}_1 + \widetilde{p}\widetilde{y}_1$ as follows:

$$\widetilde{U}(\widetilde{x}_1, \widetilde{y}_1) = \sum_{n=0}^{\infty} \widetilde{a}'_{\widetilde{s}+n}(\widetilde{x}_1 + \widetilde{p}\widetilde{y}_1)^{\widetilde{s}+2n+1} + \sum_{n=0}^{\infty} \widetilde{a}_{\widetilde{s}+n}(\widetilde{x}_1 + \widetilde{p}\widetilde{y}_1)^{\widetilde{s}+2n}, \tag{17}$$

$$\widetilde{V}(\widetilde{x}_1, \widetilde{y}_1) = \sum_{n=0}^{\infty} \widetilde{b}'_{\widetilde{s}+n}(\widetilde{x}_1 + \widetilde{p}\widetilde{y}_1)^{\widetilde{s}+2n+1} + \sum_{n=0}^{\infty} \widetilde{b}_{\widetilde{s}+n}(\widetilde{x}_1 + \widetilde{p}\widetilde{y}_1)^{\widetilde{s}+2n}, \tag{18}$$

$$\widetilde{W}(\widetilde{x}_1, \widetilde{y}_1) = \sum_{n=0}^{\infty} \widetilde{c}'_{\widetilde{s}+n}(\widetilde{x}_1 + \widetilde{p}\widetilde{y}_1)^{\widetilde{s}+2n} + \sum_{n=0}^{\infty} \widetilde{c}_{\widetilde{s}+n}(\widetilde{x}_1 + \widetilde{p}\widetilde{y}_1)^{\widetilde{s}+2n+1}. \tag{19}$$

Out of the various combinations, such as (a′, b′, c′), (a, b, c), (a′, b, c), (a, b′, c), (a, b, c′), (a′, b′, c), (a′, b, c′), and (a, b′, c′), only the first two groupings can produce meaningful solutions, for the mode I/II and mode III loading cases, respectively. This step permits the separation of mode III from modes I/II. The first grouping is described below, while the second one has already been employed for the anti-plane shear case [59–61].

### 4. Singular Stress Fields in the Vicinity of a (001)[100] Through-Crack Front Propagating under Mode I (Extension/Bending) and Mode II (Sliding Shear/Twisting) in [010] Direction

While the separability of the $\widetilde{x}_1$ and $\widetilde{y}_1$ variables is the hallmark of the derivation of the singular stress fields weakening an isotropic plate [25–49,55] subjected to the far-field mode I/II loading, the same cannot be true for its orthorhombic counterpart, wherein the solution to the system of coupled partial differential Equation (5) must be sought in the form of a modified Frobenius type series in terms of a combined variable $\widetilde{x}_1 + \widetilde{p}\widetilde{y}_1$, which is a mathematical representation of an affine transformation employed earlier by Eshelby et al. [62], Stroh [16], and Shih et al. [17], as shown below [22,53,54,60]. It may be remarked here that the solution methodology offered by these authors, especially Refs. [17,62], are entirely different from what follows.

$$\widetilde{U}(\widetilde{x}_1, \widetilde{y}_1) = \sum_{n=0}^{\infty} \widetilde{a}_{\widetilde{s}+n}(\widetilde{x}_1 + \widetilde{p}\widetilde{y}_1)^{\widetilde{s}+2n}, \tag{20}$$

$$\widetilde{V}(\widetilde{x}_1, \widetilde{y}_1) = \sum_{n=0}^{\infty} \widetilde{b}_{\widetilde{s}+n}(\widetilde{x}_1 + \widetilde{p}\widetilde{y}_1)^{\widetilde{s}+2n}, \tag{21}$$

$$\widetilde{W}(\widetilde{x}_1, \widetilde{y}_1) = \sum_{n=0}^{\infty} \widetilde{c}_{\widetilde{s}+n}(\widetilde{x}_1 + \widetilde{p}\widetilde{y}_1)^{\widetilde{s}+2n+1}. \tag{22}$$

On substitution of Equations (20)–(22) into Equations (12)–(14), and equating the coefficients of $(\widetilde{x}_1 + \widetilde{p}\widetilde{y}_1)^{\widetilde{s}+2n-2}$, a set of recurrent relationships can be derived, which, for $n = 0$, results in the following:

$$\begin{bmatrix} c_{22} + c_{44}\widetilde{p}^2 & (c_{23} + c_{44})\widetilde{p} \\ (c_{23} + c_{44})\widetilde{p} & c_{33}\widetilde{p}^2 + c_{44} \end{bmatrix} \begin{Bmatrix} \widetilde{a}_{\widetilde{s}} \\ \widetilde{b}_{\widetilde{s}} \end{Bmatrix} = \begin{Bmatrix} 0 \\ 0 \end{Bmatrix}, \text{ valid for } \widetilde{s} \neq 0, 1, \tag{23}$$

which, in turn, yields, for the non-trivial case, the characteristic equation for the coupled partial differential Equations (3)–(5) or (12)–(14) as given below:

$$\widetilde{p}^4 + 2\widetilde{\chi}\widetilde{p}^2 + \frac{c_{22}}{c_{33}} = 0, \tag{24}$$

where the normalized elastic parameter, $\widetilde{\kappa} = 1/\widetilde{\chi}$, is given by

$$\widetilde{\kappa} = \frac{1}{\widetilde{\chi}} = \frac{2c_{33}c_{44}}{\left(c_{22}c_{33} - c_{23}^2 - 2c_{23}c_{44}\right)} = \frac{2G_{23}(1 - \nu_{13}\nu_{31})}{[E_3 - 2G_{23}(\nu_{32} + \nu_{12}\nu_{13})]}, \tag{25}$$

in which $E_3$ is z-direction Young's modulus, $G_{23}$ is the shear modulus in the y-z plane, while $\nu_{12}$ is the major Poisson's ratio in the x-y plane. $\nu_{32}$ denotes the minor Poisson's ratio in the y-z plane, while $\nu_{13}$ and $\nu_{31}$ represent the major and minor Poisson's ratios, respectively, in the x-z plane. It also is sometimes convenient to relate $\widetilde{\kappa} = 1/\widetilde{\chi}$, as will be seen later, to $\widetilde{A} = 1/\widetilde{\lambda}$, with $\widetilde{A}$ being the planar anisotropic ratio (in the $\widetilde{x}$ [010]-$\widetilde{y}$ [001] plane), as shown below.

$$\widetilde{\kappa} = \frac{1}{\widetilde{\chi}} = \frac{\widetilde{A}c_{33}}{\sqrt{c_{22}c_{33}} + c_{23}(1 - \widetilde{A})}, \tag{26}$$

in which

$$\widetilde{A} = \frac{2c_{44}}{\sqrt{c_{22}c_{33}} - c_{23}}. \tag{27}$$

It can easily be seen from Equation (27) that $\widetilde{A}$ is higher when the shear stiffness (modulus) and major Poisson's ratio in the $\widetilde{x}$ [010], $\widetilde{y}$ [001] plane assume larger magnitudes. This simple fact assumes great importance as this investigation aims to solve one Holy Grail issue in the fracture mechanics of anisotropic media, of coming up with a dimensionless parameter akin to Reynold's number in fluid flow problems, crossing a critical value which signifies transition from one regime to another, such as the critical value of Reynold's number above which the flow is turbulent and below which it is laminar. It is an attendant issue relating to crack deflection in mono-crystalline orthorhombic YBCO.

Equation (24) has either (a) four complex or (b) four imaginary roots, depending on whether

$$\text{(a) } \widetilde{A} > 1 \text{ or, equivalently, } \widetilde{\kappa} > \sqrt{\frac{c_{33}}{c_{22}}} = \sqrt{\frac{E_3(1 - \nu_{12}\nu_{21})}{E_2(1 - \nu_{13}\nu_{31})}}, \tag{28}$$

$$\text{(b) } \widetilde{A} < 1 \text{ or, equivalently, } \widetilde{\kappa} < \sqrt{\frac{c_{33}}{c_{22}}} = \sqrt{\frac{E_3(1 - \nu_{12}\nu_{21})}{E_2(1 - \nu_{13}\nu_{31})}}. \tag{29}$$

Chaudhuri and Xie [25], and Chaudhuri [55] have earlier solved the degenerate isotropic material case, for which $\widetilde{A} = 1$ and $\widetilde{\kappa} = 1$.

*4.1. Case (a): Complex Roots*

$$\widetilde{p}_{1,2} = \widetilde{\xi} \pm i\widetilde{\eta}, \tag{30}$$

$$\widetilde{p}_{3,4} = -\widetilde{\xi} \pm i\widetilde{\eta}, \tag{31}$$

where

$$\widetilde{\xi} = \frac{1}{\sqrt{2}} \left[ \left( \frac{c_{22}}{c_{33}} \right)^{1/2} - \widetilde{\chi} \right]^{1/2}, \tag{32}$$

$$\widetilde{\eta} = \frac{1}{\sqrt{2}} \left[ \left( \frac{c_{22}}{c_{33}} \right)^{1/2} + \widetilde{\chi} \right]^{1/2}, \tag{33}$$

valid for $\widetilde{A} > 1$ or, equivalently, $\widetilde{\kappa} > \sqrt{c_{33}/c_{22}}$.

It may be noted here that a crack front and the corresponding semi-infinite crack represent a branch point and a branch cut, respectively. Therefore, the general asymptotic form for the displacement and stress fields in the vicinity of the crack front can most conveniently be obtained by employing a set of two polar coordinate systems, $(\widetilde{\rho}, \widetilde{\psi})$ and $(\widetilde{\rho}', \widetilde{\psi}')$, derived from the aforementioned affine transformation and expressed in terms of the cylindrical polar coordinate system $(\widetilde{r}, \widetilde{\theta}, \widetilde{z})$, as follows:

$$\widetilde{\rho} \cos\left(\widetilde{\psi}\right) = \widetilde{r}\left( \cos(\widetilde{\theta}) + \widetilde{\xi} \sin(\widetilde{\theta}) \right), \quad \widetilde{\rho} \sin\left(\widetilde{\psi}\right) = \widetilde{r}\left( \widetilde{\eta} \sin(\widetilde{\theta}) \right), \tag{34}$$

$$\widetilde{\rho}' \cos\left(\widetilde{\psi}'\right) = \widetilde{r}\left( \cos(\widetilde{\theta}) - \widetilde{\xi} \sin(\widetilde{\theta}) \right), \quad \widetilde{\rho}' \sin\left(\widetilde{\psi}'\right) = \widetilde{r}\left( -\widetilde{\eta} \sin(\widetilde{\theta}) \right), \tag{35}$$

where

$$\widetilde{\rho} = \widetilde{r}\left\{ \left( \cos(\widetilde{\theta}) + \widetilde{\xi} \sin(\widetilde{\theta}) \right)^2 + \widetilde{\eta}^2 \sin^2(\widetilde{\theta}) \right\}^{1/2}, \tag{36}$$

$$\widetilde{\rho}' = \widetilde{r}\left\{ \left( \cos(\widetilde{\theta}) - \xi \sin(\widetilde{\theta}) \right)^2 + \widetilde{\eta}^2 \sin^2(\widetilde{\theta}) \right\}^{1/2}, \tag{37}$$

and

$$\cos\left( \widetilde{\psi}(\widetilde{\theta}) \right) = \frac{\cos(\widetilde{\theta}) + \widetilde{\xi} \sin(\widetilde{\theta})}{\left\{ \left( \cos(\widetilde{\theta}) + \widetilde{\xi} \sin(\widetilde{\theta}) \right)^2 + \widetilde{\eta}^2 \sin^2(\widetilde{\theta}) \right\}^{1/2}}, \tag{38}$$

$$\sin\left(\widetilde{\psi}(\widetilde{\theta})\right) = \frac{\widetilde{\eta}\sin(\widetilde{\theta})}{\left\{\left(\cos(\widetilde{\theta})+\widetilde{\xi}\sin(\widetilde{\theta})\right)^2+\widetilde{\eta}^2\sin^2(\widetilde{\theta})\right\}^{1/2}}, \tag{39}$$

$$\cos\left(\widetilde{\psi}'(\widetilde{\theta})\right) = \frac{\cos(\widehat{\theta})-\widetilde{\xi}\sin(\widetilde{\theta})}{\left\{\left(\cos(\widetilde{\theta})-\widetilde{\xi}\sin(\widetilde{\theta})\right)^2+\widetilde{\eta}^2\sin^2(\widetilde{\theta})\right\}^{1/2}}, \tag{40}$$

$$\sin\left(\widetilde{\psi}'(\widetilde{\theta})\right) = \frac{-\widetilde{\eta}\sin(\widetilde{\theta})}{\left\{\left(\cos(\widetilde{\theta})-\widetilde{\xi}\sin(\widetilde{\theta})\right)^2+\widetilde{\eta}^2\sin^2(\widetilde{\theta})\right\}^{1/2}}, \tag{41}$$

The components of the displacement vector and the stress tensor in the immediate neighborhood of the crack from can now be written as shown below:

$$\widetilde{u}(\widetilde{r},\widetilde{\theta},\widetilde{z}) = \widetilde{r}^{\widetilde{s}}\widetilde{D}_b(\widetilde{z})\left(i\widetilde{k}\right)^{\widetilde{s}}\left[\left\{\left(\cos(\widetilde{\theta})+\widetilde{\xi}\sin(\widetilde{\theta})\right)^2+\widetilde{\eta}^2\sin^2(\widetilde{\theta})\right\}^{s/2}\left\{\widetilde{A}_1\,\cos(\widetilde{s}\widetilde{\psi})\,+\,\widetilde{A}_2\,\sin(\widetilde{s}\widetilde{\psi})\right\}\right.$$
$$\left.+\left\{\left(\cos(\widetilde{\theta})-\widetilde{\xi}\sin(\widetilde{\theta})\right)^2+\widetilde{\eta}^2\sin^2(\widetilde{\theta})\right\}^{\widetilde{s}/2}\left\{\widetilde{A}_3\,\cos(\widetilde{s}\widetilde{\psi}')\,+\,\widetilde{A}_4\,\sin(\widetilde{s}\widetilde{\psi}')\right\}\right]+O\left(\widetilde{r}^{\widetilde{s}+2}\right), \tag{42}$$

$$\widetilde{v}(\widetilde{r},\widetilde{\theta},\widetilde{z}) = \widetilde{r}^{\widetilde{s}}\widetilde{D}_b(\widetilde{z})\left(i\widetilde{k}\right)^{\widetilde{s}}\left[\left\{\left(\cos(\widetilde{\theta})+\widetilde{\xi}\sin(\widetilde{\theta})\right)^2+\widetilde{\eta}^2\sin^2(\widetilde{\theta})\right\}^{\widetilde{s}/2}\left\{\left(\widetilde{H}_1\widetilde{A}_1\right.\right.\right.$$
$$\left.+\widetilde{H}_2\widetilde{A}_2\right)\cos(\widetilde{s}\widetilde{\psi})+\left(\widetilde{H}_1\widetilde{A}_2-\widetilde{H}_2\widetilde{A}_1\right)\sin(\widetilde{s}\widetilde{\psi})\right\}+\left\{\left(\cos(\widetilde{\theta})-\widetilde{\xi}\sin(\widetilde{\theta})\right)^2\right.$$
$$\left.+\widetilde{\eta}^2\sin^2(\widetilde{\theta})\right\}^{\widetilde{s}/2}\left\{-\left(\widetilde{H}_1\widetilde{A}_3+\widetilde{H}_2\widetilde{A}_4\right)\cos(\widetilde{s}\widetilde{\psi}')-\left(\widetilde{H}_1\widetilde{A}_4-\widetilde{H}_2\widetilde{A}_3\right)\sin(\widetilde{s}\widetilde{\psi}')\right\}$$
$$+O\left(\widetilde{r}^{\widetilde{s}+2}\right), \tag{43}$$

$$\widetilde{w}(\widetilde{r},\widetilde{\theta},\widetilde{z}) = O\left(\widetilde{r}^{\widetilde{s}+1}\right), \tag{44}$$

and

$$\widetilde{\sigma}_{\widetilde{x}}\left(\widetilde{r},\widetilde{\theta},\widetilde{z}\right) = r^{\widetilde{s}-1}\widetilde{D}_b(\widetilde{z})\left(i\widetilde{k}\right)^{\widetilde{s}}\widetilde{s}\left\langle\left\{\left(\cos(\widetilde{\theta})+\widetilde{\xi}\sin(\widetilde{\theta})\right)^2+\widetilde{\eta}^2\sin^2(\widetilde{\theta})\right\}^{(\widetilde{s}-1)/2}\left[\left(\widetilde{A}_1\left\{c_{22}\right.\right.\right.\right.$$
$$\left.+\left(\widetilde{\xi}\widetilde{H}_1-\widetilde{\eta}\widetilde{H}_2\right)c_{23}\right\}+\widetilde{A}_2\left\{\left(\widetilde{\eta}\widetilde{H}_1+\widetilde{\xi}\widetilde{H}_2\right)c_{23}\right\}\right)\cos((\widetilde{s}-1)\widetilde{\psi})+(-\widetilde{A}_1\left\{\left(\widetilde{\eta}\widetilde{H}_1+\widetilde{\xi}\widetilde{H}_2\right)c_{23}\right\}$$
$$\left.+\widetilde{A}_2\left\{c_{22}+\left(\widetilde{\xi}\widetilde{H}_1-\widetilde{\eta}\widetilde{H}_2\right)c_{23}\right\}\right)\sin((\widetilde{s}-1)\widetilde{\psi})\right]+\left\{\left(\cos(\widetilde{\theta})-\widetilde{\xi}\sin(\widetilde{\theta})\right)^2\right.$$
$$\left.+\widetilde{\eta}^2\sin^2(\widetilde{\theta})\right\}^{(\widetilde{s}-1)/2}\left[\left(\widetilde{A}_3\left\{c_{22}+\left(\widetilde{\xi}\widetilde{H}_1-\widetilde{\eta}\widetilde{H}_2\right)c_{23}\right\}+\widetilde{A}_4\left\{\left(\widetilde{\eta}\widetilde{H}_1+\widetilde{\xi}\widetilde{H}_2\right)c_{23}\right\}\right)\cos\left(\left(\widehat{s}-1\right)\widetilde{\psi}'\right)\right.$$
$$\left.+(-\widetilde{A}_3\left\{\left(\widetilde{\eta}\widetilde{H}_1+\widetilde{\xi}\widetilde{H}_2\right)c_{23}\right\}+\widetilde{A}_4\left\{c_{22}+\left(\widetilde{\xi}\widetilde{H}_1-\widetilde{\eta}\widetilde{H}_2\right)c_{23}\right\}\right)\sin((\widetilde{s}-1)\widetilde{\psi}')\right]\right\rangle+O\left(\widetilde{r}^{\widetilde{s}+1}\right), \tag{45}$$

$$
\begin{aligned}
\widetilde{\sigma}_{\widetilde{y}}\left(\widetilde{r},\widetilde{\theta},\widetilde{z}\right) =& \widetilde{r}^{\widetilde{s}-1}\widetilde{D}_b(\widetilde{z})\left(i\widetilde{k}\right)^{\widetilde{s}}\widetilde{s}\Bigg\langle \Big\{ \left(\cos(\widetilde{\theta}) + \widetilde{\xi}\sin(\widetilde{\theta})\right)^2 + \widetilde{\eta}^2\sin^2(\widetilde{\theta}) \Big\}^{(\widetilde{s}-1)/2}\Big[\Big(\widetilde{A}_1\,\{c_{23} \\
&+\left(\widetilde{\xi}\widetilde{H}_1 - \widetilde{\eta}\widetilde{H}_2\right)\widetilde{c}_{33}\} + \widetilde{A}_2\Big\{\left(\widetilde{\eta}\widetilde{H}_1 + \widetilde{\xi}\widetilde{H}_2\right)c_{33}\Big\}\Big)\cos((\widetilde{s}-1)\widetilde{\psi}) + (-\widetilde{A}_1\Big\{\left(\widetilde{\eta}\widetilde{H}_1 + \widetilde{\xi}\widetilde{H}_2\right)c_{33}\Big\} \\
&+\widetilde{A}_2\{c_{23} + \left(\widetilde{\xi}\widetilde{H}_1 - \widetilde{\eta}\widetilde{H}_2\right)c_{33}\}\Big)\sin((\widetilde{s}-1)\widetilde{\psi})\Big] + \Big\{\left(\cos(\widetilde{\theta}) - \widetilde{\xi}\sin(\widetilde{\theta})\right)^2 \\
&+\widetilde{\eta}^2\sin^2(\widetilde{\theta})\Big\}^{(\widetilde{s}-1)/2}\Big[\Big(\widetilde{A}_3\{c_{23} + \left(\widetilde{\xi}\widetilde{H}_1 - \widetilde{\eta}\widetilde{H}_2\right)c_{33}\} + \widetilde{A}_4\Big\{\left(\widetilde{\eta}\widetilde{H}_1 + \widetilde{\xi}\widetilde{H}_2\right)c_{33}\Big\}\Big)\cos((\widetilde{s}-1)\widetilde{\psi}') \\
&+(-\widetilde{A}_3\Big\{\left(\widetilde{\eta}\widetilde{H}_1 + \widetilde{\xi}\widetilde{H}_2\right)c_{33}\Big\} + \widetilde{A}_4\{c_{23} + \left(\widetilde{\xi}\widetilde{H}_1 - \widetilde{\eta}\widetilde{H}_2\right)c_{33}\}\Big)\sin((\widetilde{s}-1)\widetilde{\psi}')\Big]\Bigg\rangle \\
&+O\left(\widetilde{r}^{\widetilde{s}+1}\right),
\end{aligned}
\tag{46}
$$

$$
\begin{aligned}
\widetilde{\tau}_{\widetilde{x}\widetilde{y}}\left(\widetilde{r},\widetilde{\theta},\widetilde{z}\right) =& \widetilde{r}^{\widetilde{s}-1}\widetilde{D}_b(\widetilde{z})\left(i\widetilde{k}\right)^{\widetilde{s}}\widetilde{s}c_{44}\Bigg\langle \Big\{ \left(\cos(\widetilde{\theta}) + \widetilde{\xi}\sin(\widetilde{\theta})\right)^2 + \widetilde{\eta}^2\sin^2(\widetilde{\theta}) \Big\}^{(\widetilde{s}-1)/2}\Big[\Big\{\widetilde{A}_1\left(\widetilde{\xi} + \widetilde{H}_1\right) \\
&+\widetilde{A}_2\left(\widetilde{\eta} + \widetilde{H}_2\right)\Big\}\cos((\widetilde{s}-1)\widetilde{\psi}) + \Big\{-\widetilde{A}_1\left(\widetilde{\eta} + \widetilde{H}_2\right) + \widetilde{A}_2\left(\widetilde{\xi} + \widetilde{H}_1\right)\Big\}\sin((\widetilde{s}-1)\widetilde{\psi})\Big] \\
&+\Big\{ \left(\cos(\widetilde{\theta}) - \widetilde{\xi}\sin(\widetilde{\theta})\right)^2 + \widetilde{\eta}^2\sin^2(\widetilde{\theta}) \Big\}^{(\widetilde{s}-1)/2}\Big[\Big\{-\widetilde{A}_3\left(\widetilde{\xi} + \widetilde{H}_1\right) - \widetilde{A}_4\left(\widetilde{\eta} + \widetilde{H}_2\right)\Big\}\cos((\widetilde{s}-1)\widetilde{\psi}') \\
&+\Big\{\widetilde{A}_3\left(\widetilde{\eta} + \widetilde{H}_2\right) - \widetilde{A}_4\left(\widetilde{\xi} + \widetilde{H}_1\right)\Big\}\sin((\widetilde{s}-1)\widetilde{\psi}')\Big]\Bigg\rangle + O\left(\widetilde{r}^{\widetilde{s}+1}\right),
\end{aligned}
\tag{47}
$$

$$
\begin{aligned}
\widetilde{\sigma}_{\widetilde{z}}\left(\widetilde{r},\widetilde{\theta},\widetilde{z}\right) =& \widetilde{r}^{\widetilde{s}-1}\widetilde{D}_b(\widetilde{z})\left(i\widetilde{k}\right)^{\widetilde{s}}\widetilde{s}\Bigg\langle \Big\{ \left(\cos(\widetilde{\theta}) + \widetilde{\xi}\sin(\widetilde{\theta})\right)^2 + \widetilde{\eta}^2\sin^2(\widetilde{\theta}) \Big\}^{(\widetilde{s}-1)/2}\Big[\Big(\widetilde{A}_1\,\{c_{12} \\
&+\left(\widetilde{\xi}\widetilde{H}_1 - \widetilde{\eta}\widetilde{H}_2\right)c_{13}\} + \widetilde{A}_2\Big\{\left(\widetilde{\eta}\widetilde{H}_1 + \widetilde{\xi}\widetilde{H}_2\right)c_{13}\Big\}\Big)\cos((\widetilde{s}-1)\widetilde{\psi}) + (-\widetilde{A}_1\Big\{\left(\widetilde{\eta}\widetilde{H}_1 + \widetilde{\xi}\widetilde{H}_2\right)c_{13}\Big\} \\
&+\widetilde{A}_2\{c_{12} + \left(\widetilde{\xi}\widetilde{H}_1 - \widetilde{\eta}\widetilde{H}_2\right)c_{13}\}\Big)\sin((\widetilde{s}-1)\widetilde{\psi})\Big] + \Big\{\left(\cos(\widetilde{\theta}) - \widetilde{\xi}\sin(\widetilde{\theta})\right)^2 \\
&+\widetilde{\eta}^2\sin^2(\widetilde{\theta})\Big\}^{(\widetilde{s}-1)/2}\Big[\Big(\widetilde{A}_3\{c_{12} + \left(\widetilde{\xi}\widetilde{H}_1 - \widetilde{\eta}\widetilde{H}_2\right)c_{13}\} + \widetilde{A}_4\Big\{\left(\widetilde{\eta}\widetilde{H}_1 + \widetilde{\xi}\widetilde{H}_2\right)c_{13}\Big\}\Big)\cos((\widetilde{s}-1)\widetilde{\psi}') \\
&+(-A_3\{(\eta H_1 + \xi H_2)c_{13}\} + A_4\{c_{12} + (\xi H_1 - \eta H_2)c_{13}\})\sin((s-1)\psi')\Big]\Bigg\rangle + O\left(\widetilde{r}^{\widetilde{s}+1}\right),
\end{aligned}
\tag{48}
$$

$$
\widetilde{\tau}_{\widetilde{x}\widetilde{z}}\left(\widetilde{r},\widetilde{\theta},\widetilde{z}\right) = O\left(\widetilde{r}^{\widetilde{s}}\right),
\tag{49}
$$

$$
\widetilde{\tau}_{\widetilde{y}\widetilde{z}}\left(\widetilde{r},\widetilde{\theta},\widetilde{z}\right) = O\left(\widetilde{r}^{\widetilde{s}}\right),
\tag{50}
$$

where

$$
\widetilde{H}_1 = -\frac{\widetilde{\xi}\left(\sqrt{c_{22}c_{33}} + c_{44}\right)}{(c_{23} + c_{44})},
\tag{51}
$$

$$\widetilde{H}_2 = \frac{\widetilde{\eta}(\sqrt{c_{22}c_{33}} - c_{44})}{(c_{23} + c_{44})}. \tag{52}$$

and

$$\widetilde{D}_b(\widetilde{z}) = \widetilde{D}_1 \sin\left(\widetilde{k}\,\widehat{\widetilde{z}}\right) + \widetilde{D}_2 \cos\left(\widetilde{k}\widetilde{z}\right). \tag{53}$$

It may be noted that since $\widetilde{s}$ or Re $\widetilde{s}$ (when $\widetilde{s}$ is complex) is positive, all the higher order terms in Equations (45)–(48) vanish as $\widetilde{r} \to 0$. The non-vanishing components of asymptotic displacement vector and stress tensor in the cylindrical polar coordinate system $(\widetilde{r}, \widetilde{\theta}, \widetilde{z})$ can now be obtained by using the standard vector and the second-rank tensor transformation rule:

$$\left\{ \begin{matrix} \widetilde{u}_{\widetilde{r}} \\ \widetilde{u}_{\widetilde{\theta}} \end{matrix} \right\} = \begin{bmatrix} \cos\widetilde{\theta} & \sin\widetilde{\theta} \\ -\sin\widetilde{\theta} & \cos\widetilde{\theta} \end{bmatrix} \left\{ \begin{matrix} \widetilde{u}_{\widetilde{x}} \\ \widetilde{u}_{\widetilde{y}} \end{matrix} \right\}, \tag{54}$$

$$\left\{ \begin{matrix} \widetilde{\sigma}_{\widetilde{r}} \\ \widetilde{\sigma}_{\widetilde{\theta}} \\ \widetilde{\tau}_{\widetilde{r}\widetilde{\theta}} \end{matrix} \right\} = \begin{bmatrix} \cos^2\widetilde{\theta} & \sin^2\widetilde{\theta} & \sin\left(2\widetilde{\theta}\right) \\ \sin^2\widetilde{\theta} & \cos^2\widetilde{\theta} & -\sin 2\widetilde{\theta} \\ -\frac{1}{2}\sin\left(2\widetilde{\theta}\right) & \frac{1}{2}\sin\left(2\widetilde{\theta}\right) & \cos\left(2\widetilde{\theta}\right) \end{bmatrix} \left\{ \begin{matrix} \widetilde{\sigma}_{\widetilde{x}} \\ \widetilde{\sigma}_{\widetilde{y}} \\ \widetilde{\tau}_{\widetilde{x}\widetilde{y}} \end{matrix} \right\}. \tag{55}$$

The stress component, $\widetilde{\sigma}_{\widetilde{z}}$, is as given in Equation (48).

The substitution of Equation (55), in conjunction with Equations (45)–(48), into the stress-free condition on the crack-side surfaces, given by Equation (7), gives rise to four homogeneous equations, which finally yields:

Either

$$\cos(\widetilde{s} - 1)\pi = 0, \tag{56}$$

or

$$\sin(\widetilde{s} - 1)\pi = 0. \tag{57}$$

Equation (56) yields the lowest nonvanishing eigenvalue, $\widetilde{s} = 1/2$, $0 < \widetilde{s} < 1$, thus meeting the criterion of locally finite energy. Equation (57) gives rise to $\widetilde{s} = 0, 1$, which can take care of rigid body translation and rotation, respectively. Interestingly, $\widetilde{s} = 1$ also accounts for the T-stress; see e.g., Nejati et al. [21].

### 4.1.1. Symmetric (Mode I) Loading (Extension/Bending)

For the case of far-field symmetric (Mode I) loading, the following boundary conditions can be applied:

$$\widetilde{\theta} = 0 : \widetilde{\tau}_{\widetilde{r}\widetilde{\theta}} = \tau_{\widetilde{\theta}\widetilde{z}} = 0, \tag{58}$$

$$\widetilde{\theta} = \pi : \widetilde{\sigma}_{\widetilde{\theta}} = \widetilde{\tau}_{\widetilde{r}\widetilde{\theta}} = \widetilde{\tau}_{\widetilde{\theta}\widetilde{z}} = 0. \tag{59}$$

when $\widetilde{s} = 1/2$, substitution of Equation (55) in conjunction with Equations (45)–(47) into Equations (58) and (59), yields the following:

$$\widetilde{A}_1 = \widetilde{A}_3, \tag{60}$$

$$\widetilde{A}_2 = \widetilde{A}_4. \tag{61}$$

$$\frac{\widetilde{A}_2}{\widetilde{A}_1} = -\frac{\widetilde{\eta}}{\widetilde{\xi}} \frac{\left(\sqrt{c_{22}c_{33}} + c_{23}\right)}{\left(\sqrt{c_{22}c_{33}} - c_{23}\right)}. \tag{62}$$

Finally, on substitution of Equations (60)–(62) into Equations (42), (43) and (45)–(48), the relevant components of the displacement vector and the stress tensor in the immediate neighborhood of a semi-infinite crack front, under Mode I loading, can be written as given below:

$$\widetilde{u}(\widetilde{r}, \widetilde{\theta}, \widetilde{z}) = \frac{\widetilde{K}_I(\widetilde{z})}{\left(c_{22}c_{33} - c_{23}^2\right)} \sqrt{\frac{\widetilde{r}}{2\pi}} \left[ \left\{ \left(\cos\widetilde{\theta} + \widetilde{\xi}\sin\widetilde{\theta}\right)^2 + \widetilde{\eta}\sin^2\widetilde{\theta} \right\}^{1/4} \left\{ \left(\sqrt{c_{22}c_{33}} - c_{23}\right)\cos\left(\widetilde{\psi}/2\right) \right. \right.$$
$$\left. - \left(\sqrt{c_{22}c_{33}} + c_{23}\right)\frac{\widetilde{\eta}}{\widetilde{\xi}}\sin\left(\widetilde{\psi}/2\right) \right\} + \left\{ \left(\cos\widetilde{\theta} - \widetilde{\xi}\sin\widetilde{\theta}\right)^2 + \widetilde{\eta}\sin^2\widetilde{\theta} \right\}^{1/4} \left\{ \left(\sqrt{c_{22}c_{33}} - c_{23}\right)\cos\left(\widetilde{\psi}'/2\right) \right.$$
$$\left. \left. + \left(\sqrt{c_{22}c_{33}} + c_{23}\right)\frac{\widetilde{\eta}}{\widetilde{\xi}}\sin\left(\widetilde{\psi}'/2\right) \right\} \right], \tag{63}$$

$$\widetilde{v}(\widetilde{r}, \widetilde{\theta}, \widetilde{z}) = \frac{\widetilde{K}_I(\widetilde{z})}{\left(c_{22}c_{33} - c_{23}^2\right)} \sqrt{\frac{\widetilde{r}}{2\pi}} \left[ \left\{ \left(\cos\widetilde{\theta} + \widetilde{\xi}\sin\widetilde{\theta}\right)^2 + \widetilde{\eta}\sin^2\widetilde{\theta} \right\}^{1/4} \left\{ -\frac{\left(c_{22}c_{33} - c_{23}^2\right)}{2c_{44}\widetilde{\xi}}\cos\left(\widetilde{\psi}/2\right) \right. \right.$$
$$\left. + 2\sqrt{c_{22}c_{33}}\widetilde{\eta}\sin\left(\widetilde{\psi}/2\right) \right\} + \left\{ \left(\cos\widetilde{\theta} - \widetilde{\xi}\sin\widetilde{\theta}\right)^2 + \widetilde{\eta}\sin^2\widetilde{\theta} \right\}^{1/4} \left\{ \frac{\left(c_{22}c_{33} - c_{23}^2\right)}{2c_{44}\widetilde{\xi}}\cos\left(\widetilde{\psi}'/2\right) \right.$$
$$\left. \left. + 2\sqrt{c_{22}c_{33}}\widetilde{\eta}\sin\left(\widetilde{\psi}'/2\right) \right\} \right], \tag{64}$$

$$\widetilde{\sigma}_{\widetilde{x}}(\widetilde{r}, \widetilde{\theta}, \widetilde{z}) = \frac{\widetilde{K}_I(\widetilde{z})}{2\sqrt{2\pi\widetilde{r}}} \left[ \left\{ \left(\cos\widetilde{\theta} + \widetilde{\xi}\sin\widetilde{\theta}\right)^2 + \widetilde{\eta}\sin^2\widetilde{\theta} \right\}^{1/4} \left\{ \cos\left(\widetilde{\psi}/2\right) - \frac{\widetilde{\eta}}{\widetilde{\xi}}\sin\left(\widetilde{\psi}/2\right) \right\} \right.$$
$$\left. + \left\{ \left(\cos\widetilde{\theta} - \widetilde{\xi}\sin\widetilde{\theta}\right)^2 + \widetilde{\eta}\sin^2\widetilde{\theta} \right\}^{1/4} \left\{ \cos\left(\widetilde{\psi}'/2\right) + \frac{\widetilde{\eta}}{\widetilde{\xi}}\sin\left(\widetilde{\psi}'/2\right) \right\} \right], \tag{65}$$

$$\widetilde{\sigma}_{\widetilde{y}}(\widetilde{r}, \widetilde{\theta}, \widetilde{z}) = \frac{\widetilde{K}_I(\widetilde{z})}{2\sqrt{2\pi\widetilde{r}}} \left[ \left\{ \left(\cos\widetilde{\theta} + \widetilde{\xi}\sin\widetilde{\theta}\right)^2 + \widetilde{\eta}\sin^2\widetilde{\theta} \right\}^{1/4} \left\{ \cos\left(\widetilde{\psi}/2\right) + \frac{\widetilde{\eta}}{\widetilde{\xi}}\sin\left(\widetilde{\psi}/2\right) \right\} \right.$$
$$\left. + \left\{ \left(\cos\widetilde{\theta} - \widetilde{\xi}\sin\widetilde{\theta}\right)^2 + \widetilde{\eta}\sin^2\widetilde{\theta} \right\}^{1/4} \left\{ \cos\left(\widetilde{\psi}'/2\right) - \frac{\widetilde{\eta}}{\widetilde{\xi}}\sin\left(\widetilde{\psi}'/2\right) \right\} \right], \tag{66}$$

$$\widetilde{\tau}_{\widetilde{x}\widetilde{y}}(\widetilde{r}, \widetilde{\theta}, \widetilde{z}) = -\frac{\widetilde{K}_I(\widetilde{z})}{2\sqrt{2\pi\widetilde{r}}\widetilde{\xi}} \left[ \left\{ \left(\cos\widetilde{\theta} + \widetilde{\xi}\sin\widetilde{\theta}\right)^2 + \widetilde{\eta}\sin^2\widetilde{\theta} \right\}^{1/4} \cos\left(\widetilde{\psi}/2\right) \right.$$
$$\left. - \left\{ \left(\cos\widetilde{\theta} - \widetilde{\xi}\sin\widetilde{\theta}\right)^2 + \widetilde{\eta}\sin^2\widetilde{\theta} \right\}^{1/4} \cos\left(\widetilde{\psi}'/2\right) \right], \tag{67}$$

where the mode I stress intensity factor, $\widetilde{K}_I(\widetilde{z})$, is defined as follows:

$$\widetilde{K}_I(\widetilde{z}) = \sqrt{2\pi}\widetilde{D}_b(\widetilde{z})\left(i\widetilde{k}\right)^{1/2}\left(\sqrt{c_{22}c_{33}} + c_{23}\right)\widetilde{A}_1. \tag{68}$$

$$\widetilde{\sigma}_{\widetilde{z}}(\widetilde{r}, \widetilde{\theta}, \widetilde{z}) = \frac{(c_{12}c_{33} - c_{13}c_{23})\widetilde{\sigma}_{\widetilde{x}}(\widetilde{r}, \widetilde{\theta}, \widetilde{z}) + (c_{13}c_{22} - c_{12}c_{23})\widetilde{\sigma}_{\widetilde{y}}(\widetilde{r}, \widetilde{\theta}, \widetilde{z})}{(c_{22}c_{33} - c_{23}^2)}. \tag{69}$$

4.1.2. Skew-Symmetric (Mode II) Loading (Sliding Shear/Twisting)

For the case of far-field skew-symmetric (Mode II) loading, the following boundary conditions can be applied:

$$\widetilde{\theta} = 0 : \ \widetilde{\sigma}_{\widetilde{\theta}} = \widetilde{\tau}_{\widetilde{\theta}\widetilde{z}} = 0, \tag{70}$$

$$\widetilde{\theta} = \pi : \ \widetilde{\sigma}_{\widetilde{\theta}} = \widetilde{\tau}_{\widetilde{r}\widetilde{\theta}} = \widetilde{\tau}_{\widetilde{\theta}\widetilde{z}} = 0. \tag{71}$$

When $\widetilde{s} = 1/2$, substitution of Equation (55) in conjunction with Equations (45)–(47) into Equations (70) and (71) yields the following:

$$\widetilde{A}_1 = -\widetilde{A}_3, \tag{72}$$

$$\widetilde{A}_2 = -\widetilde{A}_4. \tag{73}$$

$$\frac{\widetilde{A}_2}{\widetilde{A}_1} = \frac{4\sqrt{c_{22}c_{33}}c_{44}\widetilde{\xi}\widetilde{\eta}}{(c_{22}c_{33} - c_{23}^2)}. \tag{74}$$

Finally, on substitution of Equations (72)–(74) into Equations (42), (43) and (45)–(48), the relevant components of the displacement vector and the stress tensor in the immediate neighborhood of a semi-infinite crack front, under Mode II loading, can be written as given below:

$$\widetilde{u}(\widetilde{r}, \widetilde{\theta}, \widetilde{z}) = \frac{\widetilde{K}_{II}(\widetilde{z})}{(c_{22}c_{33} - c_{23}^2)} \sqrt{\frac{\widetilde{r}}{2\pi}} \left[ \left\{ \left( \cos\widetilde{\theta} + \widetilde{\xi}\sin\widetilde{\theta} \right)^2 + \widetilde{\eta}\sin^2\widetilde{\theta} \right\}^{1/4} \left\{ \frac{(c_{22}c_{33} - c_{23}^2)}{2c_{44}\widetilde{\xi}} \cos\left(\widetilde{\psi}/2\right) \right. \right.$$

$$+ 2\sqrt{c_{22}c_{33}}\widetilde{\eta}\sin\left(\widetilde{\psi}/2\right) \Big\} + \left\{ \left( \cos\widetilde{\theta} - \widetilde{\xi}\sin\widetilde{\theta} \right)^2 + \widetilde{\eta}\sin^2\widetilde{\theta} \right\}^{1/4} \left\{ -\frac{(c_{22}c_{33} - c_{23}^2)}{2c_{44}\widetilde{\xi}} \cos\left(\widetilde{\psi}'/2\right) \right.$$

$$\left. \left. + 2\sqrt{c_{22}c_{33}}\widetilde{\eta}\sin\left(\widetilde{\psi}'/2\right) \right\} \right], \tag{75}$$

$$\widetilde{v}(\widetilde{r}, \widetilde{\theta}, \widetilde{z}) = -\frac{\widetilde{K}_{II}(\widetilde{z})}{(c_{22}c_{33} - c_{23}^2)} \sqrt{\frac{\widetilde{r}}{2\pi}} \left[ \left\{ \left( \cos\widetilde{\theta} + \widetilde{\xi}\sin\widetilde{\theta} \right)^2 + \widetilde{\eta}\sin^2\widetilde{\theta} \right\}^{1/4} \left\{ (\sqrt{c_{22}c_{33}} - c_{23}) \cos\left(\widetilde{\psi}/2\right) \right. \right.$$

$$+ (\sqrt{c_{22}c_{33}} + c_{23})\frac{\widetilde{\eta}}{\widetilde{\xi}}\sin\left(\widetilde{\psi}/2\right) \Big\} + \left\{ \left( \cos\widetilde{\theta} - \widetilde{\xi}\sin\widetilde{\theta} \right)^2 + \widetilde{\eta}\sin^2\widetilde{\theta} \right\}^{1/4} \left\{ (\sqrt{c_{22}c_{33}} - c_{23}) \cos\left(\widetilde{\psi}'/2\right) \right.$$

$$\left. \left. - (\sqrt{c_{22}c_{33}} + c_{23})\frac{\widetilde{\eta}}{\widetilde{\xi}}\sin\left(\widetilde{\psi}'/2\right) \right\} \right], \tag{76}$$

$$\widetilde{\sigma}_{\widetilde{x}}(\widetilde{r},\widetilde{\theta},\widetilde{z}) = \frac{\widetilde{K}_{II}(\widetilde{z})}{2\sqrt{2\pi\widetilde{r}}}\left[\left\{\left(\cos\widetilde{\theta}+\widetilde{\xi}\sin\widetilde{\theta}\right)^2+\widetilde{\eta}\sin^2\widetilde{\theta}\right\}^{1/4}\left\{\frac{(\widetilde{\eta}^2-\widetilde{\xi}^2)}{\widetilde{\xi}}\cos(\widetilde{\psi}/2)-2\widetilde{\eta}\sin(\widetilde{\psi}/2)\right\}\right.$$
$$\left.-\left\{\left(\cos\widetilde{\theta}-\widetilde{\xi}\sin\widetilde{\theta}\right)^2+\widetilde{\eta}\sin^2\widetilde{\theta}\right\}^{1/4}\left\{\frac{(\widetilde{\eta}^2-\widetilde{\xi}^2)}{\widetilde{\xi}}\cos(\widetilde{\psi}'/2)+2\widetilde{\eta}\sin(\widetilde{\psi}'/2)\right\}\right], \tag{77}$$

$$\widetilde{\sigma}_{\widetilde{y}}(\widetilde{r},\widetilde{\theta},\widetilde{z}) = \frac{\widetilde{K}_{II}(\widetilde{z})}{2\sqrt{2\pi\widetilde{r}}\widetilde{\xi}}\left[\left\{\left(\cos\widetilde{\theta}+\widetilde{\xi}\sin\widetilde{\theta}\right)^2+\widetilde{\eta}\sin^2\widetilde{\theta}\right\}^{1/4}\cos(\widetilde{\psi}/2)\right.$$
$$\left.-\left\{\left(\cos\widetilde{\theta}-\widetilde{\xi}\sin\widetilde{\theta}\right)^2+\widetilde{\eta}\sin^2\widetilde{\theta}\right\}^{1/4}\cos(\widetilde{\psi}'/2)\right], \tag{78}$$

$$\widetilde{\tau}_{\widetilde{x}\widetilde{y}}(\widetilde{r},\widetilde{\theta},\widetilde{z}) = -\frac{\widetilde{K}_{II}(\widetilde{z})}{2\sqrt{2\pi\widetilde{r}}}\left[\left\{\left(\cos\widetilde{\theta}+\widetilde{\xi}\sin\widetilde{\theta}\right)^2+\widetilde{\eta}\sin^2\widetilde{\theta}\right\}^{1/4}\left\{\cos(\widetilde{\psi}/2)+\frac{\widetilde{\eta}}{\widetilde{\xi}}\sin(\widetilde{\psi}/2)\right\}\right.$$
$$\left.+\left\{\left(\cos\widetilde{\theta}-\widetilde{\xi}\sin\widetilde{\theta}\right)^2+\widetilde{\eta}\sin^2\widetilde{\theta}\right\}^{1/4}\left\{\cos(\widetilde{\psi}'/2)-\frac{\widetilde{\eta}}{\widetilde{\xi}}\sin(\widetilde{\psi}'/2)\right\}\right], \tag{79}$$

in which the mode II stress intensity factor, $\widetilde{K}_{II}(\widetilde{z})$, is defined as follows:

$$\widetilde{K}_{II}(\widetilde{z}) = 2\sqrt{2\pi}\widetilde{D}_b(\widetilde{z})\left(i\widetilde{k}\right)^{1/2}\widetilde{\xi}c_{44}A_1. \tag{80}$$

$\widetilde{\sigma}_{\widetilde{z}}(\widetilde{r},\widetilde{\theta},\widetilde{z})$ is given by Equation (69). A critical examination of Equations (66) and (79) reveals the following interesting relationship:

$$\frac{\widetilde{\sigma}_{\widetilde{y}}(\widetilde{r},\widetilde{\theta},\widetilde{z})\Big|_{Mode\ I}}{\widetilde{K}_I(\widetilde{z})} = -\frac{\widetilde{\tau}_{\widetilde{x}\widetilde{y}}(\widetilde{r},\widetilde{\theta},\widetilde{z})_{Mode\ II}}{\widetilde{K}_{II}(\widetilde{z})}, \quad valid\ for\ all\ \theta. \tag{81}$$

Since $\widetilde{K}_I(\widetilde{z})$ and $\widetilde{K}_{II}(\widetilde{z})$ are directly proportional to far-field loadings, $\widetilde{\sigma}_{\widetilde{y}}^{\infty}(\widetilde{z})$ and $\widetilde{\tau}_{\widetilde{x}\widetilde{y}}^{\infty}(\widetilde{z})$, respectively, this interesting result implies that both these loadings produce identical (except the negative sign for $\widetilde{\tau}_{\widetilde{x}\widetilde{y}}$) $\widetilde{\theta}$-dependence of these two near-field driving stresses, $\widetilde{\sigma}_{\widetilde{y}}(\widetilde{r},\widetilde{\theta},\widetilde{z})\Big|_{Mode\ I}/\widetilde{\tau}_{\widetilde{x}\widetilde{y}}(\widetilde{r},\widetilde{\theta},\widetilde{z})_{Mode\ II}$:

$$\frac{\widetilde{K}_I(\widetilde{z})}{\widetilde{K}_{II}(\widetilde{z})} = \frac{\widetilde{\sigma}_{\widetilde{y}}^{\infty}(\widetilde{z})}{\widetilde{\tau}_{\widetilde{x}\widetilde{y}}^{\infty}(\widetilde{z})} = -\frac{\widetilde{\sigma}_{\widetilde{y}}(\widetilde{r},\widetilde{\theta},\widetilde{z})\Big|_{Mode\ I}}{\widetilde{\tau}_{\widetilde{x}\widetilde{y}}(\widetilde{r},\widetilde{\theta},\widetilde{z})_{Mode\ II}}, \quad valid\ for\ all\ |\theta| < \pi. \tag{82}$$

This is unlike what happens in the case of an isotropic material, where no such $\widetilde{\theta}$-invariance (i.e., identical $\widetilde{\theta}$-dependence) exists, as the following relationship would clearly demonstrate [25,55]:

$$\frac{\widetilde{\sigma}_{\widetilde{y}}(\widetilde{r},\widetilde{\theta},\widetilde{z})\Big|_{Mode\ I}}{\widetilde{K}_I(\widetilde{z})\left\{1+\sin\left(\widetilde{\theta}/2\right)\sin\left(3\widetilde{\theta}/2\right)\right\}} = -\frac{\widetilde{\tau}_{\widetilde{x}\widetilde{y}}(\widetilde{r},\widetilde{\theta},\widetilde{z})_{Mode\ II}}{\widetilde{K}_{II}(\widetilde{z})\left\{1-\sin\left(\widetilde{\theta}/2\right)\sin\left(3\widetilde{\theta}/2\right)\right\}}. \tag{83}$$

$$\frac{\widetilde{K}_I(\widetilde{z})}{\widetilde{K}_{II}(\widetilde{z})} = \frac{\widetilde{\sigma}_{\widetilde{y}}^\infty}{\widetilde{\tau}_{\widetilde{xy}}^\infty} = -\frac{\widetilde{\sigma}_{\widetilde{y}}(\widetilde{r},\widetilde{\theta},\widetilde{z})\big|_{Mode\ I}\left\{1 - \sin\left(\widetilde{\theta}/2\right)\sin\left(3\widetilde{\theta}/2\right)\right\}}{\widetilde{\tau}_{\widetilde{xy}}(\widetilde{r},\widetilde{\theta},\widetilde{z})_{Mode\ II}\left\{1 + \sin\left(\widetilde{\theta}/2\right)\sin\left(3\widetilde{\theta}/2\right)\right\}}. \tag{84}$$

Comparison of Equations (81)–(84) shows that while the ratio of mode I to mode II stress intensity factors or its far-field loading counterpart is equal to negative times the ratio of the corresponding driving stresses, $\widetilde{\sigma}_{\widetilde{y}}(\widetilde{r},\widetilde{\theta},\widetilde{z})\big|_{Mode\ I}/\widetilde{\tau}_{\widetilde{xy}}(\widetilde{r},\widetilde{\theta},\widetilde{z})_{Mode\ II}$, for the case of an orthorhombic crystal with complex roots (valid for $\widetilde{A} > 1$ or, equivalently, $\widetilde{\kappa} > \sqrt{c_{33}/c_{22}}$), and this relationship is invariant with respect to coordinate transformation (here specifically with respect to $\widetilde{\theta}$ variation), the same cannot be said about the corresponding relationship for an isotropic material. The coupling between $\cos\left(\widetilde{\psi}/2\right)$ and $\sin\left(\widetilde{\psi}/2\right)$, and similar coupling between $\cos\left(\widetilde{\psi}'/2\right)$ and $\sin\left(\widetilde{\psi}'/2\right)$ in the two driving stresses, $\widetilde{\sigma}_{\widetilde{y}}(\widetilde{r},\widetilde{\theta},\widetilde{z})\big|_{Mode\ I}$ and $\widetilde{\tau}_{\widetilde{xy}}(\widetilde{r},\widetilde{\theta},\widetilde{z})_{Mode\ II}$, which has also been reported in earlier studies [22,56], is instrumental in causing the above remarkable relationship given by Equations (81) and (82), which signifies mode mixity. It is also somewhat counter-intuitive that an invariant (more specifically, with respect to $\widetilde{\theta}$) relationship, such as Equations (81) and (82), resulting from complex roots (valid for $\widetilde{A} > 1$ or, equivalently, $\widetilde{\kappa} > \sqrt{c_{33}/c_{22}}$), guarantees the fact of the cleavage system under consideration being difficult (i.e., not easy), thus further concomitant in ensuring crack deflection or turning from this difficult cleavage system onto a nearby available easy one, violating the self-similarity of crack growth or propagation in the process. In contrast, again counter-intuitively, a non-invariant (more specifically, with respect to $\widetilde{\theta}$) relationship, such as Equations (83) and (84) resulting from the degenerate case ($\widetilde{A} = 1$ or, equivalently, $\widetilde{\kappa} = 1$), guarantees the cleavage system under consideration of being supereasy, thus ensuring the self-similarity (i.e., $\widetilde{\theta}$-invariance) of crack growth or propagation in the process.

The dimensionless parameters, such as the anisotropic ratio, $\widetilde{A}$, or, equivalently, the normalized elastic parameter, $\widetilde{\kappa}$, can serve as the Holy Grail quantity for an a priori determination of the status of a cleavage system to be easy or difficult, very much akin to Reynold's number for fluid flow problems, crossing a critical value of which signifies transition from one regime to another. Here, the anisotropic ratio, $\widetilde{A}$, or, equivalently, normalized elastic parameter, $\widetilde{\kappa}$, for a (001) [100] × [010] cleavage system, crossing the critical value of 1 or $\sqrt{c_{33}/c_{22}}$, respectively, signifies transitioning from self-similar crack growth or propagation to crack deflection or turning from a difficult cleavage system onto a nearby easy one.

As has been suggested by Sedov [83], similarity analysis is an effective tool to solve complex problems in mechanics. This can be employed to establish a sorely needed sufficient condition for the problem at hand and is further elaborated in Section 7.2 below.

*4.2. Case (b): Imaginary Roots*

Equation (24) can also have four imaginary roots given by

$$\widetilde{p}_{1,2} = \pm i\left(\widetilde{\xi}' + \widetilde{\eta}'\right), \tag{85}$$

$$\widetilde{p}_{3,4} = \pm i\left(\widetilde{\xi}' - \widetilde{\eta}'\right), \tag{86}$$

where

$$\widetilde{\xi}' = \frac{1}{\sqrt{2}}\left[\left(\frac{c_{22}}{c_{33}}\right)^{1/2} + \widetilde{\chi}\right]^{1/2}, \tag{87}$$

$$\widetilde{\eta}' = \frac{1}{\sqrt{2}}\left[-\left(\frac{c_{22}}{c_{33}}\right)^{1/2} + \widetilde{\chi}\right]^{1/2}, \tag{88}$$

is valid for

$$\widetilde{A} < 1 \text{ or, equivalently, } \widetilde{\kappa} < \sqrt{c_{33}/c_{22}}, \tag{89}$$

which corresponds to a candidate plane of minimum surface energy.

As has been explained earlier, the general asymptotic form for the displacement and stress fields in the vicinity of the crack front (a branch point in 2D form) can most conveniently be obtained by employing a set of two polar coordinate systems, $(\widetilde{\rho}, \widetilde{\psi})$ and $(\widetilde{\rho}', \widetilde{\psi}')$, derived from the affine transformation discussed earlier, and expressed in terms of the cylindrical polar coordinate system $(\widetilde{r}, \widetilde{\theta}, \widetilde{z})$, as follows:

$$\widetilde{\rho}_1 \cos\left(\widetilde{\psi}_1(\widetilde{\theta})\right) = \widetilde{r}\cos(\widetilde{\theta}), \quad \widetilde{\rho}_1 \sin\left(\widetilde{\psi}_1(\widetilde{\theta})\right) = \widetilde{r}(\widetilde{\xi}' + \widetilde{\eta}')\sin(\widetilde{\theta}), \tag{90}$$

$$\widetilde{\rho}'_1 \cos\left(\widetilde{\psi}'_1(\widetilde{\theta})\right) = \widetilde{r}\cos(\widetilde{\theta}), \quad \widetilde{\rho}'_1 \sin\left(\widetilde{\psi}'_1(\widetilde{\theta})\right) = \widetilde{r}(\widetilde{\xi}' - \widetilde{\eta}')\sin(\widetilde{\theta}), \tag{91}$$

in which

$$\widetilde{\rho}_1 = \widetilde{r}\left\{\cos^2(\widetilde{\theta}) + (\widetilde{\xi}' + \widetilde{\eta}')^2\sin^2(\widetilde{\theta})\right\}^{1/2}, \tag{92}$$

$$\widetilde{\rho}'_1 = \widetilde{r}\left\{\cos^2(\widetilde{\theta}) + (\widetilde{\xi}' - \widetilde{\eta}')^2\sin^2(\widetilde{\theta})\right\}^{1/2}, \tag{93}$$

and

$$\cos\left(\widetilde{\psi}_1(\widetilde{\theta})\right) = \frac{\cos(\widetilde{\theta})}{\left\{\cos^2(\widetilde{\theta}) + (\widetilde{\xi}' + \widetilde{\eta}')^2\sin^2(\widetilde{\theta})\right\}^{1/2}}, \tag{94}$$

$$\sin\left(\widetilde{\psi}_1(\widetilde{\theta})\right) = \frac{(\widetilde{\xi}' + \widetilde{\eta}')\sin(\widetilde{\theta})}{\left\{\cos^2(\widetilde{\theta}) + (\widetilde{\xi}' + \widetilde{\eta}')^2\sin^2(\widetilde{\theta})\right\}^{1/2}}, \tag{95}$$

$$\cos\left(\widetilde{\psi}'_1(\widetilde{\theta})\right) = \frac{\cos(\widetilde{\theta})}{\left\{\cos^2(\widetilde{\theta}) + (\widetilde{\xi}' - \widetilde{\eta}')^2\sin^2(\widetilde{\theta})\right\}^{1/2}}, \tag{96}$$

$$\sin\left(\widetilde{\psi}'_1(\widetilde{\theta})\right) = \frac{(\widetilde{\xi}' - \widetilde{\eta}') \sin(\widetilde{\theta})}{\left\{\cos^2(\widetilde{\theta}) + (\widetilde{\xi}' - \widetilde{\eta}')^2 \sin^2(\widetilde{\theta})\right\}^{1/2}}, \tag{97}$$

The components of the displacement vector and the stress tensor in the immediate neighborhood of the crack front can now be written as shown below:

$$\widetilde{u}(\widetilde{r}, \widetilde{\theta}, \widetilde{z}) = \widetilde{r}^{\widetilde{s}} \widetilde{D}_b(\widetilde{z}) \left(i\widetilde{k}\right)^{\widetilde{s}} \left[\left\{\cos^2(\widetilde{\theta}) + \left(\widetilde{\xi}' + \widetilde{\eta}'\right)^2 \sin^2(\widetilde{\theta})\right\}^{s/2} \left\{\widetilde{A}_1 \, \cos(\widetilde{s}\widetilde{\psi}_1) \, + \, \widetilde{A}_2 \, \sin(\widetilde{s}\widetilde{\psi}_1)\right\}\right.$$
$$\left. + \left\{\cos^2(\widetilde{\theta}) + \left(\widetilde{\xi}' - \widetilde{\eta}'\right)^2 \sin^2(\widetilde{\theta})\right\}^{s/2} \left\{\widetilde{A}_3 \, \cos(\widetilde{s}\widetilde{\psi}'_1) \, + \, \widetilde{A}_4 \, \sin(\widetilde{s}\widetilde{\psi}'_1)\right\}\right] + O\left(\widetilde{r}^{\widetilde{s}+2}\right), \tag{98}$$

$$\widetilde{v}(\widetilde{r}, \widetilde{\theta}, \widetilde{z}) = \widetilde{r}^{\widetilde{s}} \widetilde{D}_b(\widetilde{z}) \left(i\widetilde{k}\right)^{\widetilde{s}} \left[\left\{\cos^2(\widetilde{\theta}) + \left(\widetilde{\xi}' + \widetilde{\eta}'\right)^2 \sin^2(\widetilde{\theta})\right\}^{s/2} \left\{-\widetilde{H}'_1\widetilde{A}_2 \, \cos(\widetilde{s}\widetilde{\psi}_1) \, + \, \widetilde{H}'_1\widetilde{A}_1 \, \sin(\widetilde{s}\widetilde{\psi}_1)\right\}\right.$$
$$\left. + \left\{\cos^2(\widetilde{\theta}) + \left(\widetilde{\xi}' - \widetilde{\eta}'\right)^2 \sin^2(\widetilde{\theta})\right\}^{s/2} \left\{-\widetilde{H}'_2\widetilde{A}_4 \, \cos(\widetilde{s}\widetilde{\psi}'_1) \, + \, \widetilde{H}'_2\widetilde{A}_3 \, \sin(\widetilde{s}\widetilde{\psi}'_1)\right\}\right] + O\left(\widetilde{r}^{\widetilde{s}+2}\right), \tag{99}$$

$$\widetilde{w}(\widetilde{r}, \widetilde{\theta}, \widetilde{z}) = O\left(\widetilde{r}^{\widetilde{s}+1}\right), \tag{100}$$

and

$$\widetilde{\sigma}_{\widetilde{x}}\left(\widetilde{r}, \widetilde{\theta}, \widetilde{z}\right) = \widetilde{r}^{\widetilde{s}-1} \widetilde{D}_b(\widetilde{z}) \left(i\widetilde{k}\right)^{\widetilde{s}} \widetilde{s} \left\langle \left\{\cos^2(\widetilde{\theta}) + \left(\widetilde{\xi}' + \widetilde{\eta}'\right)^2 \sin^2(\widetilde{\theta})\right\}^{(\widetilde{s}-1)/2} \{c_{22}+\right.$$
$$\widetilde{H}'_1\left(\widetilde{\xi}' + \widetilde{\eta}'\right)c_{23}\}\left\{\widetilde{A}_1 \cos((\widetilde{s} - 1)\widetilde{\psi}_1) + \widetilde{A}_2 \sin((\widetilde{s} - 1)\widetilde{\psi}_1)\right\} + \left\{\cos^2(\widetilde{\theta})+\right.$$
$$\left. \left(\widetilde{\xi}' - \widetilde{\eta}'\right)^2 \sin^2(\widetilde{\theta})\right\}^{(\widetilde{s}-1)/2} \{c_{22} + \widetilde{H}'_2\left(\widetilde{\xi}' - \widetilde{\eta}'\right)c_{23}\}\left\{\widetilde{A}_3 \cos((\widetilde{s} - 1)\widetilde{\psi}'_1) + \widetilde{A}_4 \sin((\widetilde{s} - 1)\widetilde{\psi}'_1)\right\}\right\rangle$$
$$+ O\left(r^{s+1}\right), \tag{101}$$

$$\sigma_y(r, \theta, z) = r^{s-1} D_b(z) (ik)^s s \left\langle \left\{\cos^2(\theta) + (\xi' + \eta')^2 \sin^2(\theta)\right\}^{(s-1)/2} \{c_{12}+\right.$$
$$H'_1(\xi' + \eta')c_{22}\}\{A_1 \cos((s - 1)\psi_1) + A_2 \sin((s - 1)\psi_1)\} + \{\cos^2(\theta)+$$
$$\left. (\xi' - \eta')^2 \sin^2(\theta)\right\}^{(s-1)/2} \{c_{12} + H'_2(\xi' - \eta')c_{22}\}\{A_3 \cos((s - 1)\psi'_1) + A_4 \sin((s - 1)\psi'_1)\}\right\rangle$$
$$+ O\left(\widetilde{r}^{\widetilde{s}+1}\right), \tag{102}$$

$$\widetilde{\tau}_{\widetilde{x}\widetilde{y}}\left(\widetilde{r},\widetilde{\theta},\widetilde{z}\right) = \widetilde{r}^{\widetilde{s}-1}\widetilde{D}_b(\widetilde{z})\left(i\widetilde{k}\right)^{\widetilde{s}}\widetilde{s}c_{44}\left\langle\left\{\cos^2(\widetilde{\theta}) + \left(\widetilde{\xi}' + \widetilde{\eta}'\right)^2 \sin^2(\widetilde{\theta})\right\}^{(\widehat{s}-1)/2}\left\{\widetilde{H}_1'\right.\right.$$

$$\left.-\left(\widetilde{\xi}' + \widetilde{\eta}'\right)\right\}\left\{-\widetilde{A}_2 \cos((\widetilde{s}-1)\widetilde{\psi}_1) + \widetilde{A}_1 \sin((\widetilde{s}-1)\widetilde{\psi}_1)\right\} + \left\{\cos^2(\widetilde{\theta})+\right.$$

$$\left.\left(\widetilde{\xi}' - \widetilde{\eta}'\right)^2 \sin^2(\widetilde{\theta})\right\}^{(\widetilde{s}-1)/2}\left\{\widetilde{H}_2' - \left(\widetilde{\xi}' - \widetilde{\eta}'\right)\right\}\left\{-\widetilde{A}_4 \cos((\widetilde{s}-1)\widetilde{\psi}_1) + \widetilde{A}_3 \sin((\widetilde{s}-1)\widetilde{\psi}_1)\right\}\right\rangle$$

$$+ O\left(\widetilde{r}^{\widetilde{s}+1}\right), \tag{103}$$

$$\widetilde{\sigma}_{\widetilde{z}}\left(\widetilde{r},\widetilde{\theta},\widetilde{z}\right) = \widetilde{r}^{\widetilde{s}-1}\widetilde{D}_b(\widetilde{z})\left(i\widetilde{k}\right)^{\widetilde{s}}\widetilde{s}\left\langle\left\{\cos^2(\widetilde{\theta}) + \left(\widetilde{\xi}' + \widetilde{\eta}'\right)^2 \sin^2(\widetilde{\theta})\right\}^{(\widetilde{s}-1)/2}\{c_{12}+\right.$$

$$\widetilde{H}_1'\left(\widetilde{\xi}' + \widetilde{\eta}'\right)c_{13}\right\}\left\{\widetilde{A}_1 \cos((\widetilde{s}-1)\widetilde{\psi}_1) + \widetilde{A}_2 \sin((\widetilde{s}-1)\widetilde{\psi}_1)\right\} + \left\{\cos^2(\widetilde{\theta})+\right.$$

$$\left.\left(\widetilde{\xi}' - \widetilde{\eta}'\right)^2 \sin^2(\widetilde{\theta})\right\}^{(\widetilde{s}-1)/2}\{c_{12} + \widetilde{H}_2'\left(\widetilde{\xi}' - \widetilde{\eta}'\right)c_{13}\}\left\{\widetilde{A}_3 \cos((\widetilde{s}-1)\widetilde{\psi}_1') + \widetilde{A}_4 \sin((\widetilde{s}-1)\widetilde{\psi}_1')\right\}\right\rangle$$

$$+ O\left(\widetilde{r}^{\widetilde{s}+1}\right), \tag{104}$$

$$\widetilde{\tau}_{\widetilde{x}\widetilde{z}}\left(\widetilde{r},\widetilde{\theta},\widetilde{z}\right) = O\left(\widetilde{r}^{\widetilde{s}}\right), \tag{105}$$

$$\widetilde{\tau}_{\widetilde{y}\widetilde{z}}\left(\widetilde{r},\widetilde{\theta},\widetilde{z}\right) = O\left(\widetilde{r}^{\widetilde{s}}\right) \tag{106}$$

where

$$\widetilde{H}_1' = -\frac{\left\{\sqrt{c_{22}c_{33}} - c_{44}\left(\widetilde{\xi}' + \widetilde{\eta}'\right)^2\right\}}{(c_{23} + c_{44})\left(\widetilde{\xi}' + \widetilde{\eta}'\right)}, \tag{107}$$

$$\widetilde{H}_2' = \frac{\left\{\sqrt{c_{22}c_{33}} - c_{44}\left(\widetilde{\xi}' - \widetilde{\eta}'\right)^2\right\}}{(c_{23} + c_{44})\left(\widetilde{\xi}' - \widetilde{\eta}'\right)}. \tag{108}$$

and $\widetilde{D}_b(\widetilde{z})$ is same as given earlier in Equation (53).

As before, since $\widetilde{s}$ or Re $\widetilde{s}$ (when $\widetilde{s}$ is complex) is positive, all the higher order terms in Equations (101)–(104) vanish as $\widetilde{r} \to 0$. The non-vanishing components of the asymptotic displacement vector and the stress tensor in the cylindrical polar coordinate system ($\widetilde{r}$, $\widetilde{\theta}$, $\widetilde{z}$), can now be obtained by using the standard vector and the second-rank tensor transformation rule, given earlier by Equations (54) and (55). The stress component, σz, is as given in Equation (104).

Substitution of Equation (55) in conjunction with Equations (101)–(103) into the stress-free condition on the crack-side surfaces, given by Equation (7), gives rise to four homogeneous equations, which finally yields

Either

$$\cos(\widetilde{s} - 1)\pi = 0, \tag{109}$$

or

$$\sin(\widetilde{s} - 1)\pi = 0. \tag{110}$$

Equation (109) yields the lowest nonvanishing eigenvalue, $\widetilde{s} = 1/2$, $0 < \widetilde{s} < 1$, thus meeting the criterion of locally finite energy. Equation (110), in contrast, gives rise to $\widetilde{s} = 0, 1$, which can take care of rigid body translation and rotation, respectively. Interestingly, $\widetilde{s} = 1$ also accounts for the T-stress.

4.2.1. Symmetric (Mode I) Loading (Extension/Bending)

For $\widetilde{s} = 1/2$,

$$\widetilde{A}_2 = \widetilde{A}_4 = 0; \tag{111}$$

and

$$\frac{\widetilde{A}_3}{\widetilde{A}_1} = -\frac{\left\{\sqrt{c_{22}c_{33}}\left(\widetilde{\xi}' - \widetilde{\eta}'\right) + c_{23}\left(\widetilde{\xi}' + \widetilde{\eta}'\right)\right\}}{\left\{\sqrt{c_{22}c_{33}}\left(\widetilde{\xi}' + \widetilde{\eta}'\right) + c_{23}\left(\widetilde{\xi}' - \widetilde{\eta}'\right)\right\}}. \tag{112}$$

Finally, on substitution of Equations (111) and (112) into Equations (98), (99) and (101)–(104), the relevant components of the displacement vector and the stress tensor in the immediate neighborhood of a semi-infinite crack front, under Mode I far-field loading, can be written as given below:

$$\widetilde{u}(\widetilde{r}, \widetilde{\theta}, \widetilde{z}) = \frac{\widetilde{K}_I(\widetilde{z})}{(c_{22}c_{33} - c_{23}^2)\widetilde{\eta}'}\sqrt{\frac{\widetilde{r}}{2\pi}}\left[\left\{\cos^2\widetilde{\theta} + \left(\widetilde{\xi}' + \widetilde{\eta}'\right)^2\sin^2\widetilde{\theta}\right\}^{1/4}\left\{\sqrt{c_{22}c_{33}}\left(\widetilde{\xi}' + \widetilde{\eta}'\right)\right.\right.$$
$$\left.+ c_{23}\left(\widetilde{\xi}' - \widetilde{\eta}'\right)\right\}\cos\left(\widetilde{\psi}_1/2\right) - \left\{\cos^2\widetilde{\theta} + \left(\widetilde{\xi}' - \widetilde{\eta}'\right)^2\sin^2\widetilde{\theta}\right\}^{1/4}\left\{\sqrt{c_{22}c_{33}}\left(\widetilde{\xi}' - \widetilde{\eta}'\right)\right.$$
$$\left.\left.+ c_{23}\left(\widetilde{\xi}' + \widetilde{\eta}'\right)\right\}\cos\left(\widetilde{\psi}_1'/2\right)\right], \tag{113}$$

$$\widetilde{v}(\widetilde{r}, \widetilde{\theta}, \widetilde{z}) = \frac{\widetilde{K}_I(\widetilde{z})}{(c_{22}c_{33} - c_{23}^2)\widetilde{\eta}'}\sqrt{\frac{\widetilde{r}}{2\pi}}\left[\left\{\cos^2\widetilde{\theta} + \left(\widetilde{\xi}' + \widetilde{\eta}'\right)^2\sin^2\widetilde{\theta}\right\}^{1/4}\left\{c_{23}\right.\right.$$
$$\left.+ \sqrt{c_{22}c_{33}}\left(\widetilde{\xi}' - \widetilde{\eta}'\right)^2\right\}\sin\left(\widetilde{\psi}_1/2\right) - \left\{\cos^2\widetilde{\theta} + \left(\widetilde{\xi}' - \widetilde{\eta}'\right)^2\sin^2\widetilde{\theta}\right\}^{1/4}\left\{c_{23}\right.$$
$$\left.\left.+ \sqrt{c_{22}c_{33}}\left(\widetilde{\xi}' + \widetilde{\eta}'\right)^2\right\}\sin\left(\widetilde{\psi}_1'/2\right)\right], \tag{114}$$

$$\widetilde{\sigma}_{\widetilde{x}}(\widetilde{r}, \widetilde{\theta}, \widetilde{z}) = \frac{\widetilde{K}_I(\widetilde{z})}{2\sqrt{2\pi\widetilde{r}}\,\widetilde{\eta}'}\left[\left\{\cos^2\widetilde{\theta} + \left(\widetilde{\xi}' + \widetilde{\eta}'\right)^2\sin^2\widetilde{\theta}\right\}^{1/4}\left(\widetilde{\xi}' + \widetilde{\eta}'\right)\cos\left(\widetilde{\psi}_1/2\right)\right.$$
$$\left.- \left\{\cos^2\widetilde{\theta} + \left(\widetilde{\xi}' - \widetilde{\eta}'\right)^2\sin^2\widetilde{\theta}\right\}^{1/4}\left(\widetilde{\xi}' - \widetilde{\eta}'\right)\cos\left(\widetilde{\psi}_1'/2\right)\right], \tag{115}$$

$$\widetilde{\sigma_{\widetilde{y}}}(\widetilde{r}, \widetilde{\theta}, \widetilde{z}) = -\frac{\widetilde{K}_I(\widetilde{z})}{2\sqrt{2\pi\widetilde{r}\widetilde{\eta}'}} \left[ \left\{ \cos^2\widetilde{\theta} + \left(\widetilde{\xi}' + \widetilde{\eta}'\right)^2 \sin^2\widetilde{\theta} \right\}^{1/4} \left(\widetilde{\xi}' - \widetilde{\eta}'\right) \cos\left(\widetilde{\psi}_1/2\right) \right.$$
$$\left. - \left\{ \cos^2\widetilde{\theta} + \left(\widetilde{\xi}' - \widetilde{\eta}'\right)^2 \sin^2\widetilde{\theta} \right\}^{1/4} \left(\widetilde{\xi}' + \widetilde{\eta}'\right) \cos\left(\widetilde{\psi}'_1/2\right) \right], \tag{116}$$

$$\widetilde{\tau_{\widetilde{x}\widetilde{y}}}(\widetilde{r}, \widetilde{\theta}, \widetilde{z}) = \frac{\widetilde{K}_I(\widetilde{z})}{2\sqrt{2\pi\widetilde{r}\widetilde{\eta}'}} \left[ \left\{ \cos^2\widetilde{\theta} + \left(\widetilde{\xi}' + \widetilde{\eta}'\right)^2 \sin^2\widetilde{\theta} \right\}^{1/4} \sin\left(\widetilde{\psi}_1/2\right) \right.$$
$$\left. - \left\{ \cos^2\widetilde{\theta} + \left(\widetilde{\xi}' - \widetilde{\eta}'\right)^2 \sin^2\widetilde{\theta} \right\}^{1/4} \sin\left(\widetilde{\psi}'_1/2\right), \tag{117}$$

where the mode I stress intensity factor, $\widetilde{K}_I(\widetilde{z})$, is defined as follows:

$$\widetilde{K}_I(\widetilde{z}) = \sqrt{2\pi}\, \widetilde{D}_b(\widetilde{z}) (i\widetilde{k})^{1/2} \frac{c_{44}\widetilde{\eta}'}{(c_{23} + c_{44})} \left\{ \sqrt{c_{22}c_{33}}\left(\widetilde{\xi}' - \widetilde{\eta}'\right) + c_{23}\left(\widetilde{\xi}' + \widetilde{\eta}'\right) \right\} \widetilde{A}_1. \tag{118}$$

$\widetilde{\sigma_{\widetilde{z}}}(\widetilde{r}, \widetilde{\theta}, \widetilde{z})$ is given by Equation (69).

### 4.2.2. Skew-Symmetric (Mode II) Loading (Sliding Shear/Twisting)

For $\widetilde{s} = 1/2$,

$$\widetilde{A}_1 = \widetilde{A}_3 = 0; \tag{119}$$

and

$$\frac{\widetilde{A}_4}{\widetilde{A}_2} = -\frac{\left(\widetilde{\xi}' - \widetilde{\eta}'\right) \left\{ \sqrt{c_{22}c_{33}}\left(\widetilde{\xi}' - \widetilde{\eta}'\right) + c_{23}\left(\widetilde{\xi}' + \widetilde{\eta}'\right) \right\}}{\left(\widetilde{\xi}' + \widetilde{\eta}'\right) \left\{ \sqrt{c_{22}c_{33}}\left(\widetilde{\xi}' + \widetilde{\eta}'\right) + c_{23}\left(\widetilde{\xi}' - \widetilde{\eta}'\right) \right\}}. \tag{120}$$

Finally, on substitution of Equations (119) and (120) into Equations (98), (99) and (101)–(104), the relevant components of the displacement vector and the stress tensor in the immediate neighborhood of a semi-infinite crack front, under Mode II loading, can be written as given below:

$$\widetilde{u}(\widetilde{r}, \widetilde{\theta}, \widetilde{z}) = -\frac{\widetilde{K}_{II}(\widetilde{z})}{(c_{22}c_{33} - c_{23}^2)\widetilde{\eta}'} \sqrt{\frac{\widetilde{r}}{2\pi}} \left[ \left\{ \cos^2\widetilde{\theta} + \left(\widetilde{\xi}' + \widetilde{\eta}'\right)^2 \sin^2\widetilde{\theta} \right\}^{1/4} \left\{ \sqrt{c_{22}c_{33}}\left(\widetilde{\xi}' + \widetilde{\eta}'\right)^2 \right. \right.$$
$$+ c_{23} \right\} \sin\left(\widetilde{\psi}_1/2\right) - \left\{ \cos^2\widetilde{\theta} + \left(\widetilde{\xi}' - \widetilde{\eta}'\right)^2 \sin^2\widetilde{\theta} \right\}^{1/4} \left\{ \sqrt{c_{22}c_{33}}\left(\widetilde{\xi}' - \widetilde{\eta}'\right)^2 + c_{23} \right\} \sin\left(\widetilde{\psi}'_1/2\right) \right], \tag{121}$$

$$\widetilde{v}(\widetilde{r}, \widetilde{\theta}, \widetilde{z}) = \frac{\widetilde{K}_{II}(\widetilde{z})}{(c_{22}c_{33} - c_{23}^2)\widetilde{\eta}'} \sqrt{\frac{\widetilde{r}}{2\pi}} \left[ \left\{ \cos^2\widetilde{\theta} + \left(\widetilde{\xi}' + \widetilde{\eta}'\right)^2 \sin^2\widetilde{\theta} \right\}^{1/4} \left\{ \sqrt{c_{22}c_{33}}\left(\widetilde{\xi}' - \widetilde{\eta}'\right) \right. \right.$$
$$+ c_{23}\left(\widetilde{\xi}' + \widetilde{\eta}'\right) \right\} \cos\left(\widetilde{\psi}_1/2\right) - \left\{ \cos^2\widetilde{\theta} + \left(\widetilde{\xi}' - \widetilde{\eta}'\right)^2 \sin^2\widetilde{\theta} \right\}^{1/4} \left\{ \left\{ \sqrt{c_{22}c_{33}}\left(\widetilde{\xi}' + \widetilde{\eta}'\right) \right. \right.$$
$$+ c_{23}\left(\widetilde{\xi}' - \widetilde{\eta}'\right) \right\} \cos\left(\widetilde{\psi}'_1/2\right) \right], \tag{122}$$

$$\widetilde{\sigma}_{\widetilde{x}}(\widetilde{r},\widetilde{\theta},\widetilde{z}) = -\frac{\widetilde{K}_{II}(\widetilde{z})}{2\sqrt{2\pi\widetilde{r}\widetilde{\eta}'}}\left[\left\{\cos^2\widetilde{\theta}+\left(\widetilde{\xi}'+\widetilde{\eta}'\right)^2\sin^2\widetilde{\theta}\right\}^{1/4}\left(\widetilde{\xi}'+\widetilde{\eta}'\right)^2\sin\left(\widetilde{\psi}_1/2\right)\right.$$
$$\left.-\left\{\cos^2\widetilde{\theta}+\left(\widetilde{\xi}'-\widetilde{\eta}'\right)^2\sin^2\widetilde{\theta}\right\}^{1/4}\left(\widetilde{\xi}'-\widetilde{\eta}'\right)^2\sin\left(\widetilde{\psi}_1'/2\right)\right],$$
(123)

$$\widetilde{\sigma}_{\widetilde{y}}(\widetilde{r},\widetilde{\theta},\widetilde{z}) = \frac{\widetilde{K}_{II}(\widetilde{z})}{2\sqrt{2\pi\widetilde{r}\widetilde{\eta}'}}\left[\left\{\cos^2\widetilde{\theta}+\left(\widetilde{\xi}'+\widetilde{\eta}'\right)^2\sin^2\widetilde{\theta}\right\}^{1/4}\sin\left(\widetilde{\psi}_1/2\right)\right.$$
$$\left.-\left\{\cos^2\widetilde{\theta}+\left(\widetilde{\xi}'-\widetilde{\eta}'\right)^2\sin^2\widetilde{\theta}\right\}^{1/4}\sin\left(\widetilde{\psi}_1'/2\right)\right],$$
(124)

$$\widetilde{\tau}_{\widetilde{x}\widetilde{y}}(\widetilde{r},\widetilde{\theta},\widetilde{z}) = \frac{\widetilde{K}_{II}(\widetilde{z})}{2\sqrt{2\pi\widetilde{r}\widetilde{\eta}'}}\left[\left\{\cos^2\widetilde{\theta}+\left(\widetilde{\xi}'+\widetilde{\eta}'\right)^2\sin^2\widetilde{\theta}\right\}^{1/4}\left(\widetilde{\xi}'+\widetilde{\eta}'\right)\cos\left(\widetilde{\psi}_1/2\right)\right.$$
$$\left.-\left\{\cos^2\widetilde{\theta}+\left(\widetilde{\xi}'-\widetilde{\eta}'\right)^2\sin^2\widetilde{\theta}\right\}^{1/4}\left(\widetilde{\xi}'-\widetilde{\eta}'\right)\cos\left(\widetilde{\psi}_1'/2\right)\right],$$
(125)

in which the mode II stress intensity factor, $\widetilde{K}_{II}(\widetilde{z})$, is defined as follows:

$$\widetilde{K}_{II}(\widetilde{z}) = \sqrt{2\pi}\,\widetilde{D}_b(\widetilde{z})\,(i\widetilde{k})^{1/2}\,\frac{c_{44}\widetilde{\eta}'}{(c_{23}+c_{44})\left(\widetilde{\xi}'+\widetilde{\eta}'\right)}\left\{\sqrt{c_{22}c_{33}}\left(\widetilde{\xi}'-\widetilde{\eta}'\right)+c_{23}\left(\widetilde{\xi}'+\widetilde{\eta}'\right)\right\}\widetilde{A}_2.$$
(126)

$\widetilde{\sigma}_{\widetilde{z}}(\widetilde{r},\widetilde{\theta},\widetilde{z})$ is given by Equation (69). A critical examination of Equations (116) and (126) reveals the following relationship involving the two near-field crack driving stresses, $\widetilde{\sigma}_{\widetilde{y}}(\widetilde{r},\widetilde{\theta},\widetilde{z})\big|_{Mode\ I}$ and $\widetilde{\tau}_{\widetilde{x}\widetilde{y}}(\widetilde{r},\widetilde{\theta},\widetilde{z})\big|_{Mode\ II}$, and corresponding far-field applied stresses, $\widetilde{\sigma}_{\widetilde{y}}^{\infty}(\widetilde{z})$ and $\widetilde{\tau}_{\widetilde{x}\widetilde{y}}^{\infty}(\widetilde{z})$, proportional to $\widetilde{K}_I(\widetilde{z})$ and $\widetilde{K}_{II}(\widetilde{z})$:

$$\frac{\widetilde{K}_{II}(\widetilde{z})\left(\widetilde{\xi}'+\widetilde{\eta}'\right)\widetilde{\sigma}_{\widetilde{y}}(\widetilde{r},\widetilde{\theta},\widetilde{z})\big|_{Mode\ I}+\widetilde{K}_I(\widetilde{z})\left(\widetilde{\xi}'-\widetilde{\eta}'\right)\widetilde{\tau}_{\widetilde{x}\widetilde{y}}(\widetilde{r},\widetilde{\theta},\widetilde{z})\big|_{Mode\ II}}{\widetilde{K}_{II}(\widetilde{z})\left(\widetilde{\xi}'-\widetilde{\eta}'\right)\widetilde{\sigma}_{\widetilde{y}}(\widetilde{r},\widetilde{\theta},\widetilde{z})\big|_{Mode\ I}+\widetilde{K}_I(\widetilde{z})\left(\widetilde{\xi}'+\widetilde{\eta}'\right)\widetilde{\tau}_{\widetilde{x}\widetilde{y}}(\widetilde{r},\widetilde{\theta},\widetilde{z})\big|_{Mode\ II}}$$
$$= \frac{\left\{\cos^2\widetilde{\theta}+\left(\widetilde{\xi}'-\widetilde{\eta}'\right)^2\sin^2\widetilde{\theta}\right\}^{1/4}}{\left\{\cos^2\widetilde{\theta}+\left(\widetilde{\xi}'+\widetilde{\eta}'\right)^2\sin^2\widetilde{\theta}\right\}^{1/4}}.$$
(127)

Equation (127) shows an absence of $\widetilde{\theta}$-invariance in the relationship involving the two far-field and two near-field stresses mentioned above in the case of an orthorhombic crystal with imaginary roots (valid for $\widetilde{A} < 1$ or, equivalently, $\widetilde{\kappa} < \sqrt{c_{33}/c_{22}}$) as a result of the absence of coupling between $\cos\left(\widetilde{\psi}_1/2\right)$ and $\sin\left(\widetilde{\psi}_1/2\right)$ (and similar coupling between $\cos\left(\widetilde{\psi}_1'/2\right)$ and $\sin\left(\widetilde{\psi}_1'/2\right)$) in the two driving stresses, $\widetilde{\sigma}_{\widetilde{y}}(\widetilde{r},\widetilde{\theta},\widetilde{z})\big|_{Mode\ I}$ and $\widetilde{\tau}_{\widetilde{x}\widetilde{y}}(\widetilde{r},\widetilde{\theta},\widetilde{z})_{Mode\ II}$. The lack of coupling between $\cos\left(\widetilde{\psi}_1/2\right)$ and $\sin\left(\widetilde{\psi}_1/2\right)$ (and similar coupling between $\cos\left(\widetilde{\psi}_1'/2\right)$ and $\sin\left(\widetilde{\psi}_1'/2\right)$) in the two driving stresses, $\widetilde{\sigma}_{\widetilde{y}}(\widetilde{r},\widetilde{\theta},\widetilde{z})\big|_{Mode\ I}$ and $\widetilde{\tau}_{\widetilde{x}\widetilde{y}}(\widetilde{r},\widetilde{\theta},\widetilde{z})_{Mode\ II}$, which has also been reported in earlier studies [22,56], is instrumental in guaranteeing the fact of the cleavage system under consideration being easy, thus ensuring self-similar crack growth or propagation in the process, in a manner similar to what happens in an isotropic solid. Comparison of Equations (83), (84) and (127) also shows a measure of similarity in terms of the $\widetilde{\theta}$-dependence discussed above; see further discussion in Section 7.2 below.

Finally, it may be noted that the above expressions for displacements, given by Equations (63), (64), (75), (76), (113), (114), (121) and (122) and stresses, given by (65)–(67), (77)–(79), (115)–(117) and (123)–(125), reduce to their two-dimensional counterparts (see, e.g., Sih et al. [17]).

### 5. Plate Surface Boundary Conditions and Through-Thickness Distribution of Singular Stress Fields

#### 5.1. Satisfaction of Traction-Free Boundary Conditions

The stress field in the vicinity of the front of a semi-infinite crack under in-plane extension and out-of-plane bending, respectively, can be recovered if in Equations (68), (118) or (80), (126), even and odd functions are selected from $\widetilde{D}_b(\widetilde{z}^*)$:

$$\widetilde{D}_b(\widetilde{z}^*) = \widetilde{D}_{bs}(\widetilde{z}^*) = \widetilde{D}_2 \cos\left(\widetilde{k}\widetilde{z}^*\right), \tag{128}$$

$$\widetilde{D}_b(\widetilde{z}^*) = \widetilde{D}_{ba}(\widetilde{z}^*) = \widetilde{D}_1 \sin\left(\widetilde{k}\widetilde{z}^*\right). \tag{129}$$

where $\widetilde{z}^* = \widetilde{z}/h$. By using the boundary condition on the free plate surfaces, $\widetilde{z}^* = \widetilde{z}/h = \pm 1$, the general form of $\widetilde{D}_{bs}(\widetilde{z}^*)$ can be obtained as

$$\widetilde{D}_{bs}(\widetilde{z}^*) = \sum_{n=0}^{\infty} \widetilde{D}_{2n} \cos\left(\frac{(2n+1)}{2h}\pi\widetilde{z}^*\right). \tag{130}$$

Hence, $\widetilde{K}_I(\widetilde{z}^*) = \widetilde{K}_{Is}(\widetilde{z}^*)$ and $\widetilde{K}_{II}(\widetilde{z}^*) = \widetilde{K}_{IIs}(\widetilde{z}^*)$ would represent symmetric stress intensity factors; see Section 8 and the penultimate figure below.

$\widetilde{D}_{ba}(\widetilde{z}^*)$ that satisfies the traction-free condition on the plate surfaces is, in the absence of discontinuity of the function at $\widetilde{z}^* = \widetilde{z}/h = 0$, generally given by

$$\widetilde{D}_{ba}(\widetilde{z}^*) = \sum_{n=1}^{\infty} \widetilde{D}_{1n} \sin\left(\frac{n\pi}{h}\widetilde{z}^*\right), \tag{131}$$

In the presence of discontinuity of the function at $\widetilde{z}^* = \widetilde{z}/h = 0$, $\widetilde{D}_{ba}(\widetilde{z}^*)$ must be written as follows:

$$\widetilde{D}_{ba}(\widetilde{z}^*) = \left| \sum_{n=0}^{\pm\infty} \widetilde{D}_{2n} \cos\left(\frac{(2n+1)}{2}\pi\widetilde{z}^*\right) \right|. \tag{132}$$

As a consequence, $\widetilde{K}_I(\widetilde{z}^*) = \widetilde{K}_{Ia}(\widetilde{z}^*)$ and $\widetilde{K}_{II}(\widetilde{z}^*) = \widetilde{K}_{IIa}(\widetilde{z}^*)$ would represent anti-symmetric stress intensity factors; see Section 8 below. If the boundary conditions on the traction-free plate surfaces are satisfied, all the displacements and singular stresses vanish on the plate surfaces in the vicinity of the crack front.

#### 5.2. Hyperbolic Cosine Distributed Far-Field Loading

Hyperbolic cosine distributed far-field loading, which is proportional to $\cosh(\widetilde{z}^*)$, $|\widetilde{z}^*| < 1$, is applied. The applied symmetric loading function and the corresponding "stress intensity factors" (valid for $|\widetilde{z}^*| \leq 1$) are proportional to

$$\widetilde{D}_{bs}(\widetilde{z}^*) = \cosh(\widetilde{z}^*) = \frac{\exp(\widetilde{z}^*) + \exp(-\widetilde{z}^*)}{2}. \tag{133}$$

The corresponding Fourier series can be derived as follows:

$$\widetilde{D}_{bs}(\widetilde{z}^*) = \sum_{m=0}^{\infty} \frac{\{e^{-1}(-1)^m + e(-1)^m\}}{\left\{1 + \left(m + \frac{1}{2}\right)^2 \pi^2\right\}} \left(m + \frac{1}{2}\right) \pi \cos\left(\left(\frac{2m+1}{2}\right)\pi\widetilde{z}^*\right). \tag{134}$$

The applied antisymmetric loading function (valid for $|\widetilde{z}^*| < 1$) and the corresponding "stress intensity factors" (valid for $|\widetilde{z}^*| \leq 1$) are proportional to

$$\widetilde{D}_{ba}(\widetilde{z}^*) = |\cosh(-\widetilde{z})| = \frac{1}{2}|\exp(\widetilde{z}^*) + \exp(-\widetilde{z}^*)|. \tag{135}$$

Since $\widetilde{D}_{ba}(\widetilde{z}^*)$ has a discontinuity at $\widetilde{z}^* = 0$, the corresponding Fourier series can be obtained as given below:

$$\widetilde{D}_{bs}(\widetilde{z}^*) = \left| \sum_{m=0}^{\infty} \frac{\{e^{-1}(-1)^m + e(-1)^m\}}{\left\{1 + \left(m + \frac{1}{2}\right)^2 \pi^2\right\}} \left(m + \frac{1}{2}\right) \pi \cos\left(\left(\frac{2m+1}{2}\right)\pi\widetilde{z}^*\right) \right|. \tag{136}$$

## 6. Stress Intensity Factors and Energy Release Rates for a Through-Thickness Center-Crack (Modes I and II)

### 6.1. Through-Thickness Distribution of Stress Intensity Factors (Modes I and II)

The stress intensity factors, $\widetilde{K}_I(\widetilde{z}^*)$ and $\widetilde{K}_{II}(\widetilde{z})$, cannot be determined unless the far-field loading and a characteristic length (e.g., crack geometry) are specified. Sih et al. [17] have shown the applicability of the complex variable approach in conjunction with the eigenfunction expansion approach in the derivation of the two-dimensional stress intensity factors for anisotropic plates. The stress intensity factor for an infinite orthorhombic/tetragonal mono-crystalline plate with a central crack of length, 2a, and subjected to far-field mode I/II loading is available for the two-dimensional case [17], and can easily be extended to the present three-dimensional case as follows:

$$\widetilde{K}_I(\widetilde{z}^*) + \frac{K_{II}(\widetilde{z}^*)}{\widetilde{p}_3} = 2\sqrt{2\pi}\frac{(\widetilde{p}_1 - \widetilde{p}_3)}{\widetilde{p}_3} \underset{\zeta_1 \to \zeta_0}{Lim} (\zeta_1 - \zeta_0)^{1/2} \phi_1{}'(\zeta_1)\widetilde{D}_b(\widetilde{z}^*), \tag{137}$$

where

$$\zeta_1 = a + \widetilde{x} + \widetilde{p}_1\widetilde{y}, \tag{138}$$

$$\zeta_0 = a, \tag{139}$$

$$\phi_1(\zeta_1) = \frac{a^2}{4(\widetilde{p}_1 - \widetilde{p}_3)}\left[\frac{2\widetilde{p}_3\widetilde{\sigma}_{\widetilde{y}}^{\infty} + \widetilde{\tau}_{\widetilde{x}\widetilde{y}}^{\infty}}{\zeta_1 + \sqrt{\zeta_1{}^2 - \zeta_0{}^2}}\right] + C_1\zeta_1, \tag{140}$$

with $C_1$ being a constant. On substitution of Equations (138)–(149) into it, Equation (137) can be expressed in cylindrical polar coordinates as follows:

$$\widetilde{K}_I(\widetilde{z}^*) + \frac{\widetilde{K}_{II}(\widetilde{z}^*)}{\widetilde{p}_3} = \sqrt{2\pi}a^2\left(\widetilde{\sigma}_{\widetilde{y}}^{\infty} + \frac{\widetilde{\tau}_{\widetilde{x}\widetilde{y}}^{\infty}}{\widetilde{p}_3}\right)\underset{r\to 0}{Lim}\frac{\sqrt{\widetilde{r}}}{\sqrt{(a+\widetilde{r})^2-a^2}\left\{a+\widetilde{r}+\sqrt{(a+\widetilde{r})^2-a^2}\right\}}\widetilde{D}_b(\widetilde{z}^*), \tag{141}$$

which finally gives

$$\widetilde{K}_I(\widetilde{z}^*) = \widetilde{\sigma}_{\widetilde{y}}^{\infty}\sqrt{\pi a}\,\widetilde{D}_b(\widetilde{z}^*), \tag{142}$$

$$\widetilde{K}_{II}(\widetilde{z}^*) = \widetilde{\tau}_{\widetilde{x}\widetilde{y}}^{\infty}\sqrt{\pi a}\,\widetilde{D}_b(\widetilde{z}^*), \tag{143}$$

for both complex and imaginary roots. Equations (142) and (143) reduce to their two-dimensional counterparts [53], by taking $\widetilde{D}_b(\widetilde{z}*) = 1$. It may further be noted that the normalization factor, $\widetilde{K}_i(\widetilde{z})/\widetilde{K}_{i,2D}$, i = I, II, is equal to $\widetilde{D}_b(\widetilde{z}^*)$ for a given far-field loading.

### 6.2. Through-Thickness Distribution of Energy Release Rates (Modes I and II)

The through-thickness distributions of the energy release rates due far-field loadings, $\widetilde{\sigma}_{\widetilde{y}}^{\infty}$ and $\widetilde{\tau}_{\widetilde{x}\widetilde{y}}^{\infty}$, for a center-crack of length 2a, weakening an infinite plate of finite thickness, 2 h, can be derived by introducing the thickness-wise partial crack closure method as follows:

$$\widetilde{G}_I(\widetilde{z}^*) = \underset{\Delta a\to 0}{Lim}\frac{1}{(\Delta a)}\left[\int_0^{\Delta a}\widetilde{\sigma}_{\widetilde{y}}(\widetilde{x},0,\widetilde{z}^*)\widetilde{v}(\Delta a - \widetilde{x},\pi,\widetilde{z}^*)d\widetilde{x}d\widetilde{z}^*\right], \tag{144}$$

which, on substitution of $\widetilde{\sigma}_{\widetilde{y}}(\widetilde{x},0,\widetilde{z}^*) = \widetilde{\sigma}_{\widetilde{y}}\big|_{\widetilde{\theta}=0}$ and $\widetilde{v}(\Delta a - \widetilde{x},\pi,\widetilde{z}^*) = \widetilde{v}\big|_{\widetilde{\theta}=\widetilde{\pi}}$, obtained from Equations (66) and (64), respectively, for complex roots, and Equations (116) and (114), respectively, for imaginary roots, yields

$$\widetilde{G}_I(\widetilde{z}^*) = \underset{\Delta a\to 0}{Lim}\frac{1}{(\Delta a)}\begin{cases}\frac{\sqrt{c_{22}c_{33}}}{(c_{22}c_{33}-c_{23}^2)}\widetilde{\eta}\widetilde{K}_I(\Delta a,\widetilde{z}^*)\widetilde{K}_I(0,\widetilde{z}^*)\int_0^{\Delta a}\sqrt{\frac{\Delta a - \widetilde{x}}{\widetilde{x}}}d\widetilde{x}, \textit{for complex roots}\\[3mm]\frac{\sqrt{c_{22}c_{33}}}{(c_{22}c_{33}-c_{23}^2)}\widetilde{\xi}'\widetilde{K}_I(\Delta a, z^*)\widetilde{K}_I(0,\widetilde{z}^*)\int_0^{\Delta a}\sqrt{\frac{\Delta a - \widetilde{x}}{\widetilde{x}}}d\widetilde{x}, \textit{for imaginary roots}\end{cases}$$

$$= \begin{cases}\frac{\sqrt{c_{22}c_{33}}}{(c_{22}c_{33}-c_{23}^2)}\widetilde{\eta}\left(\widetilde{\sigma}_{\widetilde{y}}^{\infty}\right)^2\pi a\left(\widetilde{D}_b(\widetilde{z}^*)\right)^2, \textit{for complex roots}\\[3mm]\frac{\sqrt{c_{22}c_{33}}}{(c_{22}c_{33}-c_{23}^2)}\widetilde{\xi}'\left(\widetilde{\sigma}_{\widetilde{y}}^{\infty}\right)^2\pi a\left(\widetilde{D}_b(\widetilde{z}^*)\right)^2, \textit{for imaginary roots}\end{cases} \tag{145}$$

Interestingly, solutions for both complex and imaginary roots reduce to the following:

$$\widetilde{G}_I(\widetilde{z}^*) = \frac{\left(\widetilde{\sigma}_{\widetilde{y}}^{\infty}\right)^2\pi a\sqrt{c_{22}c_{33}}}{\sqrt{2}(c_{22}c_{33}-c_{23}^2)}\sqrt{\sqrt{(c_{22}/c_{33})}+\widetilde{\chi}}\left[\widetilde{D}_b(\widetilde{z}^*)\right]^2$$

$$= \frac{\left(\widetilde{K}_{I,2D}\right)^2\sqrt{c_{22}c_{33}}}{\sqrt{2}(c_{22}c_{33}-c_{23}^2)}\sqrt{\sqrt{(c_{22}/c_{33})}+\widetilde{\chi}}\left[\widetilde{D}_b(\widetilde{z}^*)\right]^2. \tag{146}$$

Similarly,

$$\widetilde{G}_{II}(\widetilde{z}^*) = \underset{\Delta a \to 0}{Lim} \frac{1}{(\Delta a)} \left[ \int_0^{\Delta a} \widetilde{\tau}_{\widetilde{x}\widetilde{y}}(\widetilde{x}, 0, \widetilde{z}^*) \widetilde{u}(\Delta a - \widetilde{x}, \pi, \widetilde{z}^*) d\widetilde{x} d\widetilde{z}^* \right], \quad (147)$$

which, on substitution of $\widetilde{\tau}_{\widetilde{x}\widetilde{y}}(\widetilde{x}, 0, \widetilde{z}^*) = \widetilde{\tau}_{\widetilde{x}\widetilde{y}}\big|_{\widetilde{\theta}=0}$ and $\widetilde{u}(\Delta a - \widetilde{x}, \pi, \widetilde{z}^*) = \widetilde{u}\big|_{\widetilde{\theta}=\pi}$, obtained from Equations (79) and (75), respectively, for complex roots, and Equations (125) and (121), respectively, for imaginary roots, yields the following:

$$G_{II}(z^*) = \underset{\Delta a \to 0}{Lim} \frac{1}{(\Delta a)} \begin{cases} \frac{c_{33}}{(c_{22}c_{33}-c_{23}{}^2)} \widetilde{\eta} \widetilde{K}_{II}(\Delta a, \widetilde{z}^*) \widetilde{K}_{II}(0, \widetilde{z}^*) \int_0^{\Delta a} \sqrt{\frac{\Delta a - \widetilde{x}}{\widetilde{x}}} d\widetilde{x}, for\ complex\ roots \\[4mm] \frac{c_{33}}{(c_{22}c_{33}-c_{23}{}^2)} \widetilde{\zeta}' c_{33} \widetilde{K}_{II}(\Delta a, \widetilde{z}^*) \widetilde{K}_{II}(0, \widetilde{z}^*) \int_0^{\Delta a} \sqrt{\frac{\Delta a - \widetilde{x}}{\widetilde{x}}} d\widetilde{x}, for\ imaginary\ roots \end{cases} \quad (148)$$

$$= \begin{cases} \frac{c_{33}}{(c_{22}c_{33}-c_{23}{}^2)} \widetilde{\eta} \left( \widetilde{\tau}_{\widetilde{x}\widetilde{y}}^\infty \right)^2 \pi a \left( \widetilde{D}_b(\widetilde{z}^*) \right)^2, for\ complex\ roots \\[4mm] \frac{c_{33}}{(c_{22}c_{33}-c_{23}{}^2)} \widetilde{\zeta}' \left( \widetilde{\tau}_{\widetilde{x}\widetilde{y}}^\infty \right)^2 \pi a \left( \widetilde{D}_b(\widetilde{z}^*) \right)^2, for\ imaginary\ roots \end{cases}$$

As before, solutions for both complex and imaginary roots reduce to the following:

$$\widetilde{G}_{II}(\widetilde{z}^*) = \frac{\left( \tau_{\widetilde{x}\widetilde{y}}^\infty \right)^2 \pi a c_{33}}{\sqrt{2}(c_{22}c_{33}-c_{23}{}^2)} \sqrt{\sqrt{(c_{22}/c_{33})} + \widetilde{\chi}} \left[ \widetilde{D}_b(\widetilde{z}^*) \right]^2.$$

$$= \frac{\left( \widetilde{K}_{I,2D} \right)^2 c_{33}}{\sqrt{2}(c_{22}c_{33}-c_{23}{}^2)} \sqrt{\sqrt{c_{22}/c_{33}} + \widetilde{\chi}} \left[ \widetilde{D}_b(\widetilde{z}^*) \right]^2. \quad (149)$$

Equations (146) and (149) reduce to their two-dimensional (plane strain) counterparts [53], by taking $\widetilde{D}_b(\widetilde{z}^*) = 1$. It may further be noted that the normalization factor is equal to $\left[ \widetilde{D}_b(\widetilde{z}^*) \right]^2$ for a given far-field loading.

For the special case of a tetragonal single crystal, the above energy release rates reduce to

$$\widetilde{G}_I(\widetilde{z}^*) = \frac{\left( \widetilde{K}_{I,2D} \right)^2 \sqrt{c_{11}c_{33}}}{\sqrt{2}(c_{11}c_{33} - c_{13}{}^2)} \sqrt{\sqrt{(c_{11}/c_{33})} + \widetilde{\chi}} \left[ \widetilde{D}_b(\widetilde{z}^*) \right]^2. \quad (150)$$

$$G_{II}(z^*) == \frac{\left( \widetilde{K}_{I,2D} \right)^2 c_{33}}{\sqrt{2}(c_{11}c_{33} - c_{13}{}^2)} \sqrt{\sqrt{c_{11}/c_{33}} + \widetilde{\chi}} \left[ \widetilde{D}_b(\widetilde{z}^*) \right]^2. \quad (151)$$

## 7. Necessary and Sufficient Conditions for Easy or Difficult Cleavage Planes

### 7.1. Crack Deflection Criterion, Based on the Relative Fracture Energy

The important issue of a cleavage plane being deemed easy or difficult can be related to a crack deflection criterion, which is based on the relative fracture energy (or the energy release rate) available for possible "fracture paths" [17]. The deflection or kinking of a crack from the cleavage system 1 to the cleavage system 2 is favored if (however, not iff, i.e., if and only if):

$$G_1/(2\Gamma_1) < 1 < G_2/(2\Gamma_2) \Rightarrow G_2/G_1 > \Gamma_2/\Gamma_1 \qquad (152)$$

in which $G_i$ and $\Gamma_i$, i = 1, 2, are energy release rate and surface energy, respectively, of the ith cleavage system. The above Griffith–Irwin theory-based crack deflection criterion is not accepted as a sufficient condition (albeit being still extremely useful and widely employed, including in the present work) for a cleavage system deemed to be easy or difficult for crack propagation in single crystals, which calls for development of a new conceptual-cum-analytical tool, which would take into account short-range interactions.

Atomistic scale modeling of cracks, however, requires consideration of both the long-range elastic interactions and the short-range chemical reactions. The Griffith theory does not take the latter into account [22]. Secondly and more importantly, fracture criteria derived from equilibrium theories such as the Griffith (thermodynamics-based) energy balance criterion are not equipped to meet the sufficiency condition, because of the prevailing non-equilibrium conditions such as physico-chemical reactions during crack propagation. Hence, such criteria can only be regarded as necessary conditions for fracture [79,83], but not as sufficient [80,84]. The effect of short-range chemical reactions can obviously be encapsulated by atomic scale simulations, such as the investigation of low-speed propagation instabilities in silicon using quantum-mechanical hybrid, multi-scale modelling due to Kermode et al. [85], which, however, entails extensive computational and other resources. Alternatively, and more importantly, such short-range interactions can also be captured by elastic properties-based parameters (with a few exceptions), such as the anisotropic ratio, $\widetilde{A}$, or, equivalently, the normalized elastic parameter, $\widetilde{\kappa}$. This is because elastic properties are controlled by various aspects of the underlying structural chemistry of single crystals, such as the Bravais lattice type, bonding (covalent, ionic, and metallic), bonding (including hybridized) orbitals, electro-negativity of constituent atoms in a compound, polarity, etc. [22]. The general theory behind these characteristics pertaining to the structural chemistry of crystals are available in well-known treatises (see, e.g., [86–88]). More specifically, the elastic properties of superconducting $YBa_2Cu_3O_{7-\delta}$ are strongly influenced by oxygen non-stoichiometry (as well as various structural defects). It is known to crystallize in a defect perovskite structure consisting of layers. When $\delta = 1$, the O(1) sites in the Cu(1) layer are vacant and the structure is tetragonal. For $\delta < 0.65$, Cu-O chains along the *b*-axis of the crystal are formed; see Wikipedia, 2010. http://en.wikipedia.org/wiki/Yttrium_barium_copper_oxide (accessed on 1 February 2023).

Elongation of the *b*-axis changes the structure from tetragonal (insulator) to orthorhombic (superconductor), with lattice parameters of *a* = 3.82 Å, *b* = 3.89 Å, and *c* = 11.68 Å. Optimum superconducting properties occur when $\delta \sim 0.07$ and all of the O(1) sites are occupied with few vacancies; see Wikipedia, 2010. http://en.wikipedia.org/wiki/Yttrium_barium_copper_oxide (accessed on 1 February 2023). The coordination geometry of metal centers in YBCO, such as cubic $\{YO_8\}$, $\{BaO_{10}\}$, square planar $\{CuO_4\}$, and square pyramidal $\{CuO_5\}$, as well as structural features such as puckered Cu plane and Cu ribbons were first reported by Williams et al. [89]. Furthermore, Ledbetter [90] and Lin et al. [73] measured the elastic constants of polycrystalline YBCO using ultrasonic methods and found that while the "Elastic moduli corresponding mainly to shear modes increase monotonically with oxygen concentration", their counterparts due to "Dilation modes increase up to the values of 6.7 of the oxygen index, after which they begin to decrease"; see also Lubenets et al. [91].

In this connection, it must be noted that the invariant relationship (82), derived earlier in Section 4.1, equating the ratio of mode I to mode II stress intensity factors or its far-field loading counterpart (long range order) to negative times the ratio of the corresponding driving stresses, $\widetilde{\sigma}_{\widetilde{y}}(\widetilde{r}, \widetilde{\theta}, \widetilde{z})\big|_{Mode\ I} / \widetilde{\tau}_{\widetilde{x}\widetilde{y}}(\widetilde{r}, \widetilde{\theta}, \widetilde{z})_{Mode\ II}$ (short range interaction) for the case of an orthorhombic crystal with complex roots (valid for $\widetilde{A} > 1$ or, equivalently, $\widetilde{\kappa} > \sqrt{c_{33}/c_{22}}$), guarantees the fact of the cleavage system under consideration being difficult, thus further concomitant in ensuring crack deflection or turning from this difficult cleavage system onto a nearby available easy one, violating the self-similarity of crack growth or propagation in the process. This type of behavior of $\widetilde{A}$ (or, equivalently, $\widetilde{\kappa}$), crossing a threshold or critical value that signifies transition from one regime to another, very much establishes its Reynold's number-like character.

It may also be interesting to observe that experimental determination of surface energy, $G_i$, can sometimes be notoriously challenging due to the presence of micro-to-nano scale defects, such as porosity, dislocation, twin boundaries, misalignment of bonds [22,86] with respect to the loading axis, and the like. In contrast, the above-derived invariant relationship (38) demands only measurement of strains or stresses at a point for a given far-field loading, which are, relatively speaking, much easier in comparison to determination of surface energies.

### 7.2. Similarity/Dissimilarity of Present Solutions with Their Isotropic Counterpart

As has been mentioned earlier in Section 4.1, similarity analysis is an effective tool to solve complex problems in the fracture mechanics of single crystals [22,83]. In what follows, such similarity or lack thereof to the present solutions for singular stress fields at crack fronts weakening orthorhombic single crystals with their isotropic counterpart is investigated. Such analyses can lead to a sufficient condition for determination of a cleavage system being easy or difficult for crack propagation.

### 7.2.1. Isotropic Materials

For an isotropic material, the in-plane displacements (for $n = 0$) can be given as follows [25,55]:

$$U(x, y, z) = (ik)^s a_s(z) \rho^s e^{ip\psi}, \tag{153}$$

$$V(x, y, z) = (ik)^s b_s(z) \rho^s e^{ip\psi}, \tag{154}$$

in which

$$p = \pm(s \pm 1), \tag{155}$$

and

$$\rho = \sqrt{x^2 + y^2}, \tag{156}$$

$$\psi = \tan^{-1}\left(\frac{y}{x}\right), \tag{157}$$

giving rise to $\psi = \pi/2$ for all positive values of y, for x = 0.

### 7.2.2. Present Solution Involving Complex Roots

Going back to Equations (20), (21), (30) and (31), the in-plane displacements can be rewritten in the form (for $n = 0$), by dropping the overhead ~ over certain relevant variables and constants (in the interest of generality):

$$U(x, y, z) = (ik)^s a_s(z)(x + py)^s = (ik)^s a_s(z)\rho^s e^{is\psi}, \tag{158}$$

$$V(x, y, z) = (ik)^s b_s(z)(x + py)^s = (ik)^s b_s(z)\rho^s e^{is\psi}, \tag{159}$$

in which $\rho$ and $\psi$ can be rewritten as follows:

$$\rho = \sqrt{(x \pm \xi y)^2 + \eta^2 y^2}, \tag{160}$$

$$\psi = \tan^{-1}\left(\frac{\pm \eta y}{x \pm \xi y}\right). \tag{161}$$

Therefore, for an orthorhombic (tetragonal and cubic being special cases) crystal with complex roots when x = 0,

$$\psi = \tan^{-1}\left(\frac{\pm \eta}{\pm \xi}\right), \tag{162}$$

for all positive values of y, which completely differs from its isotropic counterpart.

### 7.2.3. Present Solution Involving Imaginary Roots

Going back to Equations (20), (21), (85) and (86), the in-plane displacements can be rewritten in the form (for $n = 0$), by dropping the overhead ~ over certain relevant variables and constants:

$$U(x, y, z) = (ik)^s a_s(z)(x + py)^s = (ik)^s a_s(z)\rho^s e^{is\psi}, \tag{163}$$

$$V(x, y, z) = (ik)^s b_s(z)(x + py)^s = (ik)^s b_s(z)\rho^s e^{is\psi}, \tag{164}$$

in which $\rho$ and $\psi$ can be rewritten as follows:

$$\rho = \sqrt{x^2 + (\pm \xi' \pm \eta')^2 y^2}, \psi = \tan^{-1}\left(\frac{(\pm \xi' \pm \eta')y}{x}\right), \tag{165}$$

again yielding $\psi = \pi/2$ for all positive values of y, when x = 0, thus affirming the similarity of crack propagation in such a cleavage system with its isotropic counterpart.

## 8. Numerical Results and Discussion

Table 1 displays the elastic stiffness constants of orthorhombic (superconductor) and tetragonal (insulator) YBCO single crystals. If otherwise not specified, the elastic stiffness constants are measured at room temperature (Approx. 300 °K). Table 2 shows the elastic stiffness ratio square root, $\sqrt{c_{22}/c_{11}}$, the normalized elastic parameter, $\kappa$, nature of the four roots of characteristic equation (complex or imaginary), and the character of the cleavageplane (easy or not) for a (010)[001] through-thickness crack with [100] being the initial propagation direction (see Figure 3), while Table 3 exhibits their counterparts for a $(0\bar{1}0)[100]$ through-thickness crack with [001] being the initial propagation direction (see Figure 4). Tables 4–7 present similar results for $(\bar{1}00)[001] \times [010]$, shown in Figure 5, $(100)[010] \times [001]$, $(001)[100] \times [010]$, shown in Figure 2, and $(001)[0\bar{1}0] \times [100]$ through-thickness crack systems, respectively.

**Table 2.** Normalized elastic parameter, roots of characteristic equation, and the nature (easy or difficult) of a (010) [001] × [100] through-thickness cleavage system, shown in Figure 3.

| Material | A | $\sqrt{\dfrac{c_{22}}{c_{11}}}$ | $\kappa$ | Roots | (010)[001] × [100] Cleavage System †: Easy or Difficult |
|---|---|---|---|---|---|
| YBCO * | 1.6266 | 1.0771 | 2.624 | Complex | Difficult |
| YBCO ** | 0.9884 | 1.046 | 1.0321 | Imaginary | Easy |
| YBCO *** | 0.8971 | 1.0771 | 0.9406 | Imaginary | Easy |
| YBCO$^\text{T}$ | 1.3077 | 1.0 | 1.5097 | Complex | Difficult |

† Cleavage system for a (010) [001] through-thickness crack, with [100] being its initial length direction. * All values measured by resonant ultrasound spectroscopy (except $c_{12}$) by Lei et al. [1]. ** Estimated by Ledbetter and Lei [79]. *** Same as *, except $c_{12}$ and $c_{66}$ measured by ultrasound by Saint-Paul and Henry [71].

**Table 3.** Normalized elastic parameter, roots of characteristic equation, and the nature (easy or difficult) of the $(0\bar{1}0)$ [100] × [001] through-thickness cleavage system, shown in Figure 4.

| Material | $A^{'}$ | $\sqrt{\dfrac{c^{'}_{22}}{c^{'}_{11}}}$ | $\kappa^{'}$ | Roots | $(0\bar{1}0)$[100] × [001] Cleavage System: Easy or Difficult |
|---|---|---|---|---|---|
| YBCO * | 0.7641 | 1.2003 | 0.8334 | Imaginary | Easy |
| YBCO ** | 1.3481 | 1.3298 | 2.1763 | Complex | Difficult |
| YBCO *** | 0.7641 | 1.2003 | 0.8334 | Imaginary | Easy |
| YBCO$^\text{T}$ | 1.1663 | 1.2382 | 1.586 | Complex | Difficult |

* All values measured by resonant ultrasound spectroscopy (except $c_{12}$) by Lei et al. [1]. ** Estimated by Ledbetter and Lei [79]. *** Same as *, except $c_{12}$ and $c_{66}$ measured by ultrasound by Saint-Paul and Henry [71].

**Table 4.** Normalized elastic parameter, roots of characteristic equation, and nature (easy or difficult) of the $(\bar{1}00)$ [001] $\times$ [010] through-thickness cleavage system, shown in Figure 5.

| Material | $A''$ | $\sqrt{\frac{c''_{22}}{c''_{11}}}$ | $\kappa''$ | Roots | $(\bar{1}00)$ [001] $\times$ [010 Cleavage System: Easy or Difficult |
|----------|-------|------|------------|-------|------|
| YBCO * | 1.6266 | 0.9284 | 2.2619 | Complex | Difficult |
| YBCO ** | 0.9884 | 0.956 | 0.9432 | Imaginary | Easy |
| YBCO *** | 0.8971 | 0.9284 | 0.817 | Imaginary | Easy |
| YBCO$^T$ | 1.3077 | 1.0 | 1.3086 | Complex | Difficult |

* All values measured by resonant ultrasound spectroscopy (except $c_{12}$) by Lei et al. [1]. ** Estimated by Ledbetter and Lei [79]. *** Same as *, except $c_{12}$ and $c_{66}$ measured by ultrasound by Saint-Paul and Henry [71].

**Table 5.** Normalized elastic parameter, roots of characteristic equation, and the nature (easy or difficult) of the (100) [010] $\times$ [001] through-thickness cleavage system.

| Material | $\bar{A}$ | $\sqrt{\frac{\bar{c}_{22}}{\bar{c}_{11}}}$ | $\bar{\kappa}$ | Roots | (100)[010] $\times$ [001] Cleavage System: Easy or Difficult |
|----------|-----------|------|------------|-------|------|
| YBCO * | 0.543 | 1.1145 | 0.5232 | Imaginary | Easy |
| YBCO ** | 1.0877 | 1.2711 | 1.447 | Complex | Difficult |
| YBCO *** | 0.543 | 1.1145 | 0.5232 | Imaginary | Easy |
| YBCO$^T$ | 1.1663 | 1.2382 | 1.5863 | Complex | Difficult |

* All values measured by resonant ultrasound spectroscopy (except $c_{12}$) by Lei et al. [1]. ** Estimated by Ledbetter and Lei [79]. *** Same as *, except $c_{12}$ and $c_{66}$ measured by ultrasound by Saint-Paul and Henry [71].

**Table 6.** Normalized elastic parameter, roots of characteristic equation, and the nature (easy or difficult) of the (001) [100] $\times$ [010] through-thickness cleavage system.

| Material | $\tilde{A}$ | $\sqrt{\frac{\tilde{c}_{22}}{\tilde{c}_{11}}}$ | $\tilde{\kappa}$ | Roots | (001)[100] $\times$ [010] Cleavage System: Easy or Difficult |
|----------|-------------|------|------------------|-------|------|
| YBCO * | 0.7641 | 0.8331 | 0.5784 | Imaginary | Easy |
| YBCO ** | 1.3481 | 0.7521 | 1.2309 | Complex | Difficult |
| YBCO *** | 0.7641 | 0.8331 | 0.5784 | Imaginary | Easy |
| YBCO$^T$ | 1.1663 | 0.8076 | 1.0345 | Complex | Difficult |

* All values measured by resonant ultrasound spectroscopy (except $c_{12}$) by Lei et al. [1]. ** Estimated by Ledbetter and Lei [79]. *** Same as *, except $c_{12}$ and $c_{66}$ measured by ultrasound by Saint-Paul and Henry [71].

**Table 7.** Normalized elastic parameter, roots of characteristic equation, the nature (easy or difficult) of the (001) [0$\bar{1}$0] $\times$ [100] through-thickness cleavage system.

| Material | $\hat{A}$ | $\sqrt{\frac{\hat{c}_{22}}{\hat{c}_{11}}}$ | $\hat{\kappa}$ | Roots | (001)[0$\bar{1}$0] $\times$ [100] Cleavage System: Easy or Difficult |
|----------|-----------|------|----------------|-------|------|
| YBCO * | 0.543 | 0.8973 | 0.4213 | Imaginary | Easy |
| YBCO ** | 1.0877 | 0.7867 | 0.8954 | Complex | Difficult |
| YBCO *** | 0.543 | 0.8973 | 0.4213 | Imaginary | Easy |
| YBCO$^T$ | 1.1663 | 0.8076 | 1.0345 | Complex | Difficult |

* All values measured by resonant ultrasound spectroscopy (except $c_{12}$) by Lei et al. [1]. ** Estimated by Ledbetter and Lei [79]. *** Same as *, except $c_{12}$ and $c_{66}$ measured by ultrasound by Saint-Paul and Henry [71].

Next, the effect of elastic constants, $c_{ij}$ (especially $c_{12}$ and, to a lesser extent, $c_{66}$), on the nature (i.e., easy or difficult) of a cleavage system in YBCO (YBa$_2$Cu$_3$O$_{7-\delta}$) is discussed. Only three complete sets of elastic constants are available in the Literature accessible to the present author, out of which those due to Ledbetter and Lei [79] are just estimates (marked ** in Table 1), while their experimental counterparts due to Reichard et al. [67] are based on the assumption of tetragonal symmetry; see Table 1. This only leaves the experimental measurements (by resonant ultrasound spectroscopy) due to Lei et al. [1], marked * in Table 1. However, their $c_{12}$ value appears to be excessively high. This is because, according to these authors themselves, "no wave speed in the crystal depends only on $c_{12}$, it is no way to estimate it directly." It also is well-known that while $c_{12}$ and $c_{66}$ can be measured independently by static tests [76], these constants are always coupled in vibrations-based measurements [77,78]. Therefore, in Table 1 of the present investigation, both $c_{12}$ and $c_{66}$, measured by ultrasound by Saint-Paul and Henry [71], have been utilized (marked ***) in replacement of their counterparts due to Lei et al. [1] in order to assess the fracture characteristics of YBCO, and to compare them with experiments by Cook et al. [6], Raynes et al. [9] and Goyal et al. [10] among others.

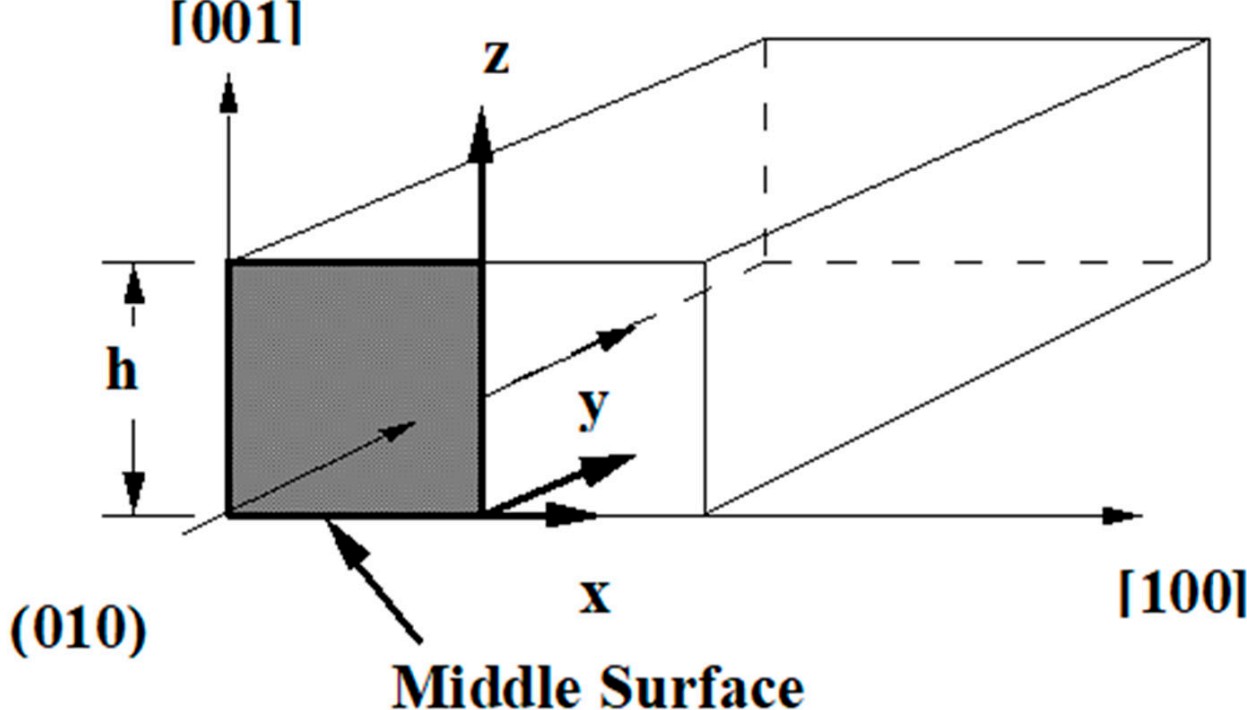

**Figure 3.** Schematic of the top half of an orthorhombic mono-crystalline plate weakened by a (010) [001] through-thickness crack initially propagating in [100] direction.

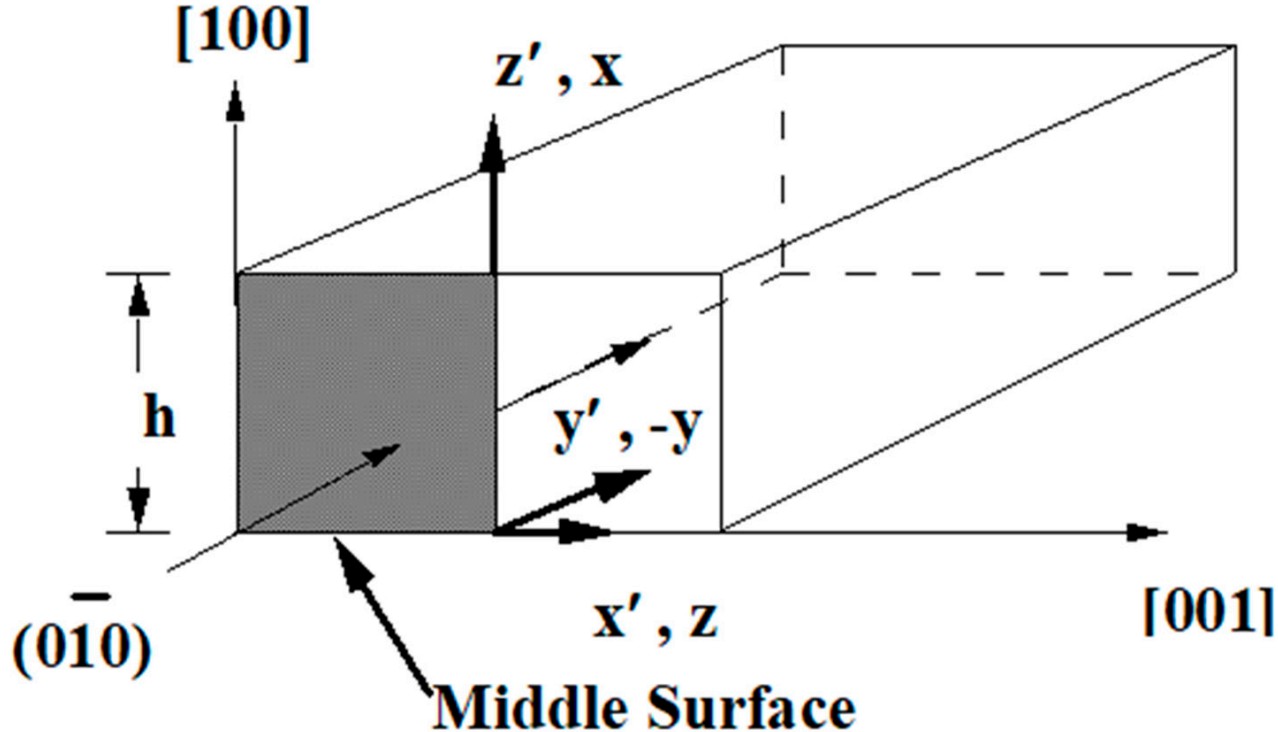

**Figure 4.** Schematic of the top half of an orthorhombic mono–crystalline plate weakened by a (0$\bar{1}$0) [100] through–thickness crack initially propagating in [001] direction.

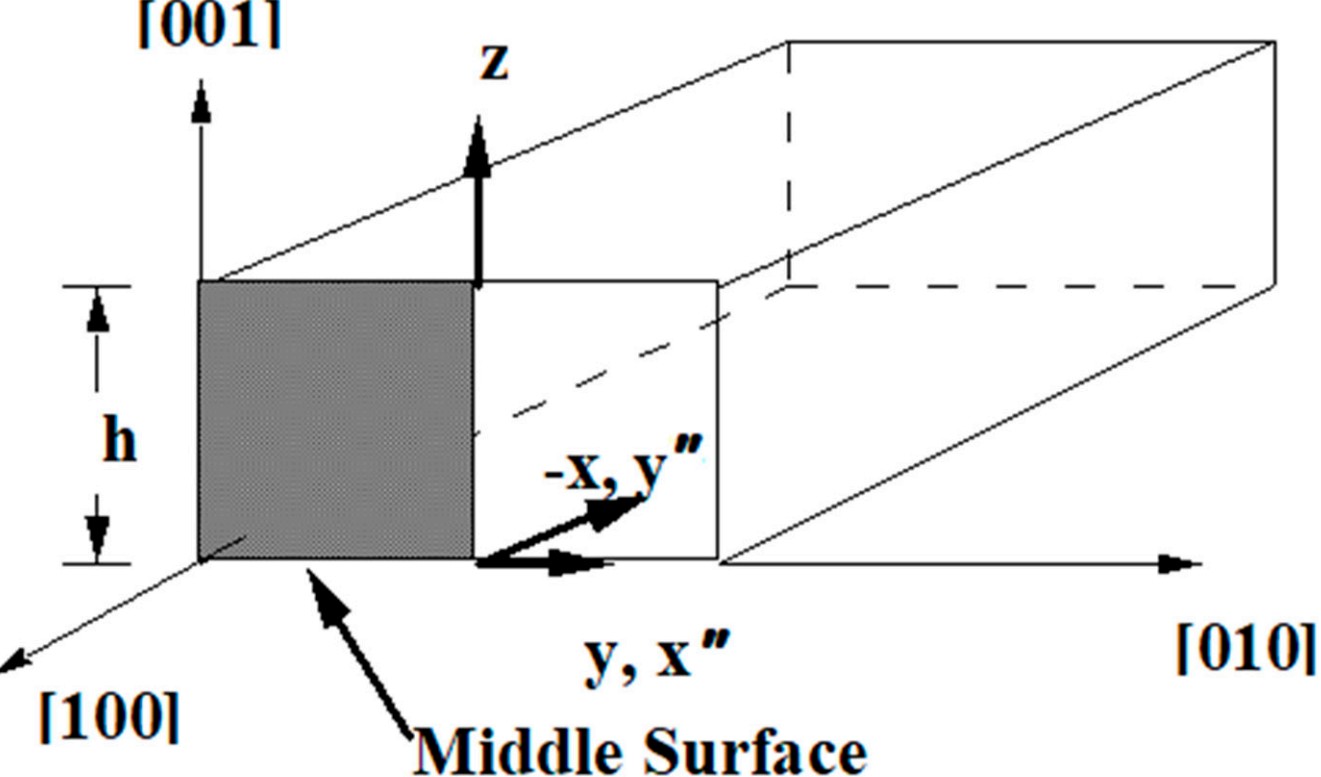

**Figure 5.** Schematic of the top half of an orthorhombic mono-crystalline plate weakened by a ($\bar{1}$00) [001] through-thickness crack initially propagating in [010] direction.

Table 2 shows that the anisotropic ratio, A = 1.6266, or, equivalently, normalized elastic parameter, κ = 2.624, for YBCO * (measurements reported by Lei et al. [1]) is larger than 1 or $\sqrt{c_{22}/c_{11}}$ = 0.9284, respectively, giving rise to complex roots (of the characteristic equation) for a (010)[001] × [100] through-crack, weakening a YBCO mono-crystalline plate. The same is true for a ($\bar{1}$00)[001] × [010] crack shown in Table 4, which shows that $A''$ (respectively, $κ''$) value of 1.6266 (resp. 2.2619) have crossed the critical magnitude of 1 (resp. 0.9284). These results predict that (010)[001] × [100] and ($\bar{1}$00)[001] × [010] are difficult cleavage systems, which are in contradiction with the experimental observations by Cook et al. [6], Raynes et al. [9], and Goyal et al. [10], among others. The reason behind this anomaly lies in the excessive values of $c_{12}$ (and $c_{66}$ to a lesser extent) used in the computation of the anisotropic ratio, A or $A''$, and normalized elastic parameter, κ or $κ''$.

Tables 3 and 5–7 show that the planar anisotropic ratios, $A'$ = 0.7641, $\overline{A}$= 0.543, $\widetilde{A}$= 0.7641, and $\hat{A}$= 0.543 (or, equivalently, normalized elastic parameters, $κ'$ = 0.8334, $\overline{κ}$ = 0.5232, $\widetilde{κ}$ = 0.5784, and $\hat{κ}$= 0.4213 for YBCO*) are less than 1 (or $\sqrt{c'_{22}/c'_{11}}$ = 1.2003, $\sqrt{\overline{c}_{22}/\overline{c}_{11}}$= 1.1145, $\sqrt{\widetilde{c}_{22}/\widetilde{c}_{11}}$= 0.8331, and $\sqrt{\hat{c}_{22}/\hat{c}_{11}}$ = 0.8973), respectively, giving rise to imaginary roots. Thus, the rest of the crack systems are predicted to constitute easy cleavage planes/directions (see Tables 3 and 5–7), the remaining elastic constants measured by Lei et al. [1] are considered to be reasonably accurate.

Tables 3 and 5–7 further show that for YBCO**, $A'$ = 1.3481, $\overline{A}$= 1.0877, $\widetilde{A}$= 1.3481, and $\hat{A}$= 0.543 (or, equivalently, normalized elastic parameters, $κ'$ = 2.1763, $\overline{κ}$= 1.447, $\widetilde{κ}$= 1.2309, and $\hat{κ}$= 1.0877) are greater than 1 (or $\sqrt{c'_{22}/c'_{11}}$= 1.2003, $\sqrt{\overline{c}_{22}/\overline{c}_{11}}$= 1.2711, $\sqrt{\widetilde{c}_{22}/\widetilde{c}_{11}}$= 0.8331, and $\sqrt{\hat{c}_{22}/\hat{c}_{11}}$ = 0.7867), respectively, giving rise to complex roots, and rendering the cleavage systems, (0$\bar{1}$0)[100] × [001], (001)[100] × [010], (100)[010] × [001], and (001)[0$\bar{1}$0] × [100], to be difficult, thus invalidating the values of the corresponding elastic constants estimated by Ledbetter and Lei [92]. Furthermore, Tables 2–7 show that for (tetragonal) YBCO$^T$, A = 1.3077, $A'$ = 1.1663, $A''$ = 1.3077, $\overline{A}$= 1.1663, $\widetilde{A}$= 1.1663, and $\hat{A}$=1.1663 (or, equivalently, normalized elastic parameters, κ = 1.5097, $κ'$ = 1.586, $κ''$ = 1.3086, $\overline{κ}$ = 1.5863, $\widetilde{κ}$ = 1.0345, and $\hat{κ}$ = 1.0345) are greater than 1 (or $\sqrt{c_{22}/c_{11}}$ = 1, $\sqrt{c'_{22}/c'_{11}}$ = 1.0, $\sqrt{c''_{22}/c''_{11}}$ = 1.0, $\sqrt{\overline{c}_{22}/\overline{c}_{11}}$ = 1.2382, $\sqrt{\widetilde{c}_{22}/\widetilde{c}_{11}}$= 0.8076, and $\sqrt{\hat{c}_{22}/\hat{c}_{11}}$ = 0.8076), respectively, giving rise to complex roots, and rendering all the six cleavage systems, namely (010)[001] × [100], (0$\bar{1}$0)[100] × [001], ($\bar{1}$00)[001] × [010], (001)[100] × [010], (100)[010] × [001], and (001)[0$\bar{1}$0] × [100], to be difficult, thus completely invalidating the values of the corresponding experimentally determined elastic constants reported by Reichard et al. [67].

As can be seen from Tables 2–7, only for YBCO***, A = 0.8971, $A'$ = 0.7641, $A''$ = 0.8971, $\overline{A}$ = 0.543, $\widetilde{A}$ = 0.7641, and $\hat{A}$ = 0.543 (or, equivalently, normalized elastic parameters, κ = 0.9406, $κ'$ = 0.8334, $κ''$ = 0.817, $\overline{κ}$ = 0.5232, $\widetilde{κ}$ = 0.5784, and $\hat{κ}$ = 0.4213) are less than 1 (or $\sqrt{c_{22}/c_{11}}$ = 1, $\sqrt{c'_{22}/c'_{11}}$ = 1.2003, $\sqrt{c''_{22}/c''_{11}}$ = 0.9284, $\sqrt{\overline{c}_{22}/\overline{c}_{11}}$ = 1.1145, $\sqrt{\widetilde{c}_{22}/\widetilde{c}_{11}}$ = 0.8331, and $\sqrt{\hat{c}_{22}/\hat{c}_{11}}$ = 0.8076), respectively, giving rise to imaginary roots, and rendering all the cleavage systems, namely (010)[001] × [100], (0$\bar{1}$0)[100] × [001], ($\bar{1}$00)[001] × [010], (001)[100] × [010], (100)[010] × [001], and (001)[0$\bar{1}$0] × [100], to be easy, which is in agreement with the experimentally observed fracture characteristics of YBCO due to Cook et al. [6], Raynes et al. [9], and Goyal et al. [10], among others; see also Granozio and di Uccio [14] for a summary of the available experimental results. They [14] have also presented approximate theoretical results of fully oxidized YBCOs (δ = 0, 1), and concluded that the three lowest surface energies follow the inequality: γ (001) < γ (100) < γ (010). Furthermore, based on experimental results from transmission electron microscopy [92], X-ray photo-emission microscopy [93], low-energy ion scattering spectroscopy [94], and surface polarity [95] analyses performed on fully oxidized YBCO crystals, these authors [14] have shown that the low energy cut is between the Ba = O and Cu = O planes.

The efficacy of the indentation test has been extensively studied in the brittle fracture Literature [96–98]. Lawn [96] and Anstis et al. [97] have presented the following relationship between fracture toughness and the size of a radial crack produced by a Vickers-type sharp indenter:

$K_c = \chi_r P c_0^{-3/2}$, where $\chi_r = \S_V^R (E/H)^{1/2}$, finally giving rise to the following:

$$K_c = \S_V^R (E/H)^{1/2} P c_0^{-3/2}, \tag{166}$$

in which $P$, $c_0$, $E$, and $H$ represent the indentation load, equilibrium half-crack length, Young's modulus, and hardness (of an isotropic material), respectively, and $\S_V^R$ denotes a material-independent constant for the Vickers-produced radial crack. Raynes et al. [9], following the lead of Anstis et al. [97], have determined the fracture toughness of mono-crystalline YBa$_2$Cu$_3$O$_{7-\delta}$, taking into account its anisotropy. Table 8 presents the critical stress intensity factor or fracture toughness ($K_c = K_{Ic}$) and the critical energy release rate or fracture energy ($G_c = G_{Ic}$) of the six easy cleavage systems of mono-crystalline superconducting YBCO. It is worthwhile to note here that there is some misconception about the computation of fracture energy, $G_c$, from the corresponding measured value of $K_c$ of an anisotropic (e.g., orthorhombic) single crystal in the Literature; see, e.g., Granozio and di Uccio [14]. The factor $\sqrt{\sqrt{(c_{22}/c_{33})} + \widetilde{\chi}}$ (see Equation (146) above), and/or its counterparts for other cleavage systems treated in Appendices A–E, are not accounted for in these authors' computations. The energy release rate in an anisotropic (e.g., orthorhombic) single crystal not only varies from one cleavage plane to another, but also varies according to propagation direction.

Figure 6a,b show that a (010)[001] crack initially propagating in [100] direction would turn into a ($\bar{1}$00)[001] crack propagating in [010] direction (also in [0$\bar{1}$0] direction because of symmetry), while a ($\bar{1}$00)[001] crack initially propagating in [010] direction would continue in its original track. This is because the $G_c$ of a (010)[001] × [100] cleavage system, 1.50177 J/m$^2$, is higher than its ($\bar{1}$00)[001] × [010] counterpart, 1.02649 J/m$^2$. In a similar vein, as shown in Figure 7a,b, a (0$\bar{1}$0)[100] crack initially propagating in [001] direction would turn into a (001)[100] crack propagating in [010] direction (also in [0$\bar{1}$0] direction because of symmetry), while a (001)[100] crack initially propagating in [010] direction would continue uninterrupted in its original track. This is because the $G_c$ of a (0$\bar{1}$0)[100] × [001] cleavage system, 1.91945 J/m$^2$, is about 2.6 times its (001)[100] × [010] counterpart, 0.73912 J/m$^2$. Likewise, as shown in Figure 8a,b, a (100)[010] crack initially propagating in [001] direction would turn into a (001) [0$\bar{1}$0] crack propagating in [100] direction (also in [$\bar{1}$00] direction because of symmetry), while a (001)[0$\bar{1}$0] crack initially propagating in [100] direction would continue unhindered in its original track. This is because the $G_c$ of a (100)[010] × [001] cleavage system, 1.02649 J/m$^2$, is more than 1.4 times its (001)[0$\bar{1}$0] × [100] counterpart, 0.71163 J/m$^2$. Finally, in mono-crystalline superconducting YBCO under triaxial-tension far-field loading, a (0$\bar{1}$0)[100] crack initially propagating in [001] direction would eventually turn into a c-plane cleavage fracture, as shown in Figure 9.

**Table 8.** Fracture Toughness ($K_c$) and Fracture Energy ($G_c$) of the Six Easy Cleavage Systems of Mono-crystalline YBCO.

| Cleavage System | (010)[001] × [100] | (0$\bar{1}$0)[100] × [001] | ($\bar{1}$00)[001] × [010] | (100)[010] × [001] | (001)[100] × [010] | (001)[0$\bar{1}$0] × [100] |
|---|---|---|---|---|---|---|
| Fracture Toughness, Kc [9] (MPa $\sqrt{m}$) | 0.59 ± 0.09 | 0.59 ± 0.09 | 0.47 ± 0.12 | 0.47 ± 0.12 | 0.32 ± 0.07 | 0.32 ± 0.07 |
| Fracture Energy, $G_c$ (J/m$^2$) | 1.50177 | 1.91945 | 1.02649 | 1.43075 | 0.73912 | 0.71163 |

Figure 10a,b shows variation of the normalized stress intensity factors, $K^*(z) = K(z)/K_{PlaneStrain}$, through the thickness of an orthorhombic mono-crystalline plate weakened by a through-thickness crack. Variation of the normalized stress intensity factor, $K^*(z)$, through the thickness of the same plate weakened by any of the six through-cracks investigated here is identical. Figure 10a shows the through-thickness variation of $K_S^*(z) = K_S(z)/K_{PlaneStrain}$ for a far-field symmetrically distributed hyperbolic cosine load for mode I (stretching) or mode II (in-plane shear), while its skew-symmetric counterpart $K_A^*(z) = K_A(z)/K_{PlaneStrain}$ for mode I (bending) or mode II (twisting) is displayed in Figure 10b. Of special significance is the discontinuity in the stress intensity factor at z* = 0 in the skew-symmetric loading case, shown in Figure 9. Figure 11 shows the corresponding variation of energy release rate, $G_*$, through the top half of the plate thickness. For through-thickness symmetric far-field loading, the crack is expected to grow through thickness in a stable manner until the stress intensity factor or the energy release rate reaches its critical value at mid-thickness. With further increase in the magnitude of far-field loading, unstable crack growth is expected to progressively spread throughout the plate thickness. For skew-symmetric loading, as reported on earlier occasions [55], the bottom half will experience crack closure. Such types of results describing the three-dimensional distribution of stress intensity factors and energy release rates are generally unavailable in the fracture mechanics Literature.

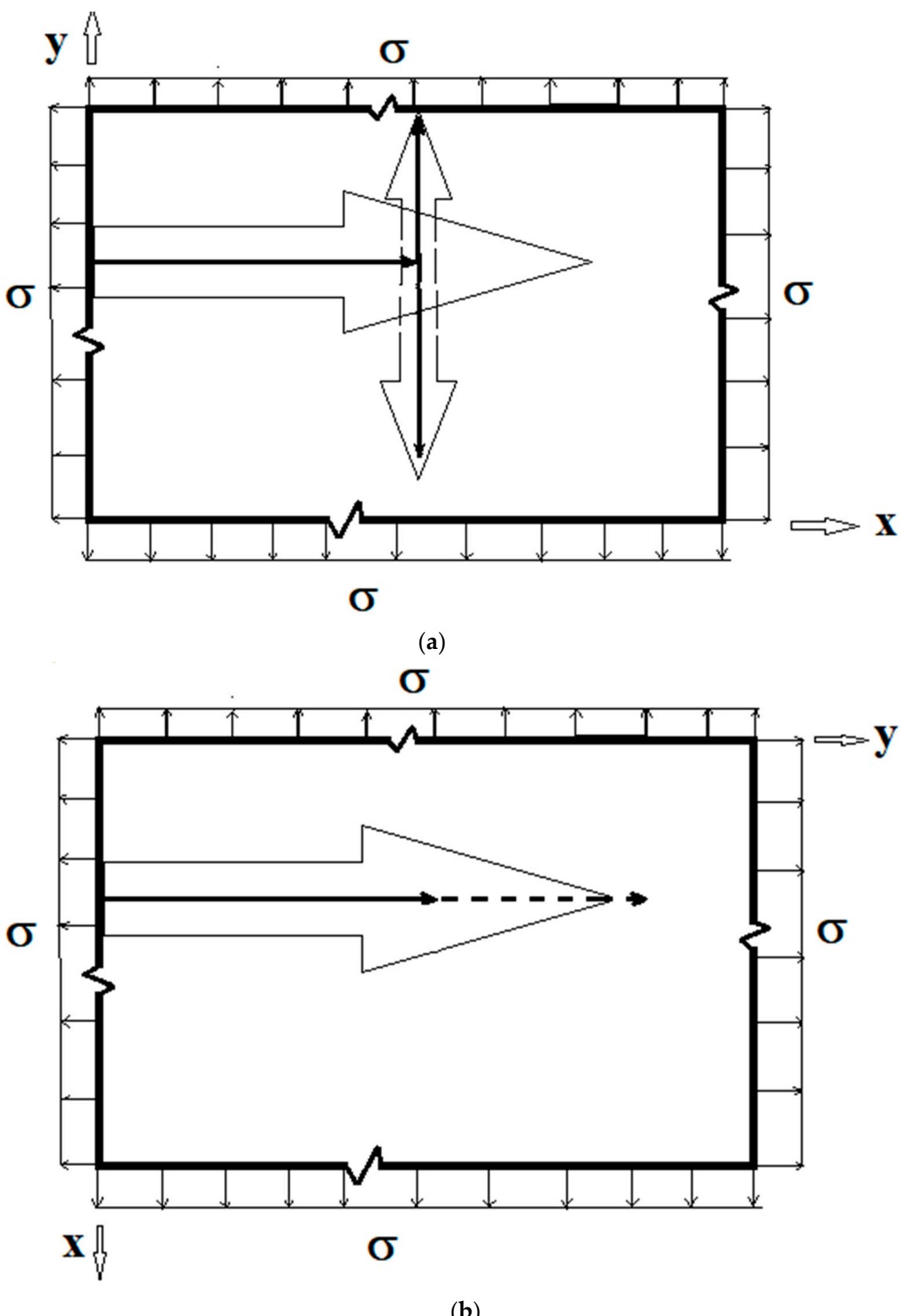

(**a**)

(**b**)

**Figure 6.** (**a**) A (010)[001] crack initially propagating in [100] direction turning into a ($\bar{1}$00)[001] crack propagating in [010] direction, and (**b**) a ($\bar{1}$00)[001] crack initially propagating in [010] direction continuing in its original track.

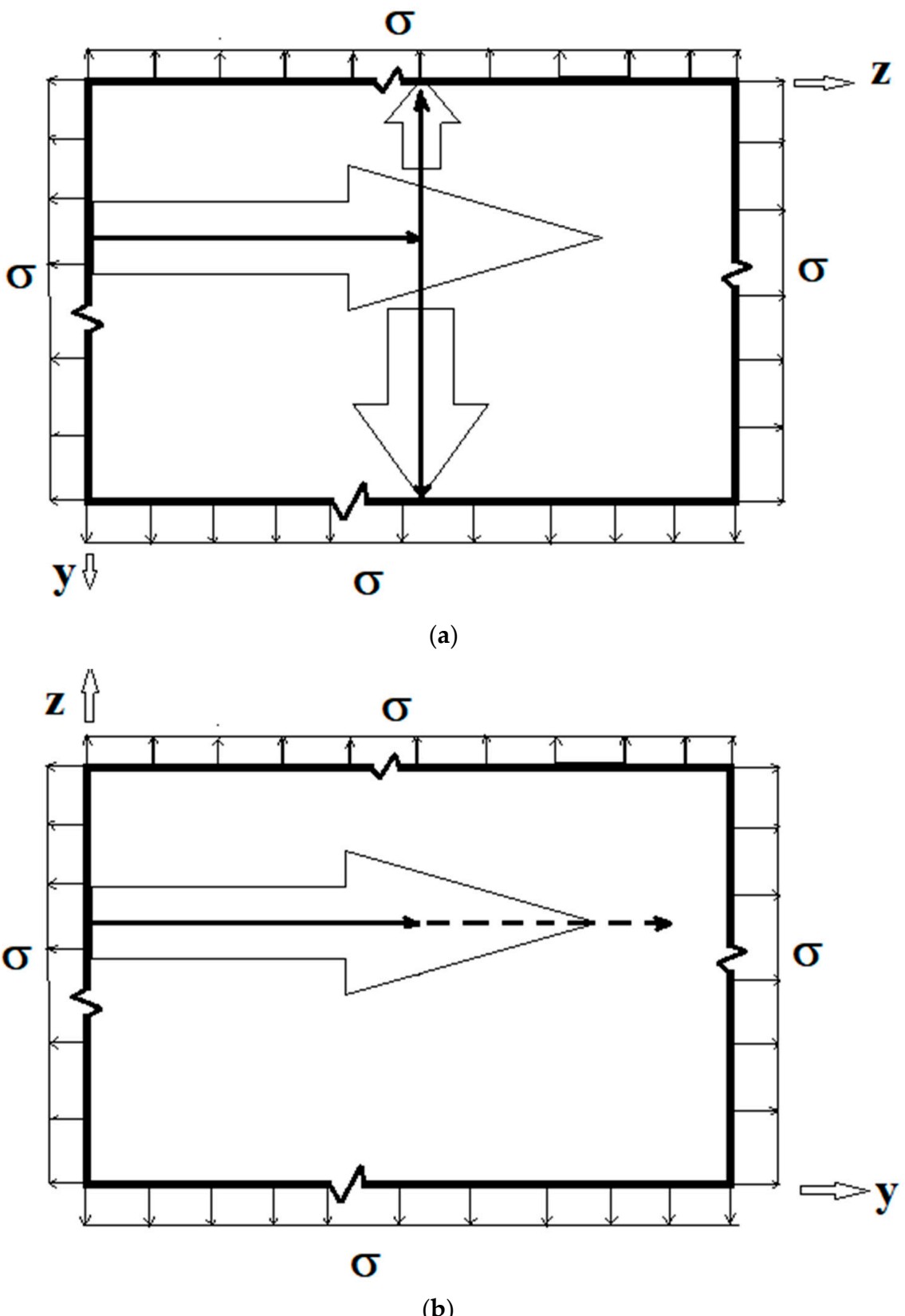

**Figure 7.** (**a**) A (0$\bar{1}$0)[100] crack initially propagating in [001] direction turning into a (001)[100] crack propagating in [010] direction, and (**b**) a (001)[100] crack initially propagating in [010] direction continuing uninterrupted in its original track.

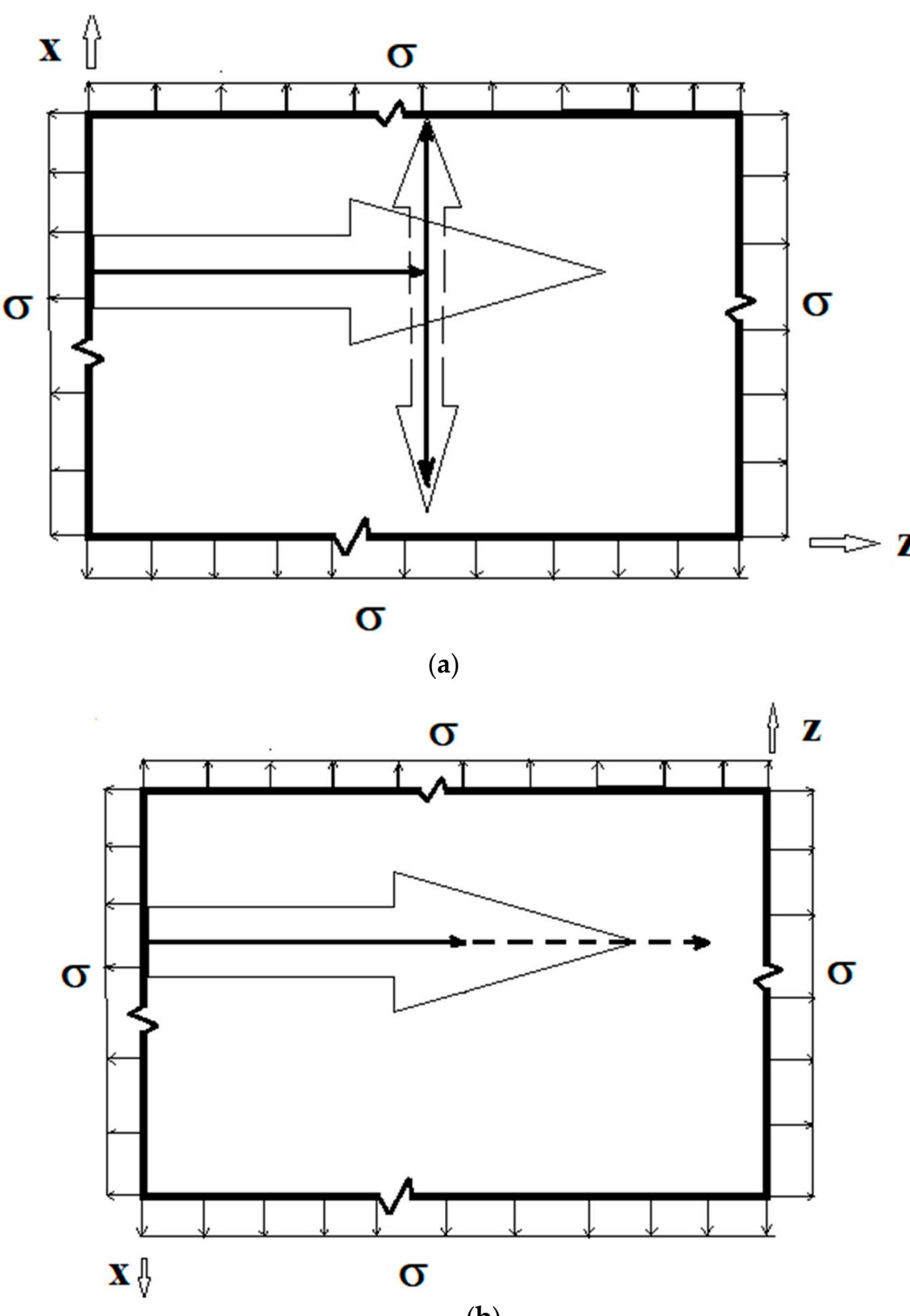

(**a**)

(**b**)

**Figure 8.** (**a**) A (100)[010] crack initially propagating in [001] direction turning into a (001)[ $0\bar{1}0$ ] crack propagating in [100] direction, and (**b**) a (001)[010] crack initially propagating in [100] direction continuing unhindered in its original track.

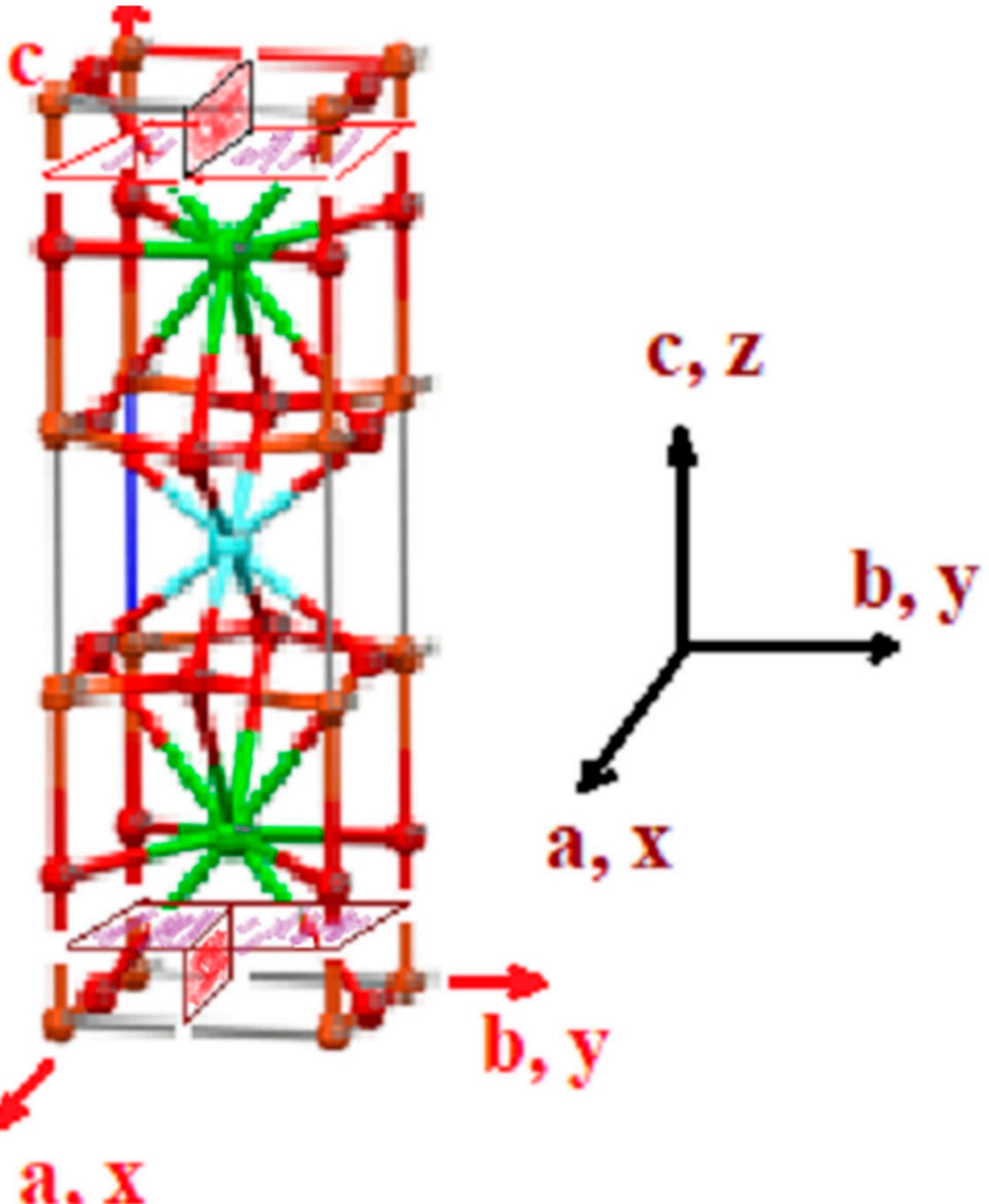

**Figure 9.** A (0$\bar{1}$0)[100] crack initially propagating in [001] direction turning into a c-plane cleavage fracture.

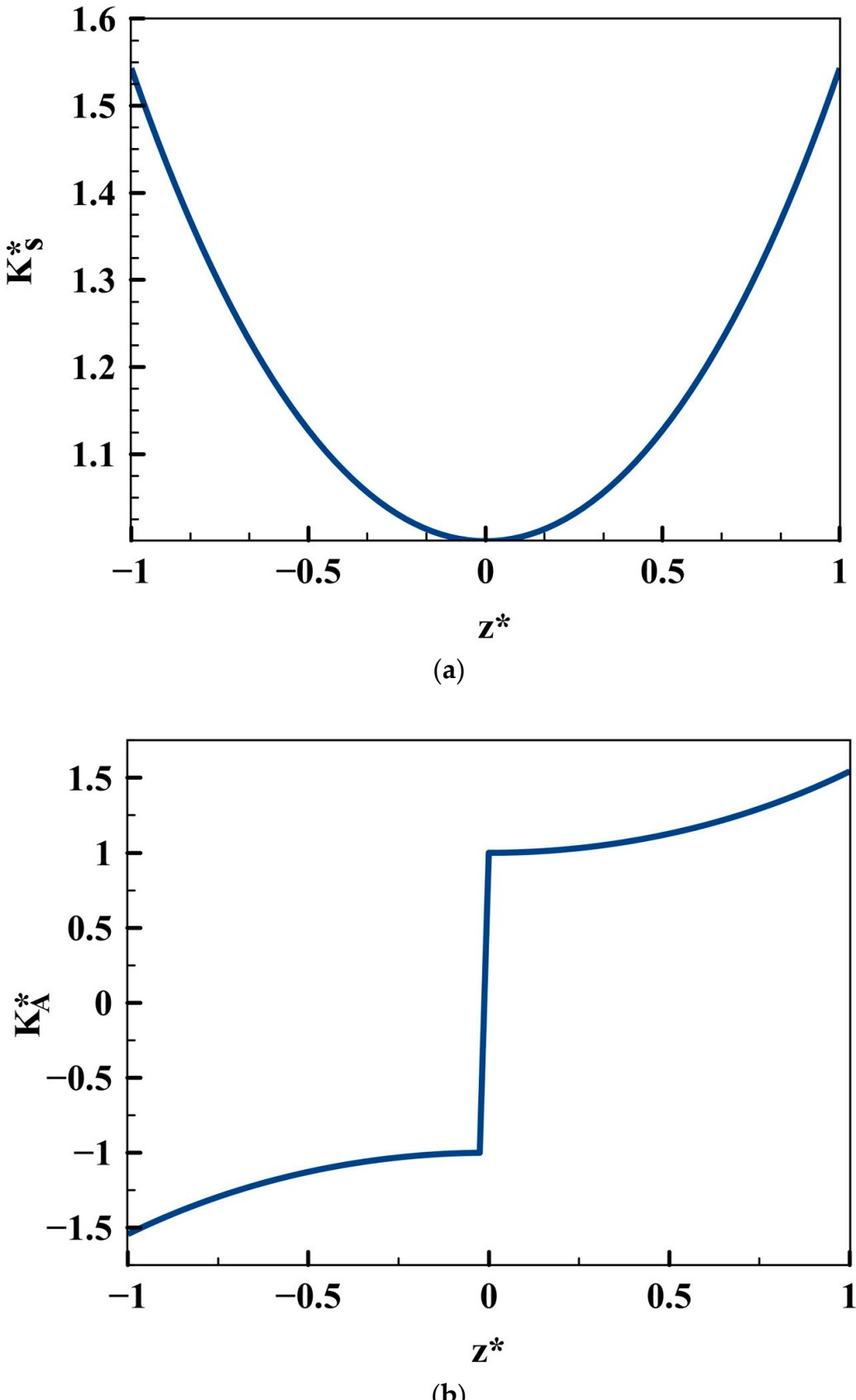

**Figure 10.** The variation of (mode I or II) stress intensity factors through thickness due to far-field cosine hyperbolic load: (**a**) symmetric, (**b**) skew-symmetric.

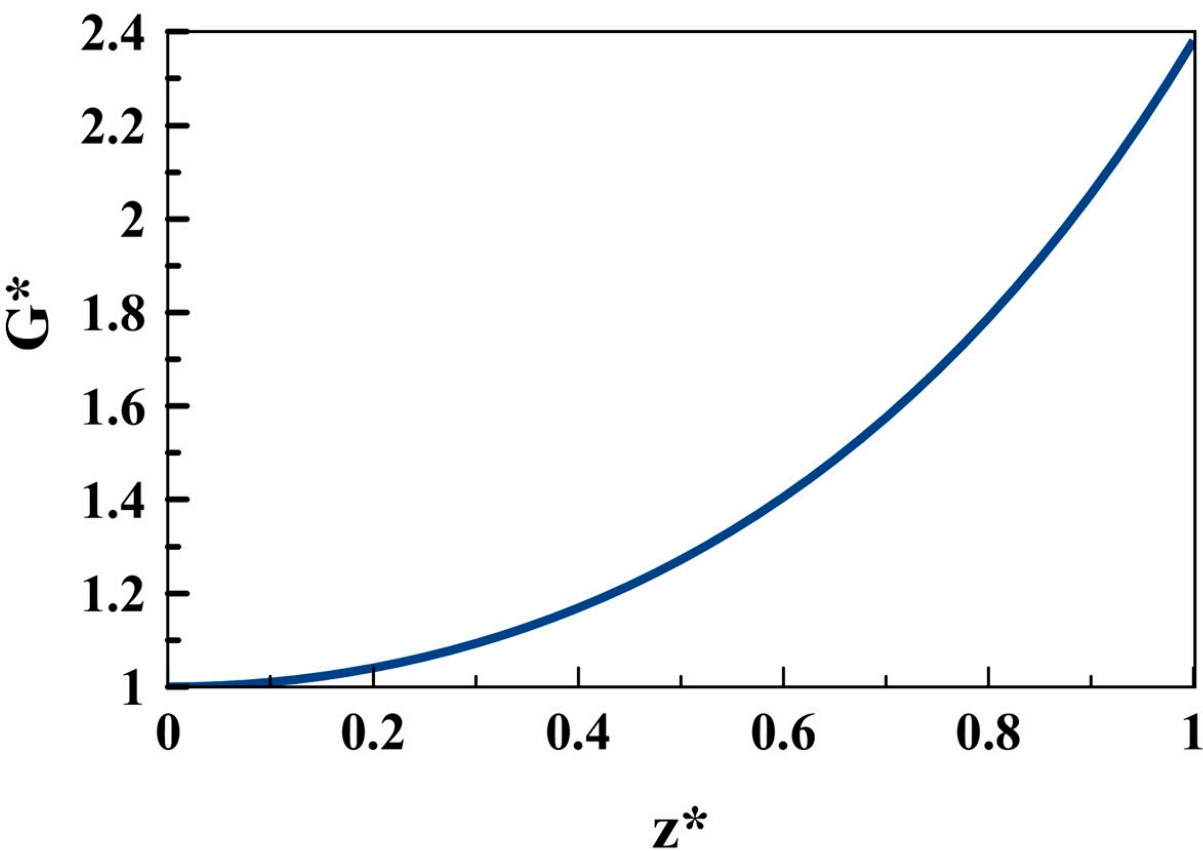

**Figure 11.** The variation of (mode I, II) energy release rate through thickness due to far-field cosine hyperbolic load.

### 9. Summary and Conclusions

A modified eigenfunction expansion technique, based partly on the separation of the z-variable and, in part, on the Eshelby [60]–Stroh [15] type affine transformation, is employed to derive three-dimensional asymptotic displacement and stress fields in the vicinity of the front of a semi-infinite through-thickness crack weakening an infinite orthorhombic single crystal plate. Crack-face boundary conditions and those that are prescribed on the top and bottom (free) surfaces of the orthorhombic plate are exactly satisfied. Explicit expressions for the singular stresses in the vicinity of the front of the through-thickness crack, subjected to far-field in-plane mode I and II loadings, are presented.

The present investigation considers six through-crack systems—(010)[001] with the [100] length direction, (0$\bar{1}$0)[100] with the [001] length direction, ($\bar{1}$00)[001] with the [010] length direction, (100)[010] with the [001] length direction, (001)[0$\bar{1}$0] with the [100] length direction, and (001)[100] with the [010] length direction—weakening orthorhombic YBCO single crystal plates. More importantly, the present approach predicts whether a crack would propagate in its original plane/direction or deflect to a different one. The present study is unique in the sense that such a fracture mechanic criterion is employed for accurate determination of the full set of elastic constants of mono-crystalline YBCO.

The following interesting conclusions can be drawn from the present investigation:

(i) Atomistic scale modeling of cracks requires consideration of both the long-range elastic interactions and the short-range chemical reactions. The Griffith thermodynamic-based theory does not take the latter into account, and hence must be regarded as a necessary condition (albeit still extremely useful and widely employed) but not as sufficient.

(ii) The effect of short-range chemical reactions can be adequately captured by the elastic properties-based parameters, such as the anisotropic ratio, A, or, equivalently, the normalized elastic parameter, $\kappa$. This is because the elastic properties are controlled by various aspects of the underlying structural chemistry of single crystals, such as the Bravais lattice type, bonding (covalent, ionic, and metallic), bonding (including hybridized) orbitals, electro-negativity of constituent atoms in a compound, polarity, etc.

(iii) More specifically, the elastic properties of superconducting $YBa_2Cu_3O_{7-\delta}$ are strongly influenced by oxygen non-stoichiometry (as well as various structural defects).

(iv) It is somewhat paradoxical (or counter-intuitive) that an invariant (more specifically, with respect to $\theta$) relationship, such as Equations (81) and (82) equating the ratio of mode I to mode II stress intensity factors or its far-field loadings counterpart (long range order) to negative times the ratio of the corresponding driving stresses, $\sigma_y(r,\theta,z)|_{Mode\ I} / \tau_{xy}(r,\theta,z)_{Mode\ II}$ (short range interaction) for the case of an orthorhombic crystal with complex roots (valid for A > 1 (or, equivalently, $\kappa > \sqrt{c_{22}/c_{11}}$)), guarantees the fact of the (010)[001] $\times$ [100] cleavage system being difficult, thus further concomitant in ensuring crack deflection or turning from this difficult cleavage system onto a nearby available easy one, violating the self-similarity of crack growth or propagation in the process.

(v) In contrast to (iv), again counter-intuitively, a non-invariant (more specifically, with respect to $\theta$), relationship, such as Equations (83) and (84) or (127) resulting from the degenerate case (A = 1 or, equivalently, $\kappa = 1$) or imaginary roots (valid for A < 1 or, equivalently, $\kappa < \sqrt{c_{22}/c_{11}}$), guarantees the cleavage system (010)[001] $\times$ [100] being easy, thus ensuring the self-similarity (i.e., $\theta$ invariance) of crack growth or propagation in the process.

(vi) Similarity or dissimilarity of the present asymptotic solutions for a (010)[001] $\times$ [100] cleavage system involving complex (A > 1 or, equivalently, $\kappa > \sqrt{c_{22}/c_{11}}$) and imaginary roots (A < 1 or, equivalently, $\kappa < \sqrt{c_{22}/c_{11}}$) with their isotropic (A = $\kappa$ = 1) counterparts can lead to a sufficient condition for determination of a cleavage system being easy or difficult for crack propagation.

(vii) The dimensionless parameters, such as the anisotropic ratio, A, or, equivalently, the normalized elastic parameter, $\kappa$, can serve as the Holy Grail quantity for an a priori determination of the status of a cleavage system to be easy or difficult, very much akin to Reynold's number for fluid flow problems, crossing a critical value which signifies transition from one regime to another. Here, the anisotropic ratio, A, or, equivalently, normalized elastic parameter, $\kappa$, for a (010)[001] $\times$ [100] cleavage system, crossing the critical value of 1 or $\sqrt{c_{22}/c_{11}}$, respectively, signifies transitioning from self-similar crack growth or propagation to crack deflection or turning from a difficult cleavage system onto a nearby easy one.

(viii) Just as the introduction of Reynold's number facilitated the design and setting up of experiments in addition to experimental verification of analytical and computational solutions in fluid dynamics, the accuracy and efficacy of the available test results on elastic constants of YBCO single crystals, measured by modern experimental techniques with resolutions at the atomic scale, or nearly so, such as X-Ray diffraction, the ultrasound technique, neutron diffraction/scattering, Brillouin spectroscopy/scattering, resonant ultrasound spectroscopy, and the like, is assessed with a powerful theoretical analysis on crack path stability/instability, in part based on a dimensionless parameter, such as the planar anisotropic ratio, A.

(ix) Experimental determination of surface energy, $G_i$, can sometimes be notoriously challenging, due to the presence of micro-to-nano scale defects, such as porosity, dislocation, twin boundaries, misalignment of bonds with respect to the loading axis, and the like. In contrast, the above-derived invariant relationship, (38), requires only measurement of strains or stresses at a point for a given far-field loading, which are, relatively speaking, much easier in comparison to the determination of surface energies.

(x) The planar anisotropic ratio, A, or, equivalently, the normalized elastic parameter, $\kappa$, for YBCO * is larger than 1 or $\sqrt{c_{22}/c_{11}}$, respectively, giving rise to complex roots (of the characteristic equation) for a (010)[001] $\times$ [100] through-crack, weakening a YBCO mono-crystalline orthorhombic plate. The same is true for a $(\bar{1}00)$[001] $\times$ [010] crack. These results predict that (010) and $(\bar{1}00)$ are difficult cleavage planes, which are in contradiction with the experimental observations.

(xi) Only for YBCO ***, all the cleavage systems are predicted to be easy, which is in agreement with the experimentally observed fracture characteristics, thus ensuring that a reasonably accurate complete set of nine experimentally determined elastic constants has been arrived at, by employing the present theoretical fracture mechanic approach.

(xii) For tetragonal YBCO$^T$, all six cleavage systems investigated here are found to be difficult, thus completely invalidating the values of the corresponding experimentally determined elastic constants reported by Reichard et al. [67].

(xiii) Finally, generally unavailable results, pertaining to the through-thickness variations of stress intensity factors and energy release rates for a crack corresponding to symmetric and skew-symmetric hyperbolic cosine loads that also satisfy the boundary conditions on the top and bottom surfaces of an orthorhombic mono-crystalline plate under investigation, bridge a longstanding gap in the stress singularity/fracture mechanics Literature.

**Funding:** This research received no external funding.

**Institutional Review Board Statement:** Not applicable.

**Informed Consent Statement:** Not applicable.

**Data Availability Statement:** Not applicable.

**Conflicts of Interest:** The author declares no conflict of interest.

## Appendix A. Singular Stress Fields in the Vicinity of a (010)[001] Through-Crack Front Propagating under Mode I (Extension/Bending) and Mode II (Sliding Shear/Twisting) in [100] Direction

The cleavage plane considered is (010) (Figure 3). Here, the *z*-axis is placed along the straight crack front, while the coordinates x, y, are used to define the directions along the length of the crack and transverse to it, respectively, in the plane of the plate. u, v, and w represent the components of the displacements in the x, y, and z directions, respectively.

The stress-strain relations for an orthorhombic single crystal are given as follows:

$$
\begin{Bmatrix} \sigma_x \\ \sigma_y \\ \sigma_z \\ \tau_{yz} \\ \tau_{xz} \\ \tau_{xy} \end{Bmatrix} = \begin{bmatrix} c_{11} & c_{12} & c_{13} & 0 & 0 & 0 \\ c_{12} & c_{22} & c_{23} & 0 & 0 & 0 \\ c_{13} & c_{23} & c_{33} & 0 & 0 & 0 \\ 0 & 0 & 0 & c_{44} & 0 & 0 \\ 0 & 0 & 0 & 0 & c_{55} & 0 \\ 0 & 0 & 0 & 0 & 0 & c_{66} \end{bmatrix} \begin{Bmatrix} \varepsilon_x \\ \varepsilon_y \\ \varepsilon_z \\ \gamma_{yz} \\ \gamma_{xz} \\ \gamma_{xy} \end{Bmatrix}. \tag{A1}
$$

where $c_{ij}$, i, j = 1, ..., 6, denotes the elastic stiffness constants of an orthorhombic mono-crystalline plate. $\sigma_x$, $\sigma_y$, $\sigma_z$ represent the normal stresses, and $\tau_{xy}$, $\tau_{xz}$, $\tau_{yz}$ denote the shear stresses, while $\varepsilon_x$, $\varepsilon_y$, $\varepsilon_z$ denote normal strains, and $\gamma_{xy}$, $\gamma_{xz}$, $\gamma_{yz}$ represent the shear strains.

The three equilibrium equations for a linear elastic solid, made of an orthotropic/orthorhombic material, can be expressed in terms of the displacement components, u, v, and w, as follows:

$$
c_{11}\frac{\partial^2 u}{\partial x^2} + c_{66}\frac{\partial^2 u}{\partial y^2} + c_{55}\frac{\partial^2 u}{\partial z^2} + (c_{12}+c_{66})\frac{\partial^2 v}{\partial x \partial y} + (c_{13}+c_{55})\frac{\partial^2 w}{\partial x \partial z} = 0, \tag{A2}
$$

$$
(c_{12}+c_{66})\frac{\partial^2 u}{\partial x \partial y} + c_{66}\frac{\partial^2 v}{\partial x^2} + c_{22}\frac{\partial^2 v}{\partial y^2} + c_{44}\frac{\partial^2 v}{\partial z^2} + (c_{23}+c_{44})\frac{\partial^2 w}{\partial y \partial z} = 0, \tag{A3}
$$

$$
(c_{13}+c_{55})\frac{\partial^2 u}{\partial x \partial z} + (c_{23}+c_{44})\frac{\partial^2 v}{\partial y \partial z} + c_{55}\frac{\partial^2 w}{\partial x^2} + c_{44}\frac{\partial^2 w}{\partial y^2} + c_{33}\frac{\partial^2 w}{\partial z^2} = 0, \tag{A4}
$$

The characteristic equations for the coupled partial differential Equations (A2)–(A4) can be written as follows:

$$
p^4 + 2\chi p^2 + \frac{c_{11}}{c_{22}} = 0, \tag{A5}
$$

in which the normalized elastic parameter, $\kappa = 1/\chi$, is given by

$$
\kappa = \frac{1}{\chi} = \frac{2c_{22}c_{66}}{(c_{11}c_{22} - c_{12}^2 - 2c_{12}c_{66})} = \frac{2G_{12}(1-\nu_{13}\nu_{31})}{[E_1 - 2G_{12}(\nu_{12}+\nu_{32}\nu_{13})]}, \tag{A6}
$$

in which $E_1$ is Young's modulus in the x direction, $G_{12}$ is the shear modulus in the x-y plane, while $\nu_{12}$ is the major Poisson's ratio in the x-y plane. $\nu_{13}$ and $\nu_{31}$ denote the major and minor Poisson's ratios in the x-z plane, while $\nu_{32}$ represents the minor Poisson's ratio in the y-z plane. $\kappa$ can also be expressed in terms of the planar anisotropic ratio (in the x-y plane), A, as follows:

$$
\kappa = \frac{1}{\chi} = \frac{Ac_{22}}{\sqrt{c_{11}c_{22}} + c_{12}(1-A)}. \tag{A7}
$$

where *A* is defined as

$$
A = \frac{2c_{66}}{\sqrt{c_{11}c_{22}} - c_{12}}. \tag{A8}
$$

Equation (A5) has either (a) four complex or (b) four imaginary roots, depending on whether

$$\text{(a) } A > 1 \text{ or, equivalently, } \kappa > \sqrt{\frac{c_{22}}{c_{11}}} = \sqrt{\frac{E_2(1 - \nu_{13}\nu_{31})}{E_1(1 - \nu_{23}\nu_{32})}}, \tag{A9}$$

$$\text{(b) } A < 1 \text{ or, equivalently, } \kappa < \sqrt{\frac{c_{22}}{c_{11}}} = \sqrt{\frac{E_2(1 - \nu_{13}\nu_{31})}{E_1(1 - \nu_{23}\nu_{32})}}. \tag{A10}$$

A = 1, κ = 1 represents the degenerate isotropic material case, for which the solution is available in Chaudhuri and Xie [25].

For the extension-bending (mode I) and in-plane shear-twisting (mode II) loadings, it can easily be seen that for orthorhombic single crystals with A < 1 or, equivalently, $\kappa < \sqrt{c_{22}/c_{11}}$, the (010) plane is the easy cleavage plane, and [100] is the easy propagation direction. Conversely, A > 1 or, equivalently, $\kappa > \sqrt{c_{22}/c_{11}}$ yields complex roots, implying that neither (010) is the easy cleavage plane nor is [100] the easy propagation direction, and the crack will likely deviate from this plane and this direction under mode I/II loadings.

$$K_I(z^*) = \sigma_y^\infty \sqrt{\pi a} D_b(z^*), \tag{A11}$$

$$K_{II}(z^*) = \tau_{xy}^\infty \sqrt{\pi a} D_b(z^*). \tag{A12}$$

$$G_I(z^*) = \frac{\left(\sigma_y^\infty\right)^2 \pi a \sqrt{c_{11}c_{22}}}{\sqrt{2}(c_{11}c_{22} - c_{12}^2)} \sqrt{\sqrt{(c_{11}/c_{22})} + \chi} [D_b(z*)]^2, \tag{A13}$$

$$G_{II}(z^*) = \frac{\left(\tau_{xy}^\infty\right)^2 \pi a c_{22}}{\sqrt{2}(c_{11}c_{22} - c_{12}^2)} \sqrt{\sqrt{(c_{11}/c_{22})} + \chi} [D_b(z*)]^2. \tag{A14}$$

**Appendix B. Singular Stress Fields in the Vicinity of a (0$\overline{1}$0)[100] Through-Crack Front Weakening an Orthorhombic Single Crystal under Mode I (Extension/Bending) and Mode II (Sliding Shear/Twisting)**

The cleavage plane considered is (0$\overline{1}$0) (Figure 4). Here, the $z'$-axis is placed along the straight crack front, [100], while the coordinates $x'$ [001], $y'$ [0$\overline{1}$0] are used to define the directions along the length of the crack (propagation direction) and the direction transverse to it, respectively, in the middle plane of the plate. $u'$, $v'$ and $w'$ represent the components of the displacements in $x'$, $y'$, and $z'$ directions, respectively. The stress-strain relations for an orthorhombic single crystal are given by

$$\begin{Bmatrix} \sigma'_x \\ \sigma'_y \\ \sigma_z \\ \tau'_{yz} \\ \tau'_{xz} \\ \tau'_{xy} \end{Bmatrix} = \begin{bmatrix} c'_{11} & c'_{12} & c'_{13} & 0 & 0 & 0 \\ c'_{12} & c'_{22} & c'_{23} & 0 & 0 & 0 \\ c'_{13} & c'_{23} & c'_{33} & 0 & 0 & 0 \\ 0 & 0 & 0 & c'_{44} & 0 & 0 \\ 0 & 0 & 0 & 0 & c'_{55} & 0 \\ 0 & 0 & 0 & 0 & 0 & c'_{66} \end{bmatrix} \begin{Bmatrix} \varepsilon'_x \\ \varepsilon'_y \\ \varepsilon_z \\ \gamma'_{yz} \\ \gamma'_{xz} \\ \gamma'_{xy} \end{Bmatrix}, \tag{A15}$$

where $c'_{ij}$, i, j = 1, 2, 6, denote the elastic stiffness constants with respect to the rotated coordinate system, $x'$, $y'$ (obtained by rotation of 90o about the $z$-axis):

$$c'_{11} = c_{33}, \quad c'_{12} = c_{23}, \quad c'_{13} = c_{13}, \quad c'_{22} = c_{22}, \quad c'_{23} = c_{12}, \quad c'_{33} = c_{11},$$
$$c'_{44} = c_{66}, \quad c'_{55} = c_{55}, \quad c'_{66} = c_{44}. \tag{A16}$$

The three equilibrium equations for a linear elastic orthotropic/orthorhombic solid can now be expressed in terms of the displacement functions, $u'$, $v'$, and w, as follows:

$$c'_{11}\frac{\partial^2 u'}{\partial x'^2} + c'_{66}\frac{\partial^2 u'}{\partial y'^2} + c'_{55}\frac{\partial^2 u'}{\partial z^2} + (c'_{12} + c'_{66})\frac{\partial^2 v'}{\partial x'\partial y'} + (c'_{13} + c'_{55})\frac{\partial^2 w}{\partial x'\partial z} = 0, \tag{A17}$$

$$(c'_{12} + c'_{66})\frac{\partial^2 u'}{\partial x'\partial y'} + c'_{66}\frac{\partial^2 v'}{\partial x'^2} + c'_{22}\frac{\partial^2 v'}{\partial y'^2} + c'_{44}\frac{\partial^2 v'}{\partial z^2} + (c'_{23} + c'_{44})\frac{\partial^2 w}{\partial y'\partial z} = 0, \tag{A18}$$

$$(c'_{13} + c'_{55})\frac{\partial^2 u'}{\partial x'\partial z} + (c'_{23} + c'_{44})\frac{\partial^2 v'}{\partial y'\partial z} + c'_{55}\frac{\partial^2 w}{\partial x'^2} + c'_{44}\frac{\partial^2 w}{\partial y'^2} + c'_{33}\frac{\partial^2 w}{\partial z^2} = 0, \tag{A19}$$

The characteristic equations for the coupled partial differential Equations (A17)–(A19) can be written as follows:

$$p^4 + 2\chi' p^2 + \frac{c'_{11}}{c'_{22}} = 0, \tag{A20}$$

in which the normalized elastic parameter, $\kappa' = 1/\chi'$, is given by

$$\kappa' = \frac{1}{\chi'} = \frac{2c'_{22}c'_{66}}{\left(c'_{11}c'_{22} - c'^2_{12} - 2c'_{12}c'_{66}\right)} = \frac{2G_{12}(1 - \nu_{23}\nu_{32})}{[E_2 - 2G_{12}(\nu_{21} + \nu_{31}\nu_{23})]}, \tag{A21}$$

In which $E_2$ is y-direction Young's modulus, and $G_{12}$ is the shear modulus in the x-y plane, while $\nu_{21}$ is the minor Poisson's ratio in the x-y plane. $\nu_{31}$ denotes the minor Poisson's ratio in the x-z plane, while $\nu_{23}$ and $\nu_{32}$ represent the major and minor Poisson's ratios, respectively, in the y-z plane. $\kappa'$. can also be expressed in terms of the planar anisotropic ratio (in the $x'$ [001]-$y'$ [0$\overline{1}$0] plane), $A'$, as follows:

$$\kappa' = \frac{1}{\chi'} = \frac{A'c'_{22}}{\left(\sqrt{c'_{11}c'_{22}} + c'_{12}\right) - A'c'_{12}} = \frac{A'c_{22}}{\left[\sqrt{c_{22}c_{33}} + c_{23}(1 - A')\right]}. \tag{A22}$$

where $A'$ is defined as

$$A' = \frac{2c_{44}}{\sqrt{c_{22}c_{33}} - c_{23}}. \tag{A23}$$

Equation (A20) has either (a) four complex or (b) four imaginary roots, depending on whether

$$\text{(a) } A' > 1, \text{ or, equivalently, } \kappa' > \sqrt{\frac{c'_{22}}{c'_{11}}} = \sqrt{\frac{c_{22}}{c_{33}}} = \sqrt{\frac{E_2(1 - \nu_{13}\nu_{31})}{E_3(1 - \nu_{12}\nu_{21})}}, \tag{A24}$$

$$\text{(b) } A' < 1, \text{ or, equivalently, } \kappa' < \sqrt{\frac{c'_{22}}{c'_{11}}} = \sqrt{\frac{c_{22}}{c_{33}}} = \sqrt{\frac{E_2(1 - \nu_{13}\nu_{31})}{E_3(1 - \nu_{12}\nu_{21})}}, \tag{A25}$$

$A' = 1$, $\kappa' = 1$ represents the degenerate isotropic material case, for which the solution is available in Chaudhuri and Xie [25].

For the extension-bending (mode I) and in-plane shear-twisting (mode II) loadings, it can easily be seen that for orthorhombic single crystals with $A' < 1$, or, equivalently, $\kappa' < \sqrt{c'_{22}/c'_{11}} = \sqrt{c_{22}/c_{33}}$, the (0$\overline{1}$0) plane is the easy cleavage plane, and [001] is the easy propagation direction. Conversely, $A' > 1$, or, equivalently, $\kappa' > \sqrt{c_{22}/c_{33}}$ yields complex roots, implying that neither (0$\overline{1}$0) is the easy cleavage plane nor is [001] the easy propagation direction, and the crack will likely deviate from this plane and this direction under mode I/II loadings.

$$K'_I\left(z'^*\right) = \sigma^\infty_{y'}\sqrt{\pi a}D_b\left(z'^*\right), \tag{A26}$$

$$K'_{II}\left(z'^*\right) = \tau^\infty_{x'y'}\sqrt{\pi a}D_b\left(z'^*\right). \tag{A27}$$

$$G'_I\left(z'^*\right) = \frac{\left(\sigma^\infty_{y'}\right)^2\pi a\sqrt{c_{33}c_{22}}}{\sqrt{2}(c_{33}c_{22} - c_{23}{}^2)}\sqrt{\sqrt{(c_{33}/c_{22})} + \chi'}\left[D_b(z'*)\right]^2, \tag{A28}$$

$$G'_{II}\left(z'^*\right) = \frac{\left(\tau^\infty_{x'y'}\right)^2\pi a c_{22}}{\sqrt{2}(c_{33}c_{22} - c_{23}{}^2)}\sqrt{\sqrt{(c_{33}/c_{22})} + \chi'}\left[D_b(z'*)\right]^2. \tag{A29}$$

## Appendix C. Singular Stress Fields in the Vicinity of a $(\bar{1}00)[001]$ Through-Crack Front Propagating under Mode I (Extension/Bending) and Mode II (Sliding Shear/Twisting) in [010] Direction

The cleavage plane considered is $(\bar{1}00)$ (Figure 5). Here, the *z*-axis is placed along the straight crack front, [001], while the coordinates $x''$ [010], $y''$ [$\bar{1}00$] are used to define the directions along the length of the crack (propagation direction) and the direction transverse to it, respectively, in the middle plane of the plate. $u''$, $v''$ and $w''$ represent the components of the displacements in $x''$, $y''$, and $z$ directions, respectively. The stress-strain relations for an orthorhombic single crystal are given by

$$\begin{Bmatrix} \sigma''_x \\ \sigma''_y \\ \sigma''_z \\ \tau''_{yz} \\ \tau''_{xz} \\ \tau''_{xy} \end{Bmatrix} = \begin{bmatrix} c''_{11} & c''_{12} & c''_{13} & 0 & 0 & 0 \\ c''_{12} & c''_{22} & c''_{23} & 0 & 0 & 0 \\ c''_{13} & c''_{23} & c''_{33} & 0 & 0 & 0 \\ 0 & 0 & 0 & c''_{44} & 0 & 0 \\ 0 & 0 & 0 & 0 & c''_{55} & 0 \\ 0 & 0 & 0 & 0 & 0 & c''_{66} \end{bmatrix} \begin{Bmatrix} \varepsilon''_x \\ \varepsilon''_y \\ \varepsilon''_z \\ \gamma''_{yz} \\ \gamma''_{xz} \\ \gamma''_{xy} \end{Bmatrix}, \tag{A30}$$

where $c''_{ij}$, i, j = 1, 2, 6, denote the elastic stiffness constants with respect to the rotated coordinate system, $x''$, $y''$, $z''$ (obtained by rotation of 90° about the -*y*-axis):

$$c''_{11} = c_{22}, \quad c''_{12} = c_{12}, \quad c''_{13} = c_{23}, \quad c''_{22} = c_{11}, \quad c''_{23} = c_{13},$$
$$c''_{33} = c_{33}, \quad c''_{44} = c_{55}, \quad c''_{55} = c_{44}, \quad c''_{66} = c_{66}. \tag{A31}$$

The three equilibrium equations for a linear elastic orthotropic/orthorhombic solid can now be expressed in terms of the displacement functions, $u''$, $v''$, and $w''$, as follows:

$$c''_{11}\frac{\partial^2 u''}{\partial x''^2} + c''_{66}\frac{\partial^2 u''}{\partial y''^2} + c''_{55}\frac{\partial^2 u''}{\partial z''^2} + (c''_{12} + c''_{66})\frac{\partial^2 v''}{\partial x''\partial y''} + (c''_{13} + c''_{55})\frac{\partial^2 w''}{\partial x''\partial z''} = 0, \tag{A32}$$

$$(c''_{12} + c''_{66})\frac{\partial^2 u''}{\partial x''\partial y''} + c''_{66}\frac{\partial^2 v''}{\partial x''^2} + c''_{22}\frac{\partial^2 v''}{\partial y''^2} + c''_{44}\frac{\partial^2 v''}{\partial z''^2} + (c''_{23} + c''_{44})\frac{\partial^2 w''}{\partial y''\partial z''} = 0, \tag{A33}$$

$$(c''_{13} + c''_{55})\frac{\partial^2 u''}{\partial x''\partial z''} + (c''_{23} + c''_{44})\frac{\partial^2 v''}{\partial y''\partial z''} + c''_{55}\frac{\partial^2 w''}{\partial x''^2} + c''_{44}\frac{\partial^2 w''}{\partial y''^2} + c''_{33}\frac{\partial^2 w''}{\partial z''^2} = 0, \tag{A34}$$

The characteristic equations for the coupled partial differential Equations (A32)–(A34) can be written as

$$p^4 + 2\chi'' p^2 + \frac{c''_{11}}{c''_{22}} = 0, \tag{A35}$$

in which the normalized elastic parameter, $\kappa'' = 1/\chi''$, is given by

$$\kappa'' = \frac{1}{\chi''} = \frac{2c''_{22}c''_{66}}{\left(c''_{11}c''_{22} - c''^2_{12} - 2c''_{12}c''_{66}\right)} = \frac{2c_{22}c_{44}}{\left(c_{22}c_{33} - c^2_{23} - 2c_{23}c_{44}\right)}$$
$$= \frac{2G_{23}(1 - \nu_{13}\nu_{31})}{[E_3 - 2G_{23}(\nu_{32} + \nu_{12}\nu_{31})]}, \tag{A36}$$

in which $E_3$ is z-direction Young's modulus, and $G_{23}$ is the shear modulus in the y-z plane, while $\nu_{32}$ denotes the minor Poisson's ratio in the y-z plane. $\nu_{12}$ is the major Poisson's ratio in the x-y plane, while $\nu_{13}$ and $\nu_{31}$ represent the major and minor Poisson's ratios, respectively, in the x-z plane. $\kappa''$ can also be expressed in terms of the planar anisotropic ratio (in the $x''$ [010]-$y''$ [$\bar{1}00$] plane), $A''$ as follows:
which finally yields

$$\kappa'' = \frac{1}{\chi''} = \frac{A''c''_{22}}{(\sqrt{c''_{11}c''_{22}} + c''_{12}) - A''c''_{12}} = \frac{A''c_{11}}{\sqrt{c_{11}c_{22}} + c_{12}(1 - A'')}, \tag{A37}$$

in which $A''$ is defined as follows:

$$A'' = \frac{2c_{66}}{\sqrt{c_{11}c_{22}} - c_{12}}, \tag{A38}$$

Equation (A35) has either (a) four complex or (b) four imaginary roots, depending on whether

$$\text{(a) } A'' > 1, \text{ or, equivalently, } \kappa'' > \sqrt{\frac{c''_{22}}{c''_{11}}} = \sqrt{\frac{c_{11}}{c_{22}}} = \sqrt{\frac{E_1(1 - \nu_{23}\nu_{32})}{E_2(1 - \nu_{13}\nu_{31})}}, \tag{A39}$$

or

$$\text{(b) } A'' < 1, \text{ or, equivalently, } \kappa'' < \sqrt{\frac{c''_{22}}{c''_{11}}} = \sqrt{\frac{c_{11}}{c_{22}}} = \sqrt{\frac{E_1(1 - \nu_{23}\nu_{32})}{E_2(1 - \nu_{13}\nu_{31})}}. \tag{A40}$$

$A'' = 1, \kappa'' = 1$ represents the degenerate isotropic material case, for which the solution is available in Chaudhuri and Xie [25].

For the extension-bending (mode I) and in-plane shear-twisting (mode II) loadings, it can easily be seen that for orthorhombic single crystals with $A'' < 1$, or, equivalently, $\kappa'' < \sqrt{c''_{22}/c''_{11}} = \sqrt{c_{11}/c_{22}}$, the (100) plane is the easy cleavage plane (and y [010]-direction is the easy propagation direction). Conversely, $A'' > 1$ or, equivalently, $\kappa'' > \sqrt{c_{11}/c_{22}}$ yields complex roots, implying that neither (100) is the easy cleavage plane nor is [010] the easy propagation direction, and the crack will likely deviate from this plane and this direction under mode I/II loadings.

$$K''_I(z''^*) = \sigma_{y''}^\infty \sqrt{\pi a} D_b(z''^*), \tag{A41}$$

$$K''_{II}(z''^*) = \tau_{x''y''}^\infty \sqrt{\pi a} D_b(z''^*). \tag{A42}$$

$$G''_I(z''^*) = \frac{\left(\sigma_{y''}^\infty\right)^2 \pi a \sqrt{c_{11}c_{22}}}{\sqrt{2}(c_{11}c_{22} - c_{12}{}^2)} \sqrt{\sqrt{(c_{22}/c_{11})} + \chi''} [D_b(z''*)]^2, \tag{A43}$$

$$G''_{II}(z''^*) = \frac{\left(\tau_{x''y''}^\infty\right)^2 \pi a c_{22}}{\sqrt{2}(c_{11}c_{22} - c_{12}{}^2)} \sqrt{\sqrt{(c_{22}/c_{11})} + \chi''} [D_b(z''*)]^2. \tag{A44}$$

For the special case of a tetragonal single crystal, the above energy release rates reduce to

$$G''_I(z''^*) = \frac{\left(\sigma_{y''}^\infty\right)^2 \pi a c_{11}}{\sqrt{2}(c_{11}{}^2 - c_{12}{}^2)} \sqrt{1 + \chi''} [D_b(z''*)]^2, \tag{A45}$$

$$G''_{II}(z''^*) = \frac{\left(\tau_{x''y''}^\infty\right)^2 \pi a c_{22}}{\sqrt{2}(c_{11}{}^2 - c_{12}{}^2)} \sqrt{1 + \chi''} [D_b(z''*)]^2. \tag{A46}$$

**Appendix D. Singular Stress Fields in the Vicinity of a (100)[010] Through-Crack Front Propagating under Mode I (Extension/Bending) and Mode II (Sliding Shear/Twisting) in [001] Direction**

The cleavage plane considered is (100). Here, the $\overline{z}$-axis is placed along the straight crack front, [010], while the coordinates $\overline{x}$ [001], $\overline{y}$ [100] are used to define the directions along the length of the crack (propagation direction) and the direction transverse to it, respectively, in the middle plane of the plate. $\overline{u}$, $\overline{v}$ and $\overline{w}$ represent the components of the displacements in $\overline{x}$ [001], $\overline{y}$ [100], and $\overline{z}$ [010] directions, respectively. The stress-strain relations for an orthorhombic single crystal are given by

$$
\begin{Bmatrix} \overline{\sigma}_x \\ \overline{\sigma}_y \\ \overline{\sigma}_z \\ \overline{\tau}_{yz} \\ \overline{\tau}_{xz} \\ \overline{\tau}_{xy} \end{Bmatrix} = \begin{bmatrix} \overline{c}_{11} & \overline{c}_{12} & \overline{c}_{13} & 0 & 0 & 0 \\ \overline{c}_{12} & \overline{c}_{22} & \overline{c}_{23} & 0 & 0 & 0 \\ \overline{c}_{13} & \overline{c}_{23} & \overline{c}_{33} & 0 & 0 & 0 \\ 0 & 0 & 0 & \overline{c}_{44} & 0 & 0 \\ 0 & 0 & 0 & 0 & \overline{c}_{55} & 0 \\ 0 & 0 & 0 & 0 & 0 & \overline{c}_{66} \end{bmatrix} \begin{Bmatrix} \overline{\varepsilon}_x \\ \overline{\varepsilon}_y \\ \overline{\varepsilon}_z \\ \overline{\gamma}_{yz} \\ \overline{\gamma}_{xz} \\ \overline{\gamma}_{xy} \end{Bmatrix},
\tag{A47}
$$

where $\overline{c}_{ij}$, i, j = 1, 2, 6, denote the elastic stiffness constants with respect to the transformed coordinate system, $\overline{x}$ [001], $\overline{y}$ [100], and $\overline{z}$ [010]

$$
\begin{aligned}
&\overline{c}_{11} = c_{33}, \quad \overline{c}_{12} = c_{13}, \quad \overline{c}_{13} = c_{23}, \quad \overline{c}_{22} = c_{11}, \quad \overline{c}_{23} = c_{12}, \quad \overline{c}_{33} = c_{22}, \\
&\qquad\quad \overline{c}_{44} = c_{66}, \quad \overline{c}_{55} = c_{44}, \quad \overline{c}_{66} = c_{55}.
\end{aligned}
\tag{A48}
$$

The three equilibrium equations for a linear elastic orthotropic/orthorhombic solid can now be expressed in terms of the displacement functions, $\overline{u}$, $\overline{v}$ and $\overline{w}$, as follows:

$$
c_{33}\frac{\partial^2 \overline{u}}{\partial \overline{x}^2} + c_{55}\frac{\partial^2 \overline{u}}{\partial \overline{y}^2} + c_{44}\frac{\partial^2 \overline{u}}{\partial \overline{z}^2} + (c_{13} + c_{55})\frac{\partial^2 \overline{v}}{\partial \overline{x}\partial\overline{y}} + (c_{23} + c_{44})\frac{\partial^2 \overline{w}}{\partial x' \partial z} = 0,
\tag{A49}
$$

$$
(c_{13} + c_{55})\frac{\partial^2 \overline{u}}{\partial \overline{x}\partial\overline{y}} + c_{55}\frac{\partial^2 \overline{v}}{\partial \overline{x}^2} + c_{11}\frac{\partial^2 \overline{v}}{\partial \overline{y}^2} + c_{66}\frac{\partial^2 \overline{v}}{\partial \overline{z}^2} + (c_{12} + c_{66})\frac{\partial^2 \overline{w}}{\partial \overline{y}\partial\overline{z}} = 0,
\tag{A50}
$$

$$
(c_{23} + c_{44})\frac{\partial^2 \overline{u}}{\partial \overline{x}\partial\overline{z}} + (c_{12} + c_{66})\frac{\partial^2 \overline{v}}{\partial \overline{y}\partial\overline{z}} + c_{44}\frac{\partial^2 \overline{w}}{\partial \overline{x}^2} + c_{66}\frac{\partial^2 \overline{w}}{\partial \overline{y}^2} + c_{22}\frac{\partial^2 \overline{w}}{\partial \overline{z}^2} = 0,
\tag{A51}
$$

The characteristic equations for the coupled partial differential Equation (121) can be written as follows:

$$
p^4 + 2\overline{\chi}p^2 + \frac{c_{33}}{c_{11}} = 0,
\tag{A52}
$$

in which the normalized elastic parameter, $\overline{\kappa} = 1/\overline{\chi}$, is given by

$$
\overline{\kappa} = \frac{1}{\overline{\chi}} = \frac{2c_{11}c_{55}}{(c_{11}c_{33} - c_{13}^2 - 2c_{13}c_{55})} = \frac{2G_{13}(1 - \nu_{12}\nu_{21})}{[E_3 - 2G_{13}(\nu_{13} + \nu_{23}\nu_{12})]},
\tag{A53}
$$

in which $E_3$ is z-direction Young's modulus, and $G_{13}$ is the shear modulus in the x-z plane, while $\nu_{12}$ and $\nu_{21}$ denote the major and minor Poisson's ratios, respectively in the x-y plane. $\nu_{13}$ is the major Poisson's ratio in the x-z plane, while $\nu_{23}$ represents the major and minor Poisson's ratio in the y-z plane. $\overline{\kappa}$ can also be expressed in terms of the planar anisotropic ratio (in the $\overline{x}$ [001]-$\overline{y}$ [100] plane), $\overline{A}$, as follows:

$$
\overline{\kappa} = \frac{1}{\overline{\chi}} = \frac{\overline{A}\overline{c}_{22}}{\left(\sqrt{\overline{c}_{11}\overline{c}_{22}} + \overline{c}_{12}\right) - \overline{A}\overline{c}_{12}} = \frac{\overline{A}c_{11}}{\sqrt{c_{11}c_{33}} + c_{13}\left(1 - \overline{A}\right)}.
\tag{A54}
$$

where $\overline{A}$ is defined as

$$\overline{A} = \frac{2\overline{c}_{66}}{\sqrt{\overline{c}_{11}\overline{c}_{22}} - \overline{c}_{12}} = \frac{2c_{55}}{\sqrt{c_{11}c_{33}} - c_{13}}. \tag{A55}$$

Equation (A52) has either (a) four complex or (b) four imaginary roots, depending on whether

$$\text{(a) } \overline{A} > 1, \text{ or, equivalently, } \overline{\kappa} > \sqrt{\frac{c_{11}}{c_{33}}} = \sqrt{\frac{E_1(1 - \nu_{23}\nu_{32})}{E_3(1 - \nu_{12}\nu_{21})}}, \tag{A56}$$

or

$$\text{(b) } \overline{A} < 1, \text{ or, equivalently, } \overline{\kappa} < \sqrt{\frac{c_{11}}{c_{33}}} = \sqrt{\frac{E_1(1 - \nu_{23}\nu_{32})}{E_3(1 - \nu_{12}\nu_{21})}}. \tag{A57}$$

$\overline{A} = 1$, $\overline{\kappa} = 1$ represents the degenerate isotropic material case, for which the solution is available in Chaudhuri and Xie [25].

For the extension-bending (mode I) and in-plane shear-twisting (mode II) loadings, it can easily be seen that for orthorhombic single crystals with $\overline{A} < 1$, or, equivalently, $\overline{\kappa} < \sqrt{c_{11}/c_{33}}$, the (010) plane is the easy cleavage plane (and z [001] direction is the easy propagation direction). Conversely, $\overline{A} > 1$, or, equivalently, $\overline{\kappa} > \sqrt{c_{11}/c_{33}}$ yields complex roots, implying that neither (010) is the easy cleavage plane nor is [001] the easy propagation direction, and the crack will likely deviate from this plane and this direction under mode I/II loadings.

$$\overline{K}_I(\overline{z}^*) = \sigma_{\overline{y}}^\infty \sqrt{\pi a} D_b(\overline{z}^*), \tag{A58}$$

$$\overline{K}_{II}(\overline{z}^*) = \tau_{\overline{xy}}^\infty \sqrt{\pi a} D_b(\overline{z}^*). \tag{A59}$$

$$\overline{G}_I(\overline{z}^*) = \frac{\left(\sigma_{\overline{y}}^\infty\right)^2 \pi a \sqrt{c_{11}c_{33}}}{\sqrt{2}(c_{11}c_{33} - c_{13}{}^2)} \sqrt{\sqrt{(c_{33}/c_{11})} + \overline{\chi}} [D_b(\overline{z}^*)]^2. \tag{A60}$$

$$\overline{G}_{II}(\overline{z}^*) = \frac{\left(\tau_{\overline{xy}}^\infty\right)^2 \pi a c_{11}}{\sqrt{2}(c_{11}c_{33} - c_{13}{}^2)} \sqrt{\sqrt{(c_{33}/c_{11})} + \overline{\chi}} [D_b(\overline{z}^*)]^2. \tag{A61}$$

**Appendix E. Singular Stress Fields in the Vicinity of a (001)[0$\bar{1}$0]. Through-Crack Front Propagating under Mode I (Extension/Bending) and Mode II (Sliding Shear/Twisting) in [100] Direction**

The cleavage plane considered is (001). Here, the $\hat{z}$-axis is placed along the straight crack front, [0$\bar{1}$0], while the coordinates $\hat{x}$ [100], $\hat{y}$ [001] are used to define the directions along the length of the crack (propagation direction) and the direction transverse to it, respectively, in the middle plane of the plate. $\hat{u}$, $\hat{v}$ and $\hat{w}$ represent the components of the displacements in $\hat{x}$ [100], $\hat{y}$ [001], and $\hat{z}$ [0$\bar{1}$0] directions, respectively. The stress-strain relations for an orthorhombic single crystal are given by

$$\begin{Bmatrix} \hat{\sigma}_x \\ \hat{\sigma}_y \\ \hat{\sigma}_z \\ \hat{\tau}_{yz} \\ \hat{\tau}_{xz} \\ \hat{\tau}_{xy} \end{Bmatrix} = \begin{bmatrix} c_{11} & c_{13} & c_{12} & 0 & 0 & 0 \\ c_{13} & c_{33} & c_{23} & 0 & 0 & 0 \\ c_{12} & c_{23} & c_{22} & 0 & 0 & 0 \\ 0 & 0 & 0 & c_{44} & 0 & 0 \\ 0 & 0 & 0 & 0 & c_{66} & 0 \\ 0 & 0 & 0 & 0 & 0 & c_{55} \end{bmatrix} \begin{Bmatrix} \hat{\varepsilon}_x \\ \hat{\varepsilon}_y \\ \hat{\varepsilon}_z \\ \hat{\gamma}_{yz} \\ \hat{\gamma}_{xz} \\ \hat{\gamma}_{xy} \end{Bmatrix}, \tag{A62}$$

The three equilibrium equations for a linear elastic orthotropic/orthorhombic solid can now be expressed in terms of the displacement functions, $\hat{u}$, $\hat{v}$ and $\hat{w}$, as follows:

$$c_{11}\frac{\partial^2 \hat{u}}{\partial \hat{x}^2} + c_{55}\frac{\partial^2 \hat{u}}{\partial \hat{y}^2} + c_{66}\frac{\partial^2 \hat{u}}{\partial \hat{z}^2} + (c_{13}+c_{55})\frac{\partial^2 \hat{v}}{\partial \hat{x}\partial \hat{y}} + (c_{12}+c_{66})\frac{\partial^2 \hat{w}}{\partial \hat{x}\partial \hat{z}} = 0, \tag{A63}$$

$$(c_{13}+c_{55})\frac{\partial^2 \hat{u}}{\partial \hat{x}\partial \hat{y}} + c_{55}\frac{\partial^2 \hat{v}}{\partial \hat{x}^2} + c_{33}\frac{\partial^2 \hat{v}}{\partial \hat{y}^2} + c_{44}\frac{\partial^2 \hat{v}}{\partial \hat{z}^2} + (c_{23}+c_{44})\frac{\partial^2 \hat{w}}{\partial \hat{y}\partial \hat{z}} = 0, \tag{A64}$$

$$(c_{13}+c_{55})\frac{\partial^2 \hat{u}}{\partial \hat{x}\partial \hat{y}} + c_{55}\frac{\partial^2 \hat{v}}{\partial \hat{x}^2} + c_{33}\frac{\partial^2 \hat{v}}{\partial \hat{y}^2} + c_{44}\frac{\partial^2 \hat{v}}{\partial \hat{z}^2} + (c_{23}+c_{44})\frac{\partial^2 \hat{w}}{\partial \hat{y}\partial \hat{z}} = 0, \tag{A65}$$

The characteristic equations for the coupled partial differential Equations (A63)–(A65) can be written as follows:

$$p^4 + 2\hat{\chi}p^2 + \frac{c_{11}}{c_{33}} = 0, \tag{A66}$$

in which the normalized elastic parameter, $\hat{\kappa} = 1/\hat{\chi}$, is given by

$$\hat{\kappa} = \frac{1}{\hat{\chi}} = \frac{2c_{33}c_{55}}{\left(c_{11}c_{33} - c_{13}^2 - 2c_{13}c_{55}\right)} = \frac{2G_{13}(1 - \nu_{23}\nu_{32})}{[E_3 - 2G_{13}(\nu_{31} + \nu_{21}\nu_{23})]}, \tag{A67}$$

in which $E_3$ is y-direction Young's modulus, and $G_{13}$ is the shear modulus in the x-z plane, while $\nu_{21}$ is the minor Poisson's ratio in the x-y plane. $\nu_{31}$ denotes the minor Poisson's ratio in the x-z plane, while $\nu_{23}$ and $\nu_{32}$ represent the major and minor Poisson's ratios, respectively, in the y-z plane. $\hat{\kappa}$ can also be expressed in terms of the planar anisotropic ratio (in the $\hat{x}$ [100]-$\hat{y}$ [001] plane), $\hat{A}$, as follows:

$$\hat{\kappa} = \frac{\hat{A}c_{33}}{\sqrt{c_{11}c_{33}} + c_{13}(1 - \hat{A})}, \tag{A68}$$

where $\hat{A}$ is defined as

$$\hat{A} = \frac{2c_{55}}{\sqrt{c_{11}c_{33}} - c_{13}}. \tag{A69}$$

Equation (A66) has either (a) four complex or (b) four imaginary roots, depending on whether

$$\text{(a) } \hat{A} > 1 \text{ or, equivalently, } \hat{\kappa} > \sqrt{\frac{c_{33}}{c_{11}}} = \sqrt{\frac{E_3(1 - \nu_{12}\nu_{21})}{E_1(1 - \nu_{23}\nu_{32})}}, \tag{A70}$$

or

$$\text{(b) } \hat{A} < 1 \text{ or, equivalently, } \hat{\kappa} < \sqrt{\frac{c_{33}}{c_{11}}} = \sqrt{\frac{E_3(1 - \nu_{12}\nu_{21})}{E_1(1 - \nu_{23}\nu_{32})}}, \tag{A71}$$

$\hat{A} = 1$, $\hat{\kappa} = 1$ represents the degenerate isotropic material case, for which the solution is available in Chaudhuri and Xie [25].

For the extension-bending (mode I) and in-plane shear-twisting (mode II) loadings, it can easily be seen that for orthorhombic single crystals with $\hat{A} < 1$ or, equivalently, $\hat{\kappa} < \sqrt{c_{33}/c_{11}}$, the (001) plane is the easy cleavage plane (and [100]-direction is the easy propagation direction). Conversely, $\hat{A} > 1$ or, equivalently, $\hat{\kappa} > \sqrt{c_{33}/c_{11}}$ yields complex roots, implying that neither (001) is the easy cleavage plane nor is [100] the easy propagation direction, and the crack will likely deviate from this plane and this direction under mode I/II loadings.

$$\hat{K}_I(\hat{z}^*) = \sigma_{\hat{y}}^\infty \sqrt{\pi a} D_b(\hat{z}^*), \tag{A72}$$

$$\hat{K}_{II}(\hat{z}^*) = \tau_{\hat{x}\hat{y}}^\infty \sqrt{\pi a} D_b(\hat{z}^*). \tag{A73}$$

$$\hat{G}_I(\hat{z}^*) = \frac{\left(\sigma_{\hat{y}}^\infty\right)^2 \pi a \sqrt{c_{11}c_{33}}}{\sqrt{2}(c_{11}c_{33} - c_{13}{}^2)} \sqrt{\sqrt{(c_{11}/c_{33})} + \hat{\chi}} [D_b(\hat{z}^*)]^2, \tag{A74}$$

$$\hat{G}_{II}(\hat{z}^*) = \frac{\left(\tau_{\hat{x}\hat{y}}^\infty\right)^2 \pi a c_{33}}{\sqrt{2}(c_{11}c_{33} - c_{13}{}^2)} \sqrt{\sqrt{(c_{11}/c_{33})} + \hat{\chi}} [D_b(\hat{z}^*)]^2. \tag{A75}$$

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
