# Peer review of "Employment of Fracture Mechanics Criteria for Accurate Assessment of the Full Set of Elastic Constants of Orthorhombic/Tetragonal Mono-Crystalline YBCO"

_2673-3161, doi:10.3390/applmech4020032_

Round 1

Reviewer 1 Report

1. The study was not concluded

2. I will suggest that the author validate their equation by experimenting and comparing results to calculate the experimental error.  

Reviewer 2 Report

The effect of elastic constants, cij, on the nature (easy or difficult) of a cleavage system in YBa2Cu3O7−δ is investigated in this article, by employing a three-dimensional eigenfunction expansion technique, based on separation of the thickness-variable and on a modified Frobenius type series expansion technique in conjunction with the Eshelby-Stroh formalism.

The article emphasizes in the mathematical description of the problem under investigation.

Authors should further highlight the innovation of the proposed work.

Authors should also verify results derived from the proposed formulations with existing literature findings.

The paper is very long (56 pages). The main problem is that is not easy for the reader to follow the paper, recognize the originality and contribution of the work. So, is suggested that the length of the paper is reduced drastically, and the innovation of the paper is better highlighted.
